# Neural Networks Performance Prediction using Weights and Gradients Analysis

**Michael Bohadana**                                                            *bohadana@post.bgu.ac.il*
*Faculty of Computer and Information Science*
*Ben-Gurion University of the Negev*

**Alon Schneider**                                                               *alonshn@post.bgu.ac.il*
*Faculty of Computer and Information Science*
*Ben-Gurion University of the Negev*

**Gilad Katz**                                                                      *giladkz@bgu.ac.il*
*Faculty of Computer and Information Science*
*Ben-Gurion University of the Negev*

**Reviewed on OpenReview:** *https://openreview.net/forum?id=51TWh8tlSy*

## Abstract

Neural network performance predictors are widely used to accelerate neural architecture search, but existing methods face a persistent trade-off: learning-based predictors require costly per-dataset initialization, while lightweight proxies are fast yet struggle to exploit prior experience and often degrade under dataset shift. We introduce NAP2, a hybrid performance predictor that models early training dynamics. NAP2 tracks the temporal evolution of layer-wise weight and gradient statistics over a small number of mini-batches, producing accurate rankings from as little as 100 mini-batches per candidate. Crucially, NAP2 supports cross-dataset reuse: a predictor trained on one dataset can be applied to another without fine-tuning, avoiding the re-initialization overhead incurred by many model-based approaches. Experiments on NAS-Bench-201 across CIFAR-10, CIFAR-100, and ImageNet16-120 show that NAP2 outperforms strong hybrid baselines across all evaluated initialization and query time configurations in the in-domain setting and delivers cost-effective cross-dataset transfer, outperforming established learning-curve and zero-cost baselines at short query times. We further demonstrate robustness to significant distribution shift, with a predictor trained on CIFAR-10 transferring effectively to SVHN. Our code and trained models are available at https://anonymous.4open.science/r/NAP2-6027/README.md.

## 1 Introduction

Neural network performance prediction aims to estimate the final performance of candidate architectures without fully training each one, and it is a key ingredient in scaling neural architecture search (NAS). Performance predictors are commonly compared under constraints on (i) predictor initialization cost (the compute required before deployment), (ii) per-architecture query cost, and (iii) ranking accuracy, typically measured via rank correlation.

Following the taxonomy of (White et al., 2021), existing predictors can be broadly grouped into: *(a)* model-based predictors that learn from previously evaluated architectures (Ji et al., 2024); *(b)* learning-curve predictors that extrapolate performance from partial training traces (Ding et al., 2025); *(c)* zero-cost predictors that compute training-free (or nearly training-free) proxies at/near initialization (Lee & Ham, 2024); and *(d)* weight-sharing approaches that evaluate many sub-networks using shared parameters (Zhang et al., 2023). Model-based predictors can yield low query cost after an expensive initialization phase, but they often require re-training or adaptation when moving to a new dataset, limiting their practical reuse under

dataset shift. Zero-cost proxies avoid initialization and minimize query time, but their static nature typically imposes a ceiling on ranking fidelity, since the predictor does not improve as more training signal becomes available. Learning-curve predictors can be accurate, but they generally require partial training per candidate, increasing query time.

This study focuses on cross-dataset reuse of performance predictors: given a predictor trained on a source dataset, the goal is to rank architectures on a target dataset with minimal per-architecture training and without target-specific predictor re-initialization. No existing method combines reusable learned prediction with early-training dynamics for cross-dataset ranking under limited query budgets (e.g., ≤100 mini-batches per candidate, ∼9 seconds on a GTX 1080 Ti). The main technical difficulty is that dataset shift changes both attainable accuracies and early optimization behavior, so predictors that rely on dataset-specific calibrations or purely static signals often degrade when transferred.

To address the limitations described above, we propose Neural Architectures Performance Prediction (NAP2), a hybrid predictor (in the sense of combining a learned predictor with learning-curve signals) that represents early training dynamics via the temporal evolution of layer-wise summary statistics of weights and gradients. Unlike static, architecture-only descriptors, NAP2 explicitly leverages short learning-curve information (e.g., the first few hundred mini-batches) while still amortizing experience through a learned predictor that can be reused across datasets. Empirically, we show that NAP2 can produce strong rankings from very short partial training runs (e.g., 100 mini-batches per candidate) and that predictors trained on one dataset can be applied to other datasets without fine-tuning while retaining competitive rank correlation.

We evaluate NAP2 on NAS-Bench-201 across CIFAR-10, CIFAR-100, and ImageNet16-120, demonstrating competitive results against strong hybrid baselines (LcSVR, Omni-seminas) under limited budgets, and cost-effective cross-dataset transfer—reusing a predictor trained on one dataset to rank architectures on another without fine-tuning—compared to established learning-curve (SoTL-E, LCE-m, LC-PFN) and zero-cost baselines (SynFlow, GradNorm, SNIP). NAP2 differs from learning-curve methods by amortizing predictor experience across architectures via source-dataset training, and from zero-cost proxies by improving with more partial training rather than imposing a static correlation ceiling. Our contributions are as follows:

- We introduce NAP2, a hybrid performance predictor that models early learning dynamics using time-indexed statistics of weights and gradients.

- We demonstrate cross-dataset reuse of a learned predictor without target-dataset fine-tuning, while using only short partial training per queried architecture.

- We demonstrate robustness to significant distribution shift, showing that a predictor trained on CIFAR-10 transfers effectively to SVHN without fine-tuning, outperforming partial-training baselines from the first epoch onward.

## 2 Related Work

Neural network performance prediction methods differ in computational cost, prediction accuracy, and data types used, but can all be evaluated by three metrics: *a)* **Initialization** (pre-use compute/time to train the predictor); *b)* **Query** (compute/time per prediction); and *c)* **Performance** (accuracy after training).

In White et al. (2021), the authors propose the following categorization for performance predictors:

**Model-based methods** fully train many architectures, extract meta-features, and train a model to predict performance, resulting in high initialization but low query times. Prediction methods include Bayesian optimization (Shi et al., 2020; Kandasamy et al., 2018), evolutionary algorithms (Xue et al., 2023; Xiao et al., 2024), graph neural networks (Gao et al., 2023; Li et al., 2022), and Gaussian processes (Li et al., 2020; Lee et al., 2017). Recent advances, like CAP (Ji et al., 2024), leverage context to require fewer annotations.

**Learning curve-based methods** predict final performance by extrapolating from partial learning curves or losses. Techniques fit parametric models to validation accuracies (Domhan et al., 2015), or use iterative

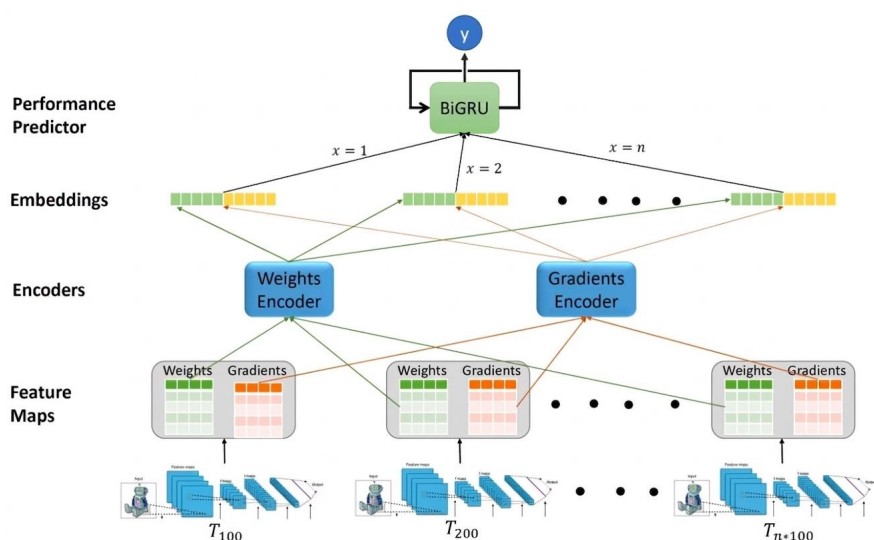

Figure 1: The proposed NAP2 architecture. At fixed batch intervals, we extract feature maps that represent the weights and gradients of the analyzed network. These feature maps are compressed using an encoder, and then provided as input to a recurrent sequence model. This model predicts the analyzed network's final performance. Note that because the recurrent model does not require a fixed-length input, NAP2 can produce predictions after any sequence length.

pruning as in Hyperband (Li et al., 2017) and SMAC3 (Lindauer et al., 2022). Bayesian neural networks can extrapolate partially trained curves (Klein et al., 2022). Recent works enhance these with dataset-agnostic learning (Ang et al., 2024), and structured neural representations for better modeling (Ding et al., 2024). However, query time is high, as each candidate generally needs partial training.

**Zero-cost methods** eliminate initialization and minimize query time by using single or few mini-batches, or even no training. Predictions rely on activation similarities (Mellor et al., 2021), or data-dependent/independent features; for example, synflow (Tanaka et al., 2020) computes a L1 path-norm from weights only. Approaches like Zen-NAS (Lin et al., 2021) require only a few forward passes for ranking, and Nas-Bench-Suite-Zero (Krishnakumar et al., 2022) provides a unified suite of such proxies. Some estimate performance using gradients or Jacobian measures (Abdelfattah et al., 2021).

**Weight-sharing methods** train a single over-parameterized network representing many sub-networks, improving efficiency. Notable approaches include DARTS (Liu et al., 2018) and its variants (Ye et al., 2022; Huang et al., 2023), ENAS (Pham et al., 2018) parameter sharing, ProxylessNAS' direct search (Cai et al., 2018), and TuNAS (Bender et al., 2020) for transferability. An approach using a single-path in the over-parameterized network further reduced search cost (Guo et al., 2020).

Our proposed approach falls into a category defined by White et al. (2021) as *hybrid*, combining model-based and learning curve-based techniques. While NAP2's query time is longer than zero-cost approaches, it delivers accurate predictions from as little as 100 mini-batches (about 9 seconds on a GTX 1080 Ti; see Section 4.5), and models trained on one dataset generalize to others without fine-tuning, positioning our method as a highly efficient learning curve-based predictor.

## 3 The Proposed Method

**Roadmap.** Section 3 develops the NAP2 pipeline in four stages: (i) snapshotting weights and gradients during early training of each candidate architecture; (ii) constructing fixed-shape feature maps from per-layer summary statistics (Section 3.2); (iii) compressing each feature map into a 128-dimensional embedding via a convolutional autoencoder (Section 3.3); and (iv) feeding the embedding sequence into a recurrent predictor (Section 3.4). The theoretical motivation for each design choice is given in Section 3.1.

**Overview.** NAP2 is presented in Figure 1. Starting with a set of architectures $|A|$, we analyze each architecture by repeatedly *snapshotting* its internal state early in training: every fixed interval of mini-batches, we log the network's weights and the corresponding gradients, and then transform these raw tensors into a fixed-shape representation. Concretely, at every snapshot we (i) extract layer-wise summary statistics from both weights and gradients, (ii) assemble them into two *feature maps* (one for weights, one for gradients) with a uniform tensor shape *independent of the architecture topology*, (iii) compress each feature map into a compact embedding using a convolutional autoencoder, and finally (iv) feed the resulting embedding sequence into a recurrent performance predictor (a bidirectional GRU with dual-path pooling). The predictor can be queried after any number of snapshots, enabling prediction from very short partial training runs. A visual summary of the full four-stage pipeline, including tensor dimensions at each stage, is provided in Appendix A.

Our approach is grounded in the hypothesis that the early training dynamics of neural networks provides a tractable proxy for the geometry of the loss landscape (Li et al., 2018). Features such as the evolution of weight magnitudes and gradient statistics reflect the sharpness or flatness of the loss surface – properties linked to generalization (Hochreiter & Schmidhuber, 1997). Recent work has shown that early-phase trajectories of weights and sharpness reveal universal, architecture-agnostic patterns predictive of later performance (Kalra et al., 2025). We hypothesize that capturing these *distinctive, yet broadly shared, aspects of learning dynamics* can support robust and generalizable performance prediction.

## 3.1 Theoretical Motivation

This section provides the theoretical grounding for NAP2's design choices. We organize the discussion around three questions: (i) why weight and gradient statistics should relate to generalization, (ii) why temporal dynamics carry more signal than static snapshots, and (iii) why these features should transfer across datasets. A detailed formal treatment is provided in Appendix G.

**Weight and gradient statistics as generalization proxies.** PAC-Bayes generalization theory establishes that the expected generalization gap of a stochastic neural network is controlled by a complexity term that depends on the parameter-space distance between the learned weights and a reference prior, typically measured through weight norms McAllester (1999); Neyshabur et al. (2017). More concretely, Bartlett et al. (2017) showed that for feedforward networks, generalization bounds scale with the product of per-layer spectral norms, while Neyshabur et al. (2015) derived bounds based on per-layer Frobenius norms. Our L1 and L2 norm meta-features track precisely these quantities at the layer level, providing the predictor with direct access to the terms that govern generalization in these frameworks. The variance and standard deviation features further characterize the distribution of norms across units within a layer, capturing intra-layer heterogeneity that aggregate norms alone miss.

Beyond norms, the distributional shape of weight matrices carries generalization-relevant information. Martin & Mahoney (2021) demonstrated that the heavy-tail structure of weight matrix spectra is predictive of test-set generalization, with networks exhibiting heavier-tailed weight distributions tending to generalize better. Our kurtosis and skewness features serve as tractable proxies for this tail behavior without requiring expensive spectral decompositions.

Gradient statistics complement the weight-based features by characterizing the local loss landscape geometry. The gradient covariance across training examples is closely related to the Fisher information matrix Amari (1998), whose eigenspectrum reflects the curvature of the loss surface. By tracking both weights – where the network is in parameter space – and gradients – what the landscape looks like at that location – NAP2 obtains a more complete picture of the optimization state than either signal alone.

**Temporal dynamics versus static snapshots.** A single snapshot of weight or gradient statistics reveals nothing about the trajectory the network is following. Training dynamics are highly structured: for example, the edge-of-stability phenomenon Cohen et al. (2021) shows that under gradient descent, sharpness evolves in a characteristic pattern constrained by the step size, rather than varying randomly. Kalra et al. (2025) further demonstrated that trajectories of weight norms and sharpness exhibit universal patterns across architectures and datasets. By feeding a temporal sequence of feature maps into a recurrent model, NAP2 captures the velocity and curvature of the optimization trajectory – not just its current position. This is strictly more

informative than any single-time-point measurement, and explains why NAP2 improves rapidly over the first several snapshots before plateauing (Section 6.1). Li et al. (2018) showed that network architecture significantly shapes loss landscape geometry, with skip connections and depth producing smoother surfaces – a structural property linked to generalization Hochreiter & Schmidhuber (1997); Keskar et al. (2016).

**Why cross-dataset transfer should work.** The key hypothesis underlying NAP2's transferability is that the optimization trajectory features we extract are more strongly governed by the network architecture than by the specific dataset. Choromanska et al. (2015) showed that under simplifying assumptions, the structure of the loss surface – including the distribution of critical points – is primarily determined by the network topology. Similarly, the implicit bias of gradient descent depends on architectural properties such as depth and layer type Gunasekar et al. (2018). While different datasets shift the loss surface, they do so largely in ways that affect absolute performance levels rather than the qualitative dynamics of how different architectures traverse the landscape. Our log-normalization of feature maps (Section 3.3) compresses the dynamic range of raw statistics without requiring per-dataset calibration, and since the predictor is evaluated by rank correlation, only relative orderings matter—not absolute predicted values. The empirical success of NAP2's cross-dataset transfer—including the challenging CIFAR-10 to SVHN shift (Section 5.3)—provides evidence that this hypothesis holds in practice. A more formal analysis of the conditions under which transfer succeeds is given in Appendix G.5.

### 3.2 Meta-Features Extraction and Representation

We use the term meta-features for layer-wise summary statistics (rather than raw weights or gradients), chosen so that the per-snapshot representation is (i) of fixed size independent of layer width, and (ii) interpretable in terms of the optimization-state quantities discussed in Section 3.1.

**Meta-Features Extraction.** For each analyzed architecture $A$, and at every snapshot time $t$, we extract *two parallel sets of layer-wise meta-features*: one computed from the current weights, and one computed from the current gradients. Meta-features are extracted from every *trainable* layer (dense or convolutional) at fixed training intervals; in our experiments we use an interval of 100 mini-batches, which we found to balance temporal resolution with storage overhead.

**Step 1: Produce a vector of unit-level values per layer.** For each layer, we first define the *unit* over which we will compute statistics, and produce a 1D list of values:

- **Dense layers:** the unit is a neuron. We compute one scalar per neuron by applying the chosen statistic to that neuron's incoming weights (or incoming gradients). Thus, a dense layer with 100 neurons yields 100 unit-level values.

- **Convolutional layers:** the unit is a filter (kernel). We compute one scalar (i.e., meta-feature value) per filter by applying the chosen statistic to that filter's kernel weights (or kernel gradients), flattened across input channels and spatial dimensions. Thus, a convolution with 128 filters yields 128 unit-level values (one per filter), regardless of the number of input channels.

**Step 2: Add a global layer value.** In addition to unit-level values, we also compute a *single global value* for the layer by applying the same statistic to all weights (or all gradients) in that layer. This global value summarizes the overall layer behavior and complements the unit-level values, which capture intra-layer heterogeneity.

**Step 3: Compute the meta-feature families.** We compute the following feature "families" for each layer, and do so identically for weights and gradients:

**1) General statistics meta-features.** This group consists of basic statistical operations: max, min, mean, variance, standard deviation, and median. Additionally, we compute five percentiles: the 0th (minimum), 25th, 50th (median), 75th, and 100th (maximum). Together, these five percentiles form the standard five-number summary, with the 25th and 75th percentiles capturing the interquartile range of the distribution.

These features reflect the overall scale and variability of the weights and gradients, helping to characterize general learning behavior across layers.

**2) Distribution-based meta-features.** We calculate the co-variance, kurtosis and skewness of each analyzed layer.

$$\text{Skewness - } \frac{\mu_3}{\sigma^3} \tag{1}$$

$$\text{Kurtosis - } \frac{\mu_4}{\sigma^4} \tag{2}$$

where $\mu_i = \frac{1}{N}\sum_k (w_{ij_k} - \overline{w_{ij}})^i$, $w_{ij}$ is the weight, $i$ is the index of the layer and $j$ is the index of the neuron, $\sigma$ is the standard deviation and $N$ is the number of weights of the current neuron. Skewness and kurtosis capture asymmetry and sharpness in the distribution, which relate to curvature and geometry of the local loss surface.

**3) Norm-based meta-features**. We calculate the $L_1$ and $L_2$ norms over the weights/gradients of each layer. We apply the following approach: for a layer with weights $W$ the norms are: $L_1 : ||W||_1 = \sum_i abs(w_i)$ ; $L_2 : ||W||_2 = \sqrt{\sum_i w_i^2}$. These features quantify update magnitudes, serving as a proxy for learning intensity and layer-wise signal strength.

Overall, we extract 24 meta-features per layer (12 for each of our two parallel sets). All statistical computations use NaN-aware implementations to ensure robust feature extraction even when layers contain inactive units. Formal definitions of all twelve meta-features, the sampling strategy, and the complete feature map construction procedure are given in Appendix B.

**Rationale for weights and gradients.** Weights-based features describe the evolving representation stored in the parameters, while gradients-based features describe the current optimization signal driving parameter updates. Combining both is intended to capture learning dynamics rather than relying on static architectural descriptors, supporting transfer across datasets by conditioning on *how* the model learns rather than only on *what* it is.

**Meta-Features Representation.** A key challenge is that architectures differ in depth and layer widths, while the predictor requires a fixed-size input. We therefore convert the per-layer meta-features at a snapshot into two fixed-shape *feature maps* (FM): one for weights and one for gradients. The feature maps are built with three explicit axes: (i) *layer index*, (ii) *values-per-feature*, and (iii) *feature type.*

**Step 1: Per-layer matrices of shape** $[100, 12]$**.** For each layer, and separately for weights and gradients, we create a matrix with 12 columns (one column per meta-feature) and 100 rows (one row per stored value of that feature). The 100 stored values are constructed as follows:

- For each meta-feature (each of the 12 columns), we place the **global layer value** in the first row.

- We then fill the remaining 99 rows with **unit-level values** (one per neuron/filter), using random sampling when the layer has more than 99 units.

- If a layer contains fewer than 99 units, we pad with zeros to reach exactly 100 rows.

This step ensures that every layer, regardless of width, is represented by a fixed $[100, 12]$ matrix, while still preserving a coarse notion of within-layer variability via the unit-level rows.

**Step 2: Stack layers into architecture-level feature maps of shape** $[65, 100, 12]$**.** We stack the per-layer $[100, 12]$ matrices along the depth axis to form an architecture-level feature map. For each architecture at each snapshot we thus obtain *two* feature maps (weights and gradients), each of shape $[65, 100, 12]$:

- We keep the *first 65 layers* of the architecture; deeper networks are truncated by discarding the remaining layers.

- Networks with fewer than 65 layers are padded with all-zero layers at the end.

After this construction, each architecture snapshot is represented by two tensors with identical shape, enabling a single downstream encoder and predictor to be trained across heterogeneous architectures.

### 3.3  Generating Feature Maps-based Embeddings

While informative, our feature maps have high dimensionality, which makes training the prediction model computationally expensive. A more condensed FM representation will enable us to deploy a smaller performance prediction model and use less training data.

We use convolutional autoencoders (CAE) to create embeddings for each feature map. Before encoding, we apply a log-normalization transform $x \mapsto \text{sign}(x) \cdot \log(1 + |x|)$ to each feature map. This compresses the dynamic range of the raw statistics without requiring per-dataset normalization parameters. The architecture of the CAE is symmetric, consisting of five layers with dimensions: 512, 256, 128, 256, and 512. We use the ReLU activation function and apply batch normalization after each layer, with MSE reconstruction loss. The embedding layer is a vector $v$, where $|v| = 128$. At inference time, only the two CAE encoders and the predictor are needed, totaling approximately 19M parameters. Full architectural specifications for both the CAE and the predictor, including layer configurations, training hyperparameters, and parameter counts, are provided in Appendix C.

### 3.4  Training the Prediction Model

Our training process has two stages. First, we train two autoencoders – one for each feature map type (weights-based and gradients-based) – using training samples generated by randomly selecting architectures and time steps and retrieving the corresponding FM. In the second stage, we train a performance prediction model: a bidirectional GRU (BiGRU) with dual-path pooling (concatenating the last hidden state with an attention-weighted context vector), followed by a dense layer and a Sigmoid output. The predictor is trained with L1 loss and a OneCycleLR schedule.

We augment each architecture in our training set with feature map embedding sequences of lengths 1 to 22, creating diverse training examples to help the model learn accurate predictions even with limited data. Training on varying sequence lengths enables efficient predictions from incomplete information. We also use the maximal sequence (23 snapshots) to ensure the model can utilize all available data, producing 23 training sequences per architecture (22 of varying lengths and 1 maximal).

Our training process yields a performance prediction model that is both generic – by training on diverse neural architectures, encouraging generalization – and robust, as it accurately predicts using anywhere from a single snapshot to the full sequence.

**Predictor hyperparameter selection.** The predictor's architectural and training hyperparameters – hidden width, loss function, learning-rate schedule, augmentation range, and pooling strategy – were selected via held-out Kendall $\tau$ search over a discrete grid. The configuration described above was the best-performing point in this search; wider hidden widths did not improve held-out $\tau$ and increased the train–test gap. See Appendix C.4 for the full search space and selected alternatives.

## 4  Experimental Setup

### 4.1  Evaluated Datasets

We replicate the setup of in White et al. (2021), and use three well-known datasets: CIFAR-10, CIFAR-100, and ImageNet16-120, each with 1,000 randomly sampled, disjoint architectures from NAS-Bench-201 (Dong & Yang, 2020), for a total of 3,000 distinct architectures. Following White et al. (2021), all sampled architectures were trained to convergence, and their meta-features (see previous section) were extracted to train our prediction model. Random subsets of these architectures were used for both training and testing, with varying train set sizes to control initialization times. Results were averaged over multiple runs to assess performance and stability. Detailed setup and methodology are provided in the following sections.

### 4.2 Baselines

We organize baselines around the two evaluation setups of the paper. In-domain comparison (hybrid family). LcSVR (Baker et al., 2017) and Omni White et al. (2021) are the strongest existing methods that, like NAP2, combine learned prediction with early-training signals on the source dataset; they define the right ceiling for in-domain hybrid prediction. NAP2 differs conceptually in what learning-curve signal it consumes: LcSVR uses hand-crafted early-stopping features and Omni extends model-based predictors with SoTL-E learning-curve extrapolation, while NAP2 uses the temporal evolution of layer-wise weight and gradient statistics — a denser characterization of the optimization trajectory than scalar accuracy/loss curves. Cross-dataset comparison (learning-curve and zero-cost families). Hybrid and model-based methods would need to be re-initialized on the target dataset, which is precisely the cost NAP2 is designed to avoid. The natural comparison set is therefore methods that operate without target-side initialization: learning-curve extrapolation (SoTL-E, SoTL, Early-Stop, LCE-m, LC-PFN) and zero-cost proxies (SynFlow, GradNorm, SNIP). NAP2 differs from learning-curve methods in that its predictor is amortized across architectures via training on the source dataset (rather than fitting a parametric extrapolation per candidate), and from zero-cost proxies in that it improves with more partial training (zero-cost proxies are static and impose a correlation ceiling that NAP2 exceeds at its very first snapshot).

In our first experimental setup, we *compare NAP2 to leading hybrid performance prediction methods*, which require fully trained architectures for model building and partial training on test architectures for feature extraction. As baselines, we use LcSVR (Baker et al., 2017) and Omni (White et al., 2021), two strong hybrid predictors that combine model-based learning with training-stage signals. LcSVR uses SVR on handcrafted and early stopping features, while Omni extends model-based predictors by integrating learning curve extrapolation (SoTL-E) into SemiNAS to balance predictive strength and efficiency. Omni is a strong, generalizable baseline, as shown in (White et al., 2021).

In our second experimental setup, we *evaluate NAP2 in a cross-dataset scenario*, training the prediction model on one dataset and applying it to new, unseen datasets and architectures. This setup allows the predictor to be reused without retraining. We compare NAP2 against baselines that similarly require no initialization on the target dataset. These include learning curve extrapolation methods, such as SoTL-E (Ru et al., 2021), which use partial training, and zero-cost proxies like SynFlow (Tanaka et al., 2020), GradNorm (Abdelfattah et al., 2021), and SNIP (Lee et al., 2018). The latter are training-free methods that estimate performance at initialization, providing a static baseline for our dynamic approach.

### 4.3 Evaluation Metrics

The authors of (White et al., 2021) use three metrics to report the performance of their evaluated predictors:

**Initialization Time.** This metric is applicable to predictors that pre-train a learning model (e.g., model-based methods). It measures the *total time* required to train the set of architectures based on which the learning model is created.

**Query Time.** Measures the average time the predictor needs to predict the performance of a *single architecture*.

**Kendall rank correlation.** Evaluates prediction accuracy by measuring the rank correlation (i.e., similarity in ordering) of two ranked lists. We use Kendall rank correlation to compare the rankings produced by each evaluated predictor to those reported in the NAS-Bench-201 dataset.

### 4.4 Training the NAP2 Prediction Model

NAP2 requires extraction of meta-data from neural architectures throughout their training. The training of all candidate architectures strictly followed the NAS-Bench-201 benchmark protocol to ensure consistency and comparability with established results: Nesterov momentum SGD optimizer, Cross-entropy loss function, an initial learning rate of 0.1 that decayed to 0 with a cosine annealing schedule, weight decay of 0.0005, and a batch size of 256.

For data augmentation of the candidate architectures, we applied random horizontal flips with a probability of 0.5 and random cropping. Specifically, we used a 32×32 patch with 4 pixels padding for CIFAR-10 and CIFAR-100 and a 16×16 patch with 2 pixels padding for ImageNet-16-120. Normalization was performed over the RGB channels. We extracted our meta-features at 100 mini-batch intervals.

---

**Cross-dataset transfer protocol.** When applying a predictor trained on a source dataset to a target dataset:

1. **Train (source-side, once).** Fit the full NAP2 predictor (autoencoders + BiGRU) on source-dataset architectures only. After this step the predictor is frozen; no component is re-trained or fine-tuned on any target dataset.

2. **Collect (target-side, per architecture).** For each target architecture, run a short partial training (up to 100 mini-batches by default, $\sim$9 s on a GTX 1080 Ti) and extract the per-snapshot weight and gradient statistics. *No target-dataset labels are used by the predictor at any point*; target labels are needed only for downstream evaluation, not for prediction itself.

3. **Predict (target-side, per architecture).** Pass the snapshots through the source-trained autoencoders and BiGRU. Feature maps are log-normalized in place (Section 3.3); no target-side normalization or rescaling is applied.

---

Complete dataset specifications, normalization parameters, augmentation pipelines, and data split details are provided in Appendix D.

### 4.5 Experimental Setup

Our evaluation followed the methodology outlined by White et al. (2021). Each run samples 200 architectures and averages results over ten runs. The same training and test sets were used by all evaluated algorithms.

In our first set of experiments, comparison to hybrid methods, we obtained different initialization times by training a varying number of architectures to train the learning models of all evaluated algorithms. The number of trained architectures ranged from 50 to 700. In our second set of experiments, where the learning model is trained on architectures from one dataset and applied to another, we use 700 architectures for training. All experiments ran on a single GTX 1080 Ti, matching (White et al., 2021) for timing comparability. Full details on randomness control, hardware specifications, software versions, and computational cost are provided in Appendix E.

## 5 Evaluation Results

### 5.1 Evaluation Setup #1: Comparison to Hybrid Performance Prediction Methods

The results of our evaluation are presented in Tables 1-3. We evaluated the baselines on various initialization and query times to provide a comprehensive analysis of their performance. NAP2 outperforms both baselines across all initialization and query time combinations on all three datasets. The improvement is most pronounced at short query times and lower initialization budgets, where NAP2 achieves substantially higher Kendall Tau with lower variance.

Our results are supported by the paired-t statistical test, which showed differences between NAP2 and the baselines are significant ($p < 0.01$) across all reviewed experiments. It is important to note that NAP2 uses a novel method to predict neural network performance, which opens the way for a possible future integration of NAP2 with other methods. We leave this research direction to future work.

### 5.2 Evaluation Setup #2: Cross-Dataset Generalization

A key challenge for performance predictors is generalizing to new, unseen datasets, which saves the large computational cost of retraining for each task (White et al., 2021). As explained in our description of the baselines, *NAP2 is a hybrid approach capable of cross-dataset performance prediction without requiring*

Table 1: Hybrid baselines performance on CIFAR-10.

| Init Time | Query Time | LcSVR | Omni_seminas | NAP2 |
|---|---|---|---|---|
| 1.8e5 | 54 | 0.4415 ± 0.1036 | 0.5246 ± 0.0911 | **0.6944 ± 0.0132** |
| 1.8e5 | 72 | 0.5302 ± 0.0920 | 0.5388 ± 0.1162 | **0.7038 ± 0.0142** |
| 1.8e5 | 90 | 0.5312 ± 0.0562 | 0.5321 ± 0.1481 | **0.7079 ± 0.0136** |
| 1.8e5 | 108 | 0.5895 ± 0.0722 | 0.5598 ± 0.1131 | **0.7119 ± 0.0140** |
| 1.8e5 | 198 | 0.7000 ± 0.0518 | 0.5119 ± 0.1113 | **0.7198 ± 0.0127** |
| 3.6e5 | 54 | 0.4572 ± 0.0651 | 0.5667 ± 0.1230 | **0.7469 ± 0.0072** |
| 3.6e5 | 72 | 0.4992 ± 0.0671 | 0.5673 ± 0.1295 | **0.7538 ± 0.0080** |
| 3.6e5 | 90 | 0.5669 ± 0.0729 | 0.5958 ± 0.1483 | **0.7571 ± 0.0089** |
| 3.6e5 | 108 | 0.6403 ± 0.0759 | 0.5840 ± 0.1125 | **0.7593 ± 0.0094** |
| 3.6e5 | 198 | 0.7309 ± 0.0601 | 0.6240 ± 0.1115 | **0.7675 ± 0.0088** |
| 1.1e6 | 54 | 0.5009 ± 0.0821 | 0.5798 ± 0.0995 | **0.7982 ± 0.0102** |
| 1.1e6 | 72 | 0.5779 ± 0.0559 | 0.5864 ± 0.1098 | **0.8040 ± 0.0118** |
| 1.1e6 | 90 | 0.6126 ± 0.0713 | 0.6243 ± 0.1124 | **0.8071 ± 0.0103** |
| 1.1e6 | 108 | 0.6422 ± 0.0542 | 0.6209 ± 0.1268 | **0.8085 ± 0.0108** |
| 1.1e6 | 198 | 0.7570 ± 0.0562 | 0.6862 ± 0.1130 | **0.8096 ± 0.0104** |
| 2.2e6 | 54 | 0.4837 ± 0.0619 | 0.6278 ± 0.0886 | **0.8068 ± 0.0160** |
| 2.2e6 | 72 | 0.5868 ± 0.0596 | 0.6305 ± 0.0810 | **0.8104 ± 0.0164** |
| 2.2e6 | 90 | 0.6395 ± 0.0565 | 0.6574 ± 0.1212 | **0.8137 ± 0.0165** |
| 2.2e6 | 108 | 0.6744 ± 0.0732 | 0.6705 ± 0.0927 | **0.8160 ± 0.0168** |
| 2.2e6 | 198 | 0.7900 ± 0.0536 | 0.7456 ± 0.1049 | **0.8195 ± 0.0176** |

Table 2: Hybrid baselines performance on CIFAR-100.

| Init Time | Query Time | LcSVR | Omni_seminas | NAP2 |
|---|---|---|---|---|
| 1.8e5 | 54 | 0.2943 ± 0.0733 | 0.5624 ± 0.1960 | **0.6112 ± 0.0090** |
| 1.8e5 | 72 | 0.4133 ± 0.1081 | 0.5673 ± 0.1860 | **0.6173 ± 0.0092** |
| 1.8e5 | 90 | 0.4032 ± 0.0790 | 0.5540 ± 0.1680 | **0.6226 ± 0.0088** |
| 1.8e5 | 108 | 0.4434 ± 0.0655 | 0.6110 ± 0.1481 | **0.6256 ± 0.0089** |
| 1.8e5 | 198 | 0.4992 ± 0.0847 | 0.6165 ± 0.1041 | **0.6362 ± 0.0106** |
| 3.6e5 | 54 | 0.3148 ± 0.0540 | 0.5778 ± 0.1648 | **0.6583 ± 0.0136** |
| 3.6e5 | 72 | 0.3986 ± 0.0618 | 0.5783 ± 0.1881 | **0.6690 ± 0.0127** |
| 3.6e5 | 90 | 0.4452 ± 0.0392 | 0.5982 ± 0.1874 | **0.6752 ± 0.0126** |
| 3.6e5 | 108 | 0.4754 ± 0.0386 | 0.6090 ± 0.1684 | **0.6786 ± 0.0129** |
| 3.6e5 | 198 | 0.5882 ± 0.0294 | 0.6587 ± 0.1238 | **0.6900 ± 0.0110** |
| 1.1e6 | 54 | 0.4113 ± 0.0507 | 0.6187 ± 0.1571 | **0.7193 ± 0.0092** |
| 1.1e6 | 72 | 0.4643 ± 0.0407 | 0.6324 ± 0.0824 | **0.7251 ± 0.0100** |
| 1.1e6 | 90 | 0.5003 ± 0.0363 | 0.6387 ± 0.1309 | **0.7285 ± 0.0093** |
| 1.1e6 | 108 | 0.5360 ± 0.0433 | 0.6575 ± 0.0861 | **0.7321 ± 0.0089** |
| 1.1e6 | 198 | 0.6253 ± 0.0361 | 0.7225 ± 0.0553 | **0.7351 ± 0.0099** |
| 2.2e6 | 54 | 0.4180 ± 0.0399 | 0.6456 ± 0.1031 | **0.7450 ± 0.0088** |
| 2.2e6 | 72 | 0.4581 ± 0.0622 | 0.6562 ± 0.1058 | **0.7514 ± 0.0090** |
| 2.2e6 | 90 | 0.5105 ± 0.0368 | 0.6855 ± 0.0708 | **0.7542 ± 0.0098** |
| 2.2e6 | 108 | 0.5513 ± 0.0441 | 0.6934 ± 0.0794 | **0.7555 ± 0.0106** |
| 2.2e6 | 198 | 0.6362 ± 0.0280 | 0.7567 ± 0.0868 | **0.7588 ± 0.0107** |

*fine-tuning on the target dataset.* Since NAP2 does not require initialization in cross-dataset setups, we compare NAP2 to top-performing learning curve extrapolation methods, which also have no initialization time. We structure our evaluation in two parts: first, we assess generalization across common image classification benchmarks. Second, to test robustness, we evaluate NAP2's performance under a significant dataset distribution shift.

The results of our evaluation are shown in Tables 4-6. From *all* the algorithms evaluated in (White et al., 2021), we report only those achieving top-3 performance for at least one query time. For NAP2, we present results with performance prediction models trained on one of the other two datasets. We also compare against LC-PFN (Adriaensen et al., 2023), a pre-trained learning curve predictor trained on over 1 million learning curves from diverse tasks. While LC-PFN benefits from extensive pre-training on external data, it provides a strong baseline for cross-dataset performance prediction, as it operates without fine-tuning on the target dataset. Note that LC-PFN requires at least 5 observations to produce reliable predictions, as its Bayesian prior-based approach needs sufficient data points to stabilize; consequently, LC-PFN shows negative or unreliable correlations in the earliest steps (1–4), which is reflected in our tables. In contrast,

Table 3: Hybrid baselines performance on ImageNet16-120.

| Init Time | Query Time | LcSVR | Omni_seminas | NAP2 |
|---|---|---|---|---|
| 1.8e5 | 54 | 0.2708 ± 0.0769 | 0.5370 ± 0.1755 | **0.5745 ± 0.0156** |
| 1.8e5 | 72 | 0.3689 ± 0.0615 | 0.5579 ± 0.1886 | **0.5859 ± 0.0146** |
| 1.8e5 | 90 | 0.4038 ± 0.0538 | 0.5793 ± 0.1476 | **0.5932 ± 0.0136** |
| 1.8e5 | 108 | 0.4147 ± 0.0693 | 0.5458 ± 0.1251 | **0.5985 ± 0.0125** |
| 1.8e5 | 198 | 0.5289 ± 0.0604 | 0.5919 ± 0.1139 | **0.6185 ± 0.0170** |
| 3.6e5 | 54 | 0.3204 ± 0.0510 | 0.5732 ± 0.1570 | **0.6525 ± 0.0166** |
| 3.6e5 | 72 | 0.4114 ± 0.0646 | 0.5797 ± 0.1934 | **0.6609 ± 0.0161** |
| 3.6e5 | 90 | 0.4326 ± 0.0428 | 0.5905 ± 0.2362 | **0.6672 ± 0.0162** |
| 3.6e5 | 108 | 0.4472 ± 0.0500 | 0.6072 ± 0.1868 | **0.6708 ± 0.0163** |
| 3.6e5 | 198 | 0.5887 ± 0.0420 | 0.6169 ± 0.1898 | **0.6825 ± 0.0166** |
| 1.1e6 | 54 | 0.3555 ± 0.0576 | 0.6595 ± 0.2000 | **0.7118 ± 0.0076** |
| 1.1e6 | 72 | 0.4120 ± 0.0538 | 0.6607 ± 0.1572 | **0.7177 ± 0.0084** |
| 1.1e6 | 90 | 0.4706 ± 0.0455 | 0.7018 ± 0.1703 | **0.7229 ± 0.0092** |
| 1.1e6 | 108 | 0.5060 ± 0.0539 | 0.7033 ± 0.1730 | **0.7256 ± 0.0102** |
| 1.1e6 | 198 | 0.6165 ± 0.0411 | 0.7273 ± 0.1167 | **0.7352 ± 0.0130** |
| 2.2e6 | 54 | 0.3424 ± 0.0458 | 0.6973 ± 0.1592 | **0.7474 ± 0.0129** |
| 2.2e6 | 72 | 0.4136 ± 0.0452 | 0.7124 ± 0.1285 | **0.7562 ± 0.0158** |
| 2.2e6 | 90 | 0.4293 ± 0.0572 | 0.7190 ± 0.1317 | **0.7607 ± 0.0173** |
| 2.2e6 | 108 | 0.4926 ± 0.0469 | 0.7217 ± 0.1758 | **0.7631 ± 0.0177** |
| 2.2e6 | 198 | 0.6105 ± 0.0353 | 0.7517 ± 0.1722 | **0.7689 ± 0.0183** |

Table 4: The performance of learning curve-based predictors on CIFAR-10. NAP2 is trained on the two other datasets. SynFlow, GradNorm, and SNIP are zero-cost proxies (static performance). **Bold** marks the best value per row; underline marks the second-best. *Takeaway: NAP2 leads at short query times ($Q \leq 27$ s) by $\Delta \tau \geq 0.20$ over the strongest non-NAP2 baseline; from $Q = 63$ s onward SoTL-E catches up and exceeds NAP2, but at 7–23× more partial training per architecture.*

| Alg./ Q. time | SynFlow | GradNorm | SNIP | SoTL-E | SoTL | Early Stop (ACC.) | LCE-m | LC-PFN | NAP2 (IN) | NAP2 (C-100) |
|---|---|---|---|---|---|---|---|---|---|---|
| 9 | 0.52 ± 0.04 | 0.46 ± 0.04 | 0.46 ± 0.04 | 0.48 ± 0.04 | 0.41 ± 0.04 | 0.43 ± 0.03 | 0.23 ± 0.04 | 0.31 ± 0.04 | **0.74 ± 0.02** | 0.71 ± 0.02 |
| 18 | 0.52 ± 0.04 | 0.46 ± 0.04 | 0.46 ± 0.04 | 0.63 ± 0.02 | 0.51 ± 0.03 | 0.52 ± 0.03 | 0.36 ± 0.04 | 0.37 ± 0.03 | **0.77 ± 0.02** | 0.73 ± 0.03 |
| 27 | 0.52 ± 0.04 | 0.46 ± 0.04 | 0.46 ± 0.04 | 0.7 ± 0.01 | 0.58 ± 0.02 | 0.56 ± 0.02 | 0.4 ± 0.03 | 0.48 ± 0.03 | **0.78 ± 0.02** | 0.75 ± 0.02 |
| 63 | 0.52 ± 0.04 | 0.46 ± 0.04 | 0.46 ± 0.04 | **0.79 ± 0.01** | 0.69 ± 0.02 | 0.59 ± 0.02 | 0.54 ± 0.03 | 0.61 ± 0.03 | 0.79 ± 0.02 | 0.78 ± 0.02 |
| 99 | 0.52 ± 0.04 | 0.46 ± 0.04 | 0.46 ± 0.04 | **0.81 ± 0.01** | 0.72 ± 0.02 | 0.63 ± 0.02 | 0.54 ± 0.04 | 0.66 ± 0.02 | 0.79 ± 0.02 | 0.78 ± 0.02 |
| 153 | 0.52 ± 0.04 | 0.46 ± 0.04 | 0.46 ± 0.04 | **0.83 ± 0.01** | 0.76 ± 0.01 | 0.64 ± 0.02 | 0.55 ± 0.04 | 0.69 ± 0.02 | 0.79 ± 0.02 | 0.78 ± 0.02 |
| 207 | 0.52 ± 0.04 | 0.46 ± 0.04 | 0.46 ± 0.04 | **0.84 ± 0.01** | 0.78 ± 0.01 | 0.69 ± 0.02 | 0.59 ± 0.03 | 0.71 ± 0.02 | 0.79 ± 0.02 | 0.78 ± 0.02 |

NAP2 demonstrates consistent performance from the very first observation, highlighting an advantage for early-stage prediction scenarios.

Our findings show that NAP2 consistently outperforms all baselines at short query times across all three datasets; SoTL-E catches up only at query times that require 7–23× more partial training per candidate. *The improvement is most pronounced at short query times*, where the new pipeline achieves Kendall Tau scores exceeding 0.6 from just 100 mini-batches (9 seconds) in the cross-dataset setting.

Notably, even with the shortest query time (nine seconds, or 100 mini-batches), NAP2 achieves a Kendall Tau score exceeding 0.6 across all datasets in the cross-dataset setting – substantially higher than all baselines. This strong early performance highlights the transferability and effectiveness of weight and gradient modeling in predicting future network performance.

We further compare NAP2 against zero-cost proxies (SynFlow, GradNorm, SNIP), which represent the limit of "training-free" prediction. As shown in Tables 4-6, static proxies achieve a fixed correlation ceiling (e.g., ∼0.52 on CIFAR-10, ∼0.47 on ImageNet) that cannot improve with more compute. NAP2 exceeds this ceiling at its very first step (100 mini-batches), achieving correlations of 0.71–0.74 (CIFAR-10), 0.64–0.66 (CIFAR-100), and 0.62–0.64 (ImageNet). Crucially, unlike static proxies, NAP2 leverages learning dynamics to rapidly improve, reaching correlations of >0.70 within 600 mini-batches on CIFAR-10 and CIFAR-100, and approaching that level on ImageNet. This demonstrates that while zero-cost priors provide a good starting point, capturing the dynamic trajectory of training offers significantly higher predictive fidelity for a small computational cost.

Table 5: The performance of learning curve-based predictors on CIFAR-100. NAP2 is trained on the two other datasets. SynFlow, GradNorm, and SNIP are zero-cost proxies (static performance). Same bold/underline convention as Table 4. *Takeaway: same short-query lead as on CIFAR-10 — NAP2(IN) reaches $\tau = 0.66/0.69/0.71$ at $Q = 9/18/27$ s, well above the next-best baseline ($\leq 0.56$). SoTL-E catches up from $Q = 63$ s onward with the same compute trade-off.*

| Alg./ Q. time | SynFlow | GradNorm | SNIP | SoTL-E | SoTL | Early Stop (ACC.) | LCE-m | LC-PFN | NAP2 (IN) | NAP2 (C-10) |
|---|---|---|---|---|---|---|---|---|---|---|
| 9 | 0.51 ± 0.04 | 0.46 ± 0.03 | 0.46 ± 0.03 | 0.27 ± 0.05 | 0.21 ± 0.04 | 0.22 ± 0.03 | 0.15 ± 0.04 | -0.13 ± 0.02 | **0.66 ± 0.01** | 0.64 ± 0.02 |
| 18 | 0.51 ± 0.04 | 0.46 ± 0.03 | 0.46 ± 0.03 | 0.43 ± 0.04 | 0.33 ± 0.04 | 0.39 ± 0.03 | 0.24 ± 0.04 | -0.10 ± 0.03 | **0.69 ± 0.01** | 0.66 ± 0.02 |
| 27 | 0.51 ± 0.04 | 0.46 ± 0.03 | 0.46 ± 0.03 | 0.56 ± 0.02 | 0.44 ± 0.03 | 0.47 ± 0.03 | 0.27 ± 0.04 | 0.08 ± 0.03 | **0.71 ± 0.02** | 0.67 ± 0.02 |
| 63 | 0.51 ± 0.04 | 0.46 ± 0.03 | 0.46 ± 0.03 | **0.74 ± 0.01** | 0.62 ± 0.02 | 0.57 ± 0.02 | 0.41 ± 0.03 | 0.33 ± 0.03 | 0.73 ± 0.02 | 0.69 ± 0.01 |
| 99 | 0.51 ± 0.04 | 0.46 ± 0.03 | 0.46 ± 0.03 | **0.79 ± 0.01** | 0.68 ± 0.02 | 0.58 ± 0.02 | 0.53 ± 0.02 | 0.43 ± 0.02 | 0.73 ± 0.03 | 0.70 ± 0.01 |
| 153 | 0.51 ± 0.04 | 0.46 ± 0.03 | 0.46 ± 0.03 | **0.82 ± 0.01** | 0.72 ± 0.02 | 0.59 ± 0.02 | 0.52 ± 0.04 | 0.54 ± 0.02 | 0.73 ± 0.03 | 0.71 ± 0.01 |
| 207 | 0.51 ± 0.04 | 0.46 ± 0.03 | 0.46 ± 0.03 | **0.84 ± 0.01** | 0.76 ± 0.01 | 0.61 ± 0.02 | 0.48 ± 0.04 | 0.61 ± 0.02 | 0.73 ± 0.03 | 0.72 ± 0.01 |

Table 6: The performance of learning curve-based predictors on ImageNet. NAP2 is trained on the two other datasets. SynFlow, GradNorm, and SNIP are zero-cost proxies (static performance). Same bold/underline convention as Table 4. *Takeaway: NAP2 leads at every query time from $9$ to $153$ s and ties SoTL-E at $207$ s ($\tau = 0.70$ each); on this harder target, training on CIFAR-100 transfers more reliably than training on CIFAR-10 throughout.*

| Alg./ Q. time | SynFlow | GradNorm | SNIP | SoTL-E | SoTL | Early Stop (ACC.) | LCE | LC-PFN | NAP2 (C-10) | NAP2 (C-100) |
|---|---|---|---|---|---|---|---|---|---|---|
| 9 | 0.47 ± 0.03 | 0.41 ± 0.04 | 0.42 ± 0.04 | 0.2 ± 0.06 | 0.2 ± 0.05 | 0.22 ± 0.04 | 0.1 ± 0.05 | -0.02 ± 0.05 | 0.62 ± 0.01 | **0.64 ± 0.01** |
| 18 | 0.47 ± 0.03 | 0.41 ± 0.04 | 0.42 ± 0.04 | 0.27 ± 0.05 | 0.24 ± 0.04 | 0.27 ± 0.04 | 0.16 ± 0.05 | -0.05 ± 0.04 | 0.63 ± 0.01 | **0.66 ± 0.01** |
| 27 | 0.47 ± 0.03 | 0.41 ± 0.04 | 0.42 ± 0.04 | 0.35 ± 0.04 | 0.3 ± 0.04 | 0.31 ± 0.03 | 0.17 ± 0.05 | -0.03 ± 0.05 | 0.64 ± 0.02 | **0.67 ± 0.01** |
| 63 | 0.47 ± 0.03 | 0.41 ± 0.04 | 0.42 ± 0.04 | 0.53 ± 0.03 | 0.42 ± 0.04 | 0.4 ± 0.03 | 0.26 ± 0.05 | 0.24 ± 0.04 | 0.64 ± 0.02 | **0.68 ± 0.01** |
| 99 | 0.47 ± 0.03 | 0.41 ± 0.04 | 0.42 ± 0.04 | 0.58 ± 0.02 | 0.47 ± 0.03 | 0.42 ± 0.03 | 0.35 ± 0.05 | 0.30 ± 0.04 | 0.65 ± 0.02 | **0.69 ± 0.01** |
| 153 | 0.47 ± 0.03 | 0.41 ± 0.04 | 0.42 ± 0.04 | 0.65 ± 0.02 | 0.53 ± 0.02 | 0.46 ± 0.02 | 0.43 ± 0.04 | 0.34 ± 0.04 | 0.67 ± 0.02 | **0.70 ± 0.01** |
| 207 | 0.47 ± 0.03 | 0.41 ± 0.04 | 0.42 ± 0.04 | 0.70 ± 0.02 | 0.58 ± 0.02 | 0.52 ± 0.02 | 0.48 ± 0.04 | 0.40 ± 0.04 | 0.69 ± 0.02 | **0.70 ± 0.01** |

Table 7: Accuracy score distribution comparison between CIFAR-10 and SVHN, highlighting the distribution shift.

| Performance Range | CIFAR-10 (%) | SVHN (%) |
|---|---|---|
| [0–0.85] | 40.0 | 5.3 |
| [0.85–0.92] | 41.5 | 6.7 |
| [0.92–0.95] | 18.5 | 13.7 |
| [0.95–1] | 0.0 | 74.3 |

Table 8: The probability of returning a top-performing architecture at the top of NAP2's ranked list of architectures. NAP2 was trained on CIFAR-10 and tested on CIFAR-100.

| mini-batch step ($\times 10^2$) | 1 | 2 | 3 | 4 | 5 | 10 | 15 | 20 |
|---|---|---|---|---|---|---|---|---|
| top-3 | 0.5 | 0.6 | 0.4 | 0.6 | 0.6 | 0.7 | 0.7 | 0.8 |
| top-5 | 0.9 | 1.0 | 1.0 | 1.0 | 1.0 | 1.0 | 1.0 | 0.9 |
| top-10 | 1.0 | 1.0 | 1.0 | 1.0 | 1.0 | 1.0 | 1.0 | 1.0 |
| top-20 | 1.0 | 1.0 | 1.0 | 1.0 | 1.0 | 1.0 | 1.0 | 1.0 |

Table 9: The probability of returning a top-performing architecture at the top of NAP2's ranked list of architectures. NAP2 was trained on CIFAR-10 and tested on ImageNet.

| mini-batch step ($\times 10^2$) | 1 | 2 | 3 | 4 | 5 | 10 | 15 | 20 |
|---|---|---|---|---|---|---|---|---|
| top-3 | 0.0 | 0.0 | 0.0 | 0.0 | 0.0 | 0.0 | 0.0 | 0.0 |
| top-5 | 0.0 | 0.0 | 0.0 | 0.0 | 0.0 | 0.0 | 0.0 | 0.0 |
| top-10 | 0.0 | 0.0 | 0.0 | 0.0 | 0.0 | 0.0 | 0.1 | 0.1 |
| top-20 | 0.0 | 0.1 | 0.1 | 0.1 | 0.2 | 0.4 | 0.4 | 0.5 |

Table 10: The probability of returning a top-performing architecture at the top of NAP2's ranked list of architectures. NAP2 was trained on CIFAR-100 and tested on ImageNet.

| mini-batch step ($\times 10^2$) | 1 | 2 | 3 | 4 | 5 | 10 | 15 | 20 |
|---|---|---|---|---|---|---|---|---|
| top-3 | 0.3 | 0.7 | 0.8 | 0.7 | 0.7 | 0.6 | 0.7 | 0.7 |
| top-5 | 0.6 | 0.8 | 0.9 | 0.9 | 0.8 | 1.0 | 1.0 | 1.0 |
| top-10 | 0.9 | 0.9 | 1.0 | 1.0 | 1.0 | 1.0 | 1.0 | 1.0 |
| top-20 | 0.9 | 1.0 | 1.0 | 1.0 | 1.0 | 1.0 | 1.0 | 1.0 |

Across the three cross-dataset target tables (Tables 4–6), NAP2 ranks first at every query time below 63 s on CIFAR-10 and CIFAR-100 (with $\Delta\tau$ from 0.08 to 0.22 over the strongest non-NAP2 baseline) and at every query time on ImageNet (with one tie at 207 s). Learning-curve baselines that close the gap at longer

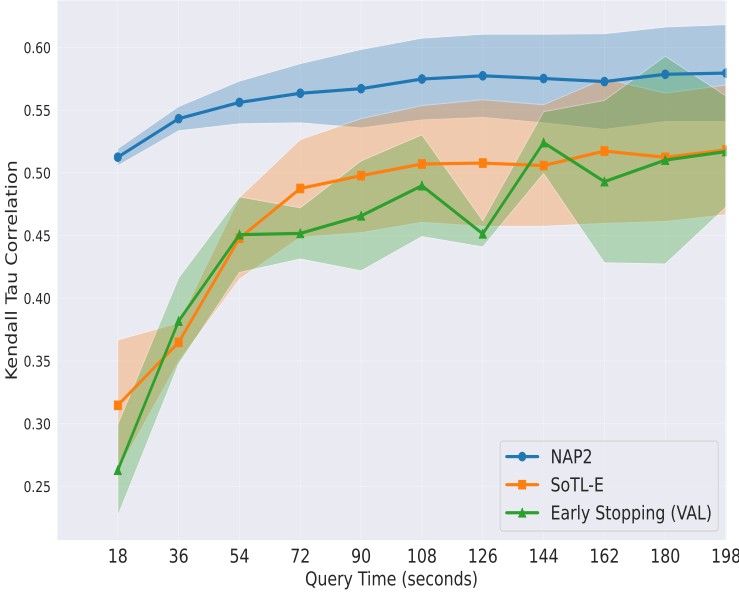

Figure 2: Kendall Tau on SVHN under cross-dataset transfer (NAP2 trained on CIFAR-10, no fine-tuning). Baselines are learning curve-based algorithms.

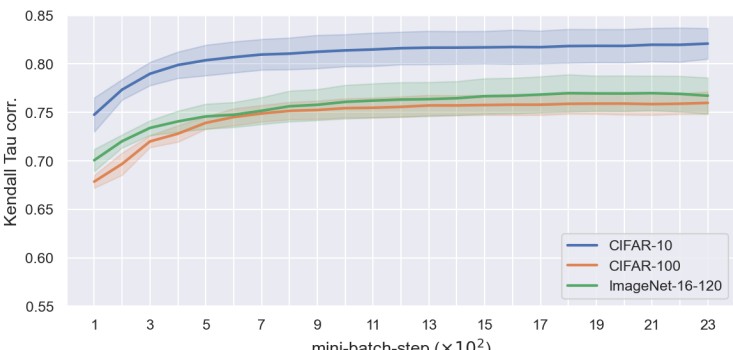

Figure 3: The Kendall Tau Rank Correlation achieved by NAP2 on the evaluated datasets, as a function of the number of mini-batches used to train each neural architecture.

query times — SoTL-E in particular — require partial training of each candidate architecture for 7–23× longer than NAP2's first observation, which defeats the purpose of fast prediction.

### 5.3 Robustness to Distribution Shift: A Case Study on SVHN

While CIFAR-10/100 and ImageNet differ in many ways, they also share notable similarities, potentially enhancing NAP2's effectiveness in cross-dataset scenarios. To further assess NAP2's generalizability, we evaluated it on the Street View House Numbers (SVHN) dataset, which is markedly different from the other datasets.

We randomly sampled 100 NAS-Bench-201 architectures and trained them on SVHN until convergence, ensuring none overlapped with previous experiments. We repeated this process three times. As shown in Table 7, the performance distribution on SVHN (compared to CIFAR-10) is distinct and challenging: when most architectures perform well, ranking becomes more difficult.

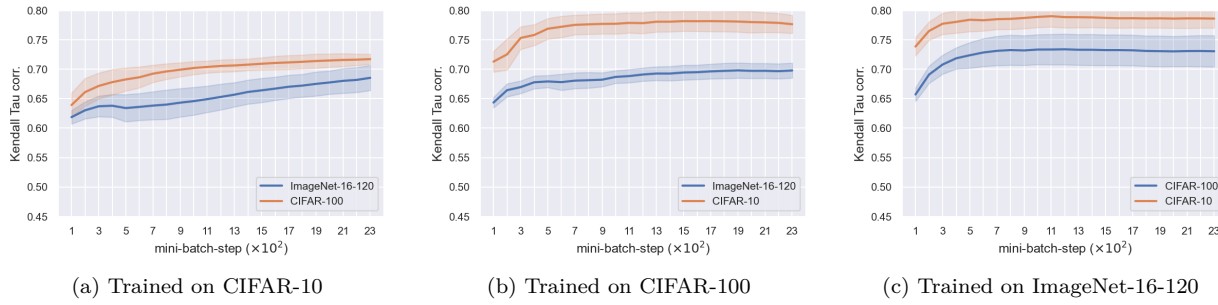

Figure 4: Kendall Tau achieved by NAP2 in cross-dataset transfer. Each panel shows a model trained on one dataset, evaluated on the other two, as a function of query mini-batches ($\times$100).

Figure 2 presents our cross-dataset results. We applied the model trained on CIFAR-10 (with no fine-tuning) and compared its performance to two baselines: SoTL-E (Ru et al., 2021), the top baseline from our experiments, and Early stop, a straightforward but effective benchmark. NAP2 outperforms both baselines from the first epoch onward, and the gap is preserved as the baselines accumulate more partial-training observations. These results strengthen our claims regarding NAP2's transferability.

## 6 Analysis & Discussion

### 6.1 Analyzing performance as a function of query time

Our evaluation shows that NAP2 achieves Kendall Tau correlation values exceeding 0.6 after only a single input (100 mini-batches) in cross-dataset settings, and around 0.7 in-domain. Performance improves monotonically with query time. NAP2 outperforms all hybrid baselines at every initialization and query time combination on the three in-domain tables (Sec. 5.1), and outperforms all learning-curve and zero-cost baselines at short query times in the cross-dataset tables (Sec. 5.2). Given this consistent dominance at short query times, a natural question is at what query budget NAP2's predictive power saturates and whether the saturation point depends on the source/target pair.

Figures 3 and 4 show performance in our two evaluation setups: single- and cross-dataset setup. Despite differences in absolute performance, two key conclusions emerge: (1) the largest improvement in NAP2's performance occurs within the first ten inputs, after which gains slow significantly; (2) predictive abilities plateau after about 20 inputs.

We hypothesize that as training progresses, network weight and gradient changes diminish, limiting NAP2's ability to provide further insights at later stages. We also explored alternative predictor architectures, including separate recurrent models for weights and gradients (fusing only at the final hidden state), but found no improvement over early fusion via concatenation (see Appendix C).

### 6.2 Identifying Top-Performing Architectures

While NAP2 performs well in cross-dataset setups, a key practical question is whether it can reliably identify top-performing architectures. The Kendall Tau metric from previous evaluations does not address this, as it weights all architectures equally. Additionally, prediction models trained on datasets with different average performance may struggle to select the best architectures in new datasets. To investigate this, we assess NAP2's ability to identify top performers *in a cross-dataset setup*. After every 100 mini-batches, we checked if the true best architecture appeared in the top 3/5/10/20 of NAP2's rankings. Across ten runs per dataset (with 200-architecture test sets), we calculated how often the best architecture was highly ranked.

Tables 8–10 report results for three cross-dataset directions. NAP2 surfaces top candidates quickly: CIFAR-10 $\rightarrow$ CIFAR-100 reaches Top-5 recall = 1.0 by step 2 (200 mini-batches) and remains at $\geq$ 0.9 thereafter, and CIFAR-100 $\rightarrow$ ImageNet reaches Top-10 = 1.0 from 300 mini-batches. Although comparable baselines

are not provided in White et al. (2021), these results show that NAP2 not only ranks but also identifies top-performing architectures early—a crucial trait for NAS scenarios requiring quick identification of promising models. Top-K recall (for $K \geq 5$) generally improves with query time and plateaus within 300–500 mini-batches; Top-3 trajectories are noisier because they involve resolving very small accuracy differences between the top architectures. The CIFAR-10 $\rightarrow$ ImageNet-16-120 direction is an extreme case where Top-3 stays at zero across all query times. A diagnostic analysis traces this to the predictor head fitting CIFAR-10's tightly clustered top-architecture accuracies (top-20 spread $\approx 0.013$) under L1 loss, which leaves insufficient training signal for fine head discrimination on transfer; we leave a ranking-loss replacement to future work. For a comprehensive analysis including in-domain results, Top-N Coverage metrics, and additional cross-dataset configurations, see Appendix F.

### 6.3 Robustness to optimizer choice: SGD to AdamW

NAP2 is trained on SGD dynamics, so we test whether it still ranks architectures well when they are trained with a different optimizer. We use AdamW, which changes gradient scales far more than SGD does.

We trained 200 NB-201 architectures on CIFAR-100 with AdamW, keeping the same architectures, dataset, and training budget as the main paper, and applied the existing SGD-trained predictor to these AdamW runs without any retraining. The optimizer change genuinely reorders the architectures: the SGD and AdamW ground-truth accuracy rankings agree at only Kendall $\tau = 0.731$ rather than 1, so transfer across it is a real test and not a formality.

Under AdamW, the weight statistics NAP2 reads barely move (within $0.6$–$1.4\times$ their SGD scale), and although the gradient statistics change sharply in magnitude (by $5$–$275\times$), they keep the same ordering of architectures. Since ranking is all the predictor needs, it retains most of its power on AdamW with no AdamW data: gradients alone fall only from $\tau = 0.786$ in-distribution to $0.665$, and weights alone from $0.769$ to $0.562$.

The one configuration that does not transfer directly is the two modalities combined ($\tau = -0.07$), because the combined predictor relies on the SGD-specific scale balance between weights and gradients that the gradient shift disturbs. This is easily restored: rescaling the gradient half back to its SGD range recovers the combined predictor to $\tau = 0.558$. Doing the same to the weights, which barely shift, is less effective ($0.502$), as expected, and pinpoints the gradient scale as the cause. No ranking information is lost, only its scaling, so the simplest cross-optimizer recipe is to deploy gradients alone, with gradient rescaling as a fallback when the combined predictor is preferred. Full per-modality results are in Appendix I.

### 6.4 Transfer to a larger search space: NAS-Bench-101

We evaluated the NB-201 (CIFAR-100) predictor on NAS-Bench-101, whose largest architectures are roughly $31\times$ larger than NB-201's. Without any NB-101 training, NAP2 reached $\tau = 0.510$ (Spearman $\rho = 0.705$) on the standard NB-101 sample, exceeding the strongest published zero-cost proxy on this benchmark (SPW, $\rho = 0.569$; Jing et al., 2025); transfer also runs in the inverse direction (NB-101 $\rightarrow$ NB-201, mean $\tau = 0.564$). On a size-controlled NB-101 sub-sample where the parameter-count shortcut is anti-correlated by construction, the same predictor still reaches $\tau = 0.343$, ruling out the "learned size proxy" reading; an encoder matched to the controlled regime raises this to $\tau = 0.557$ (Appendix J).

## 7 Conclusions and Future Work

We presented NAP2, a neural network performance prediction method that analyzes the temporal evolution of weight and gradient statistics during the early stages of training to predict final performance. On NAS-Bench-201, NAP2 outperforms strong hybrid baselines across all evaluated initialization and query time configurations in the in-domain setting, and outperforms learning-curve and zero-cost baselines at short query times in cross-dataset settings. A distinguishing property of NAP2 is its support for cross-dataset transfer: prediction models trained on one dataset can be applied to unseen datasets without re-initialization or fine-tuning, a capability that distinguishes it from many existing hybrid or model-based approaches

that require target-domain training. This transferability extends even to significant distribution shift, as demonstrated by the SVHN case study, where NAP2 trained on CIFAR-10 outperforms all baselines from the first epoch onward, and to a change of training optimizer, where an SGD-trained predictor transfers zero-shot to AdamW-trained architectures and recovers $\sim$91% of the cross-optimizer ranking correlation (§6.3).

Several directions for future work emerge from this study. The most immediate is the integration of NAP2 into a full NAS framework. While the present work focuses on performance prediction in isolation, deploying NAP2 as the evaluation engine within a NAS loop would require balancing exploration of the architecture search space with exploitation of NAP2's predictions. We plan to investigate the use of deep reinforcement learning to guide this process, using NAP2's early predictions as reward signals to adaptively allocate training budget across candidate architectures.

Beyond NAS, NAP2's layer-wise weight and gradient representations may find applications in related domains. In neural network robustness and verification, methods such as ReluPlex rely on analyzing individual neurons to verify network properties; NAP2's per-layer statistics could serve as a guide for identifying which layers or neurons are most informative to target, potentially reducing the computational cost of verification. More broadly, the ability to characterize a network's learning trajectory from short training runs opens possibilities for tasks such as predicting training stability, detecting pathological optimization behavior early, or guiding hyperparameter selection across datasets.

**Scope and limitations.** NAP2 has been evaluated on two cell-based image-classification search spaces – NAS-Bench-201 throughout the main paper and NAS-Bench-101 in App. J, the latter providing an approximately 31$\times$ larger maximum architecture ($\sim$47M vs $\sim$1.5M parameters) – and across two optimizer families (SGD throughout the main paper, AdamW in App. I, where gradient-only zero-shot transfer recovers $\sim$91% of the SGD$\leftrightarrow$AdamW ranking correlation). The following are explicitly outside the scope of the current evidence and are noted as future work: (i) transformer or language-modeling architectures – transformer-scale evaluation goes beyond what current NAS-predictor papers attempt, and existing NAS benchmarks were designed precisely to enable fair comparison at tractable scale; (ii) optimizer families beyond SGD and AdamW (such as Muon or LAMB), which may exhibit different gradient-distribution dynamics, as well as finer-grained hyperparameter sweeps within a fixed optimizer; (iii) search spaces materially different from cell-based image-classification benchmarks. The 100-unit uniform sampling is a practical choice for current-scale architectures; for substantially larger layers, the sampling scheme is a modular component that admits adaptation (importance-weighted sampling, structured subsampling, or larger sample size), with the autoencoder already supporting variable-resolution input.

**Broader impact.** The dominant practical benefit of NAP2 is computational: by replacing full architecture training with a learned predictor that operates from the first $\sim$100 mini-batches of training, it reduces per-architecture evaluation cost by orders of magnitude relative to full-training NAS pipelines. This lowers both the energy footprint of large-scale architecture search and the entry barrier for groups with limited compute, contributing to broader access to NAS-based research. We do not foresee direct ethical risks specific to NAP2 as a prediction tool; downstream considerations depend on the application of the architectures discovered and on the choice of search objective.

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

## Appendix

## A Pipeline Overview

This appendix details the end-to-end pipeline of NAP2, visualizing the data transformations and tensor dimensions from raw architecture to performance prediction. The process, illustrated in Figure 5, consists of four stages.

- **Stage 1: Architecture Training & Snapshotting**. The neural architecture is trained for a limited number of steps ($T'$ snapshots). At each snapshot $t$, we extract the model's weights ($\mathbf{W}_t$) and gradients ($\mathbf{G}_t$).

- **Stage 2: Feature Map Construction**. For each snapshot, we compute 12 statistical meta-features (e.g., mean, variance, quantiles) for sampled units across the network's layers. As shown in Figure 5, we sample $S = 99$ units per layer (plus one for the layer index, totaling 100 values). This results in a structured **Feature Map (FM)** for weights and gradients, with dimensions:

$$\mathrm{FM}_t \in \mathbb{R}^{12 \times 65 \times 100} \tag{3}$$

representing (12 features, 65 layers, 100 values).

- **Stage 3: Feature Compression (CAE Encoders)**. Feature maps are first log-normalized ($x \mapsto \mathrm{sign}(x) \cdot \log(1 + |x|)$) to compress the dynamic range, then passed through pre-trained Convolutional Autoencoder (CAE) encoders. We use separate encoders for weights ($\mathrm{CAE}_W$) and gradients ($\mathrm{CAE}_G$). Each encoder maps the high-dimensional feature map to a compact embedding:

$$\mathbf{e}_{W_t}, \mathbf{e}_{G_t} \in \mathbb{R}^{128} \tag{4}$$

These embeddings are concatenated to form a fused representation $\mathbf{z}_t \in \mathbb{R}^{256}$ for step $t$.

- **Stage 4: Temporal Modeling & Prediction**. The sequence of fused embeddings $\mathbf{Z} = (\mathbf{z}_1, \ldots, \mathbf{z}_{T'})$ is fed into a bidirectional GRU predictor with a hidden size of 128. The model uses dual-path pooling (concatenating the last hidden state with an attention-weighted context vector) followed by a dense head (Linear $\rightarrow$ ReLU $\rightarrow$ Linear $\rightarrow$ Sigmoid) to output the predicted performance score $\hat{y} \in [0, 1]$.

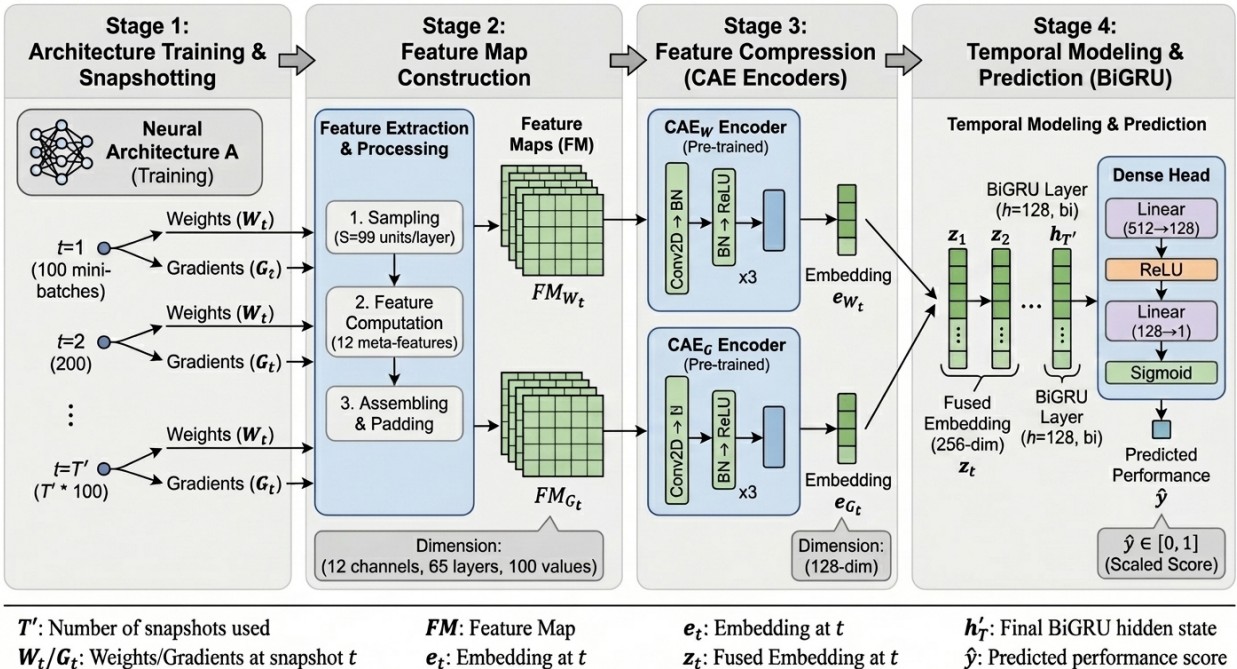

Figure 5: High-level overview of the NAP2 pipeline, detailing the four stages: (1) Partial training and snapshotting, (2) Construction of feature maps from weights and gradients, (3) Compression of feature maps using pre-trained CAE encoders into 128-dimensional embeddings, and (4) Temporal modeling using a bidirectional GRU to predict the final performance score.

## B  Feature Definitions & Notation

At each step of our proposed approach, we extract features from each layer in the analyzed network during its early training stages. These features capture the evolution of weights and gradients, which serve as proxies for the network's learning dynamics and ultimately its final performance. We now expand on the various features and the way they are generated.

Our features consist of two groups, each containing 12 meta-features. One group is computed from the network's weights at each training snapshot, and the other from the gradients. These features are computed independently for weights and gradients, allowing us to capture both the evolving parameter state and the optimization dynamics simultaneously. We describe each feature in detail below, explaining what it measures, how it is computed, and why it is informative for performance prediction.

### B.1  Overview of the Feature Extraction Process

During training, we log weights and gradients every 100 mini-batches, capturing $T = 23$ snapshots of the network's state. For each snapshot $t$ and each layer $\ell$, we extract 12 meta-features from the weights $\mathbf{W}_{\ell,t}$ and 12 from the gradients $\mathbf{G}_{\ell,t}$.

To handle the varying sizes of layers across different architectures, we employ a sampling strategy: for layers with more than $S = 99$ units (neurons or filters), we deterministically sample exactly $S$ units using a fixed seed. This ensures reproducibility while maintaining computational tractability. We then compute the 12 features over these sampled values, combining both a global statistic computed over all sampled units and per-unit statistics that capture layer heterogeneity.

The resulting feature vectors are assembled into fixed-shape feature maps. Each snapshot produces a feature map with shape $(L, V, F) = (65, 100, 12)$, where $L$ is the maximum number of layers, $V$ is the values dimension storing multiple values per feature, and $F$ is the number of feature types. We capture $T = 23$

such snapshots over time. This uniform representation enables us to train a single predictor model across diverse architectures while preserving the temporal evolution of layer-wise statistics.

## B.2 Motivation and Rationale

**Why these features?** The selected meta-features jointly capture *scale* (min/max/mean), *dispersion* (variance/std), *shape* (skewness/kurtosis), *energy* (L1/L2 norms), and *robust order statistics* (quantiles). This balances sensitivity to training dynamics with robustness to outliers and initialization noise. Gradients reflect optimization signals; weights reflect the evolving representation. Their combination improves early predictive power and transferability across datasets.

**Why fixed-shape maps?** NAS architectures vary in depth and layer width. To train one predictor, we require a uniform input shape. We therefore map per-layer statistics into fixed-size feature maps with consistent axes and padding/truncation policies. This enables a single encoder to learn invariant patterns of training dynamics across heterogeneous architectures.

## B.3 Notation

- $\mathbf{W}_{\ell,t} \in \mathbb{R}^{d_\ell}$: Vectorized weight parameters of layer $\ell$ at snapshot $t$ (after sampling)
- $\mathbf{G}_{\ell,t} \in \mathbb{R}^{d_\ell}$: Vectorized gradient parameters of layer $\ell$ at snapshot $t$ (after sampling)
- $d_\ell$: Effective dimensionality of layer $\ell$ after sampling $S$ units/filters
- $L = 65$: Maximum number of layers (architectures with $< 65$ layers are zero-padded)
- $T = 23$: Total number of training snapshots (one per $\Delta t = 100$ mini-batches)
- $S = 99$: Number of units/filters sampled per layer for computational efficiency
- $F = 12$: Number of meta-features extracted per layer per time step

## B.4 Sampling Strategy

Neural architectures vary significantly in layer widths. Some convolutional layers contain hundreds of filters, while dense layers may have thousands of neurons. Computing features over all units would be computationally expensive and create variable-sized feature maps that cannot be processed uniformly.

We therefore sample a fixed number of units per layer. If a layer contains more than $S = 99$ units, we uniformly sample exactly $S$ units using deterministic indexing based on a fixed seed. This ensures that the same layer produces identical samples across different runs, enabling reproducibility. For convolutional layers, we sample $S$ filters and compute features over their flattened kernel weights. For dense layers, we sample $S$ neurons and compute features over their weight vectors. Layers with $\leq S$ units use all available values.

$$\text{If } |\mathbf{W}_\ell| > S : \quad \mathbf{W}_{\ell,t} \leftarrow \text{sample}(\mathbf{W}_{\ell,t}, S, \text{seed} = 42) \tag{5}$$

We choose $S = 99$ because it provides a good balance between computational efficiency and representativeness. Combining one global statistic (computed over all sampled units) with 99 per-unit values gives us 100 values per feature per layer, which matches our feature map design where the values dimension is set to approximately 100. This sampling strategy allows us to capture both the overall layer behavior and its internal heterogeneity, providing enough detail to characterize the layer's state while remaining computationally manageable.

## B.5 The Twelve Meta-Features

For notational clarity, we use $\mathbf{x} \in \mathbb{R}^d$ to denote either $\mathbf{W}_{\ell,t}$ or $\mathbf{G}_{\ell,t}$ in the following definitions. Each feature is computed independently for weights and gradients, yielding 12 weight-based features and 12 gradient-based features per layer per snapshot.

We organize our 12 features into several categories: central tendency features that capture the typical value, dispersion features that measure spread and range, distribution shape features that characterize asymmetry and tail behavior, norm features that quantify magnitude, and quantile features that provide robust order statistics.

### B.5.1 Central Tendency Features

**Mean** ($f_1$): The arithmetic mean captures the average value of the weights or gradients in the layer. This feature reflects the typical scale of parameters and their updates.

$$f_1(\mathbf{x}) = \frac{1}{d} \sum_{j=1}^{d} x_j \tag{6}$$

**Median** ($f_2$): The median provides a robust measure of central tendency that is less sensitive to outliers than the mean. This feature helps characterize the typical parameter value even when the distribution is skewed or contains extreme values.

$$f_2(\mathbf{x}) = \text{median}(\{x_1, \ldots, x_d\}) \tag{7}$$

### B.5.2 Dispersion Features

**Variance** ($f_3$): Variance measures the spread of values around the mean. Higher variance indicates more diverse parameter values or gradient magnitudes, which may signal different learning dynamics.

$$f_3(\mathbf{x}) = \frac{1}{d} \sum_{j=1}^{d} (x_j - \bar{x})^2, \quad \bar{x} = f_1(\mathbf{x}) \tag{8}$$

**Standard Deviation** ($f_4$): The standard deviation is the square root of variance, providing a measure of spread in the same units as the original values. It complements variance and is used in the normalization of higher-order moments.

$$f_4(\mathbf{x}) = \sqrt{f_3(\mathbf{x})} \tag{9}$$

**Minimum** ($f_5$): The minimum value indicates the lower bound of the distribution. For weights, this captures the smallest parameter magnitude; for gradients, it reflects the smallest update magnitude in the layer.

$$f_5(\mathbf{x}) = \min_{j \in \{1, \ldots, d\}} x_j \tag{10}$$

**Maximum** ($f_6$): The maximum value indicates the upper bound of the distribution. This feature, together with the minimum, provides the range of values and helps identify extreme parameter values or gradient spikes.

$$f_6(\mathbf{x}) = \max_{j \in \{1, \ldots, d\}} x_j \tag{11}$$

### B.5.3 Distribution Shape Features

**Skewness** ($f_7$): Skewness measures the asymmetry of the distribution. Positive skewness indicates a long tail on the right, while negative skewness indicates a long tail on the left. The distribution of weights and gradients often exhibits asymmetry, which relates to the curvature and geometry of the loss landscape.

$$f_7(\mathbf{x}) = \frac{1}{d} \sum_{j=1}^{d} \left( \frac{x_j - \bar{x}}{\sigma} \right)^3, \quad \sigma = f_4(\mathbf{x}) \tag{12}$$

**Kurtosis ($f_8$):** Kurtosis measures the heaviness of the distribution's tails compared to a normal distribution. Higher kurtosis indicates heavier tails and more extreme values. We compute excess kurtosis (subtracting 3) following the SciPy convention, so a normal distribution has kurtosis of 0.

$$f_8(\mathbf{x}) = \frac{1}{d} \sum_{j=1}^{d} \left( \frac{x_j - \bar{x}}{\sigma} \right)^4 - 3 \tag{13}$$

### B.5.4   Covariance Feature

**Covariance ($f_9$):** We compute covariance as the unbiased sample variance using Bessel's correction. This is effectively the covariance of the parameter values with themselves, providing a measure of variability that accounts for the sample size. Mathematically, this is $\mathrm{Cov}(\mathbf{x}, \mathbf{x})$ using $(d-1)$ in the denominator rather than $d$.

$$f_9(\mathbf{x}) = \frac{1}{d-1} \sum_{j=1}^{d} (x_j - \bar{x})^2 = \frac{d}{d-1} \cdot f_3(\mathbf{x}) \tag{14}$$

Note that this differs from temporal covariance (covariance across time steps) or cross-layer covariance (covariance between different layers), which are not computed here.

### B.5.5   Norm Features

**L1 Norm ($f_{10}$):** The L1 norm (Manhattan distance) sums the absolute values of all parameters. This measures the total magnitude of weights or gradients without squaring, making it less sensitive to large outliers than the L2 norm.

$$f_{10}(\mathbf{x}) = \sum_{j=1}^{d} |x_j| \tag{15}$$

**L2 Norm ($f_{11}$):** The L2 norm (Euclidean distance) measures the magnitude of the weight or gradient vector. This is a fundamental quantity in optimization, as it appears in weight decay regularization and gradient clipping strategies. Larger L2 norms may indicate sharper loss landscapes or more aggressive updates.

$$f_{11}(\mathbf{x}) = \sqrt{\sum_{j=1}^{d} x_j^2} = \|\mathbf{x}\|_2 \tag{16}$$

### B.5.6   Quantile Feature (Multi-valued)

**Quantiles ($f_{12}$):** We compute five quantiles to capture the distribution shape: the minimum (0th percentile), first quartile (25th percentile), median (50th percentile), third quartile (75th percentile), and maximum (100th percentile). Together, these form the standard five-number summary, with the 25th and 75th percentiles capturing the interquartile range of the distribution.

$$f_{12}(\mathbf{x}) = \begin{bmatrix} q_0(\mathbf{x}) \\ q_{25}(\mathbf{x}) \\ q_{50}(\mathbf{x}) \\ q_{75}(\mathbf{x}) \\ q_{100}(\mathbf{x}) \end{bmatrix} = \begin{bmatrix} f_5(\mathbf{x}) \\ Q1 \\ f_2(\mathbf{x}) \\ Q3 \\ f_6(\mathbf{x}) \end{bmatrix} \tag{17}$$

Although this feature returns 5 values, we conceptually count it as a single feature in our $F = 12$ count, as the quantiles together characterize a single aspect of the distribution (its order statistics).

### B.6 Feature Map Construction

After extracting the 12 features for each layer at each snapshot, we organize them into feature maps with a fixed shape that can be processed uniformly by our encoder models. This section explains how we construct these feature maps and the rationale behind our design choices.

At each snapshot $t$, for each layer $\ell$, we compute feature vectors where each feature $f_i$ may contain multiple values (stored in dimension $V$). Conceptually, we extract 12 feature types per layer per snapshot. These features are then assembled into feature maps:

Each snapshot produces a feature map with shape $(L, V, F)$, where for each of the $L$ layers and $F = 12$ features, we store $V = 100$ values. Over $T = 23$ snapshots, we obtain sequences of feature maps that are processed temporally by the recurrent predictor.

**Why fixed-shape maps?** NAS architectures vary significantly in depth (number of layers) and width (number of units per layer). To train a single predictor model that works across all architectures, we need a uniform input representation. By mapping all architectures to fixed-size feature maps with shape $(L, V, F) = (65, 100, 12)$ per snapshot (captured over $T = 23$ snapshots), we enable our encoder to learn patterns that generalize across diverse architectures while preserving the temporal evolution captured in the sequence of snapshots.

**Depth dimension ($L = 65$):** We set the maximum number of layers to 65, chosen as a balance between capturing sufficient depth and maintaining computational efficiency. This value proved effective in our experiments. Architectures with fewer layers are zero-padded at the end. Architectures with more than 65 layers are truncated, keeping only the first 65 layers.

**Temporal dimension ($T = 23$):** We capture $T = 23$ snapshots during training, one every 100 mini-batches. This spans approximately the first 2,300 mini-batches of training, which corresponds to roughly the first 16 epochs for CIFAR-10/100 (with batch size 256). We found empirically that predictive power plateaus after about 20 snapshots, so $T = 23$ provides sufficient temporal coverage while remaining computationally efficient.

**Feature dimension ($F = 12$):** We extract exactly 12 meta-features per layer per snapshot, as described above. Each feature represents one aspect of the distribution (e.g., mean, variance, skewness). Multiple values per feature (e.g., the five percentiles from the quantile feature, or values from both global and per-unit computations) are stored in the values dimension $V$, maintaining $F = 12$ as the number of distinct feature types.

**Values dimension ($V \approx 100$):** For each feature, we store approximately 100 values: one global statistic computed over all sampled units, plus $S = 99$ per-unit values. This provides a compact yet expressive representation that captures both layer-level summary statistics and unit-level heterogeneity. The values dimension is set to exactly 100 in our implementation, with padding applied when layers have fewer than 99 units.

**Padding policy:** Missing layers (for architectures with $< 65$ layers) are zero-padded at the end. Missing unit values (for layers with $< 99$ units) are also zero-padded to reach the fixed values dimension $V = 100$. Padding is applied after feature extraction but before feature map assembly.

**Normalization**: Before encoding, each assembled feature map is log-normalized via $x \mapsto \text{sign}(x) \cdot \log(1+|x|)$, which compresses the dynamic range of raw statistics without requiring per-dataset parameters. This is critical for cross-dataset transfer, as it places feature maps from different datasets into a comparable range. Within the CAE encoder, each convolutional layer is followed by a BatchNorm2d layer (see Appendix C.1), which further normalizes activations using batch statistics during training.

### B.7 Implementation Details

**NaN Handling:** During training, some layers may contain dead neurons or experience gradient vanishing, leading to NaN values in weights or gradients. All our statistical computations use NaN-aware implementations (e.g., `np.nanmean`, `np.nanstd`) that skip NaN values, ensuring robust feature extraction even when some units are inactive.

**Dense vs. Convolutional Layers:**

- **Dense layers**: We sample $S = 99$ neurons uniformly at random (with deterministic indexing). For each sampled neuron, we extract its incoming weight vector. Features are computed over all sampled weight values concatenated together.

- **Convolutional layers**: We sample $S = 99$ filters uniformly at random (with deterministic indexing). For each sampled filter, we flatten its kernel weights into a vector. For example, a $3 \times 3$ filter with $C$ input channels is flattened to $9C$ values. Features are computed over all sampled filter weights concatenated together.

**Global and per-unit computation:** As described in the paper, we compute features in two ways: first, a single value over the entire layer's data, and second, values for each individual unit (neuron or filter). Both types of values are collected and stored in the values dimension $V$ for each feature, providing both layer-level summary statistics and unit-level details while maintaining $F = 12$ feature types.

**Reproducibility:** The sampling process uses fixed seeds and deterministic indexing, ensuring that the same architecture produces identical feature maps across runs. Architecture training also uses fixed seeds and determinism flags (see Appendix E), ensuring that weights and gradients are logged identically. With identical inputs, feature extraction is bitwise-stable across runs on the same platform.

## C    Model Specifications

This appendix provides complete architectural specifications for the convolutional auto-encoders (CAE) and the BiGRU predictor, including layer configurations, hyperparameters, fusion mechanisms, and loss functions.

### C.1    Convolutional Auto-Encoder (CAE)

We train two separate CAEs: one for weights feature maps ($\mathrm{CAE}_W$) and one for gradients feature maps ($\mathrm{CAE}_G$). Both share identical architecture but are trained independently on their respective data.

#### C.1.1    Input Specification

- **Input shape**: $(F, L, V) = (12, 65, 100)$
- $F = 12$: Number of feature channels (meta-features)
- $L = 65$: Number of layers (padded/truncated)
- $V = 100$: Feature map values dimension
- **Data type**: Double precision (`float64`)

#### C.1.2    Encoder Architecture

The encoder consists of three convolutional blocks, each comprising Conv2D, BatchNorm2D, and ReLU activation.

Table 11: CAE Encoder Layer Specifications

| Layer | Type | In Ch. | Out Ch. | Kernel | Stride | Padding |
|---|---|---|---|---|---|---|
| Input | – | – | 12 | – | – | – |
| Conv2D-1 | Conv2D | 12 | 512 | $(1, 100)$ | $(1, 1)$ | $(0, 0)$ |
| BatchNorm-1 | BatchNorm2D | 512 | 512 | – | – | – |
| Activation-1 | ReLU | 512 | 512 | – | – | – |
| Conv2D-2 | Conv2D | 512 | 256 | $(65, 1)$ | $(1, 1)$ | $(0, 0)$ |
| BatchNorm-2 | BatchNorm2D | 256 | 256 | – | – | – |
| Activation-2 | ReLU | 256 | 256 | – | – | – |
| Conv2D-3 | Conv2D | 256 | 128 | $(1, 1)$ | $(1, 1)$ | $(0, 0)$ |
| BatchNorm-3 | BatchNorm2D | 128 | 128 | – | – | – |
| Activation-3 | ReLU | 128 | 128 | – | – | – |
| Output | – | – | 128 | $(1, 1, 128)$ | – | – |

**Output shape after encoder**: $(128, 1, 1) \rightarrow$ flattened to $\mathbb{R}^{128}$ (embedding dimension).

#### C.1.3    Decoder Architecture

The decoder mirrors the encoder structure using transposed convolutions to reconstruct the input.

#### C.1.4    CAE Training Configuration

- **Loss function**: Mean Squared Error (MSE) reconstruction loss with sum reduction

$$\mathcal{L}_{\mathrm{CAE}} = \sum_{f,\ell,v} \left( \mathbf{X}_{f,\ell,v} - \hat{\mathbf{X}}_{f,\ell,v} \right)^2 \tag{18}$$

Table 12: CAE Decoder Layer Specifications

| Layer | Type | In Ch. | Out Ch. | Kernel | Stride | Padding |
|---|---|---|---|---|---|---|
| Input | – | – | 128 | $(1, 1, 128)$ | – | – |
| ConvT2D-1 | ConvTranspose2D | 128 | 256 | $(1, 1)$ | $(1, 1)$ | $(0, 0)$ |
| BatchNorm-1 | BatchNorm2D | 256 | 256 | – | – | – |
| Activation-1 | ReLU | 256 | 256 | – | – | – |
| ConvT2D-2 | ConvTranspose2D | 256 | 512 | $(65, 1)$ | $(1, 1)$ | $(0, 0)$ |
| BatchNorm-2 | BatchNorm2D | 512 | 512 | – | – | – |
| Activation-2 | ReLU | 512 | 512 | – | – | – |
| ConvT2D-3 | ConvTranspose2D | 512 | 12 | $(1, 100)$ | $(1, 1)$ | $(0, 0)$ |
| Output | – | – | 12 | $(12, 65, 100)$ | – | – |

- **Optimizer**: Adam with learning rate 0.001

- **Learning rate schedule**: StepLR with step size 20 and gamma 0.15, preceded by a 5-epoch warmup from $10^{-6}$ to 0.001

- **Batch size**: 32

- **Epochs**: 150

- **Early stopping**: Patience of 25 epochs on validation loss (or training loss if no validation set)

- **Weight initialization**: PyTorch default (Kaiming for Conv2D)

### C.2 BiGRU Predictor

The BiGRU predictor takes a sequence of fused embeddings and outputs a scalar performance score using dual-path pooling (last hidden state + attention-weighted context).

### C.2.1 Input Specification

- **Input shape**: $(T', D) = (T', 256)$

- $T' \in \{1, 2, \ldots, 22\}$: Sequence length during training (variable for augmentation). During inference, the full sequence up to $T = 23$ snapshots can be used.

- $D = 256$: Fused embedding dimension (128 from $\text{CAE}_W$ + 128 from $\text{CAE}_G$)

- **Batch-first format**: Tensor shape is $(B, T', D)$ where $B$ is batch size

### C.2.2 Architecture Specification

### C.2.3 BiGRU Forward Pass

The forward computation uses dual-path pooling:

$$\mathbf{O}, \mathbf{h} = \text{BiGRU}(\mathbf{z}_1, \ldots, \mathbf{z}_{T'}), \quad \mathbf{O} \in \mathbb{R}^{T' \times 256} \tag{19}$$

$$\mathbf{h}_{\text{last}} = [\mathbf{h}_{[-2]}; \mathbf{h}_{[-1]}] \in \mathbb{R}^{256} \quad \text{(last fwd + last bwd hidden)} \tag{20}$$

$$\alpha_t = \text{softmax}_t(\mathbf{w}_a^\top \mathbf{O}_t), \quad \text{masked for zero-padded steps} \tag{21}$$

$$\mathbf{c} = \sum_t \alpha_t \mathbf{O}_t \in \mathbb{R}^{256} \quad \text{(attention context)} \tag{22}$$

$$\mathbf{o}_1 = \text{ReLU}(\mathbf{W}_1[\mathbf{h}_{\text{last}}; \mathbf{c}] + \mathbf{b}_1), \quad \mathbf{W}_1 \in \mathbb{R}^{128 \times 512} \tag{23}$$

$$\hat{y} = \sigma(\mathbf{W}_2 \mathbf{o}_1 + \mathbf{b}_2), \quad \mathbf{W}_2 \in \mathbb{R}^{1 \times 128} \tag{24}$$

Table 13: BiGRU Predictor Layer Specifications

| Layer | Type | Input Dim | Output Dim |
|---|---|---|---|
| Input | – | $(B, T', 256)$ | $(B, T', 256)$ |
| BiGRU | GRU | 256 | 256 |
| | *hidden_size*=128, *num_layers*=2 | | |
| | *bidirectional*=True, *batch_first*=True, *dropout*=0.1 | | |
| Path 1: Last Hidden | Concat | $\mathbf{h}_{[-2]}, \mathbf{h}_{[-1]}$ | $(B, 256)$ |
| Path 2: Attention | Linear+Softmax | $(B, T', 256)$ | $(B, 256)$ |
| Concat Paths | Concat | $(B, 256) \times 2$ | $(B, 512)$ |
| Dropout | Dropout(0.1) | 512 | 512 |
| Dense-1 | Linear | 512 | 128 |
| Activation-1 | ReLU | 128 | 128 |
| Dense-2 | Linear | 128 | 1 |
| Activation-2 | Sigmoid | 1 | 1 |
| Output | – | $(B, 1)$ | $\hat{y} \in [0, 1]$ |

where $\sigma(\cdot)$ is the sigmoid function ensuring $\hat{y} \in [0, 1]$, and $\mathbf{w}_a \in \mathbb{R}^{256}$ are learned attention weights. The attention mask sets scores to $-\infty$ for zero-padded timesteps (from truncation augmentation), ensuring the model attends only to observed snapshots.

### C.2.4 BiGRU Training Configuration

- **Loss function**: Mean Absolute Error (MAE / L1)

$$\mathcal{L}_{\text{BiGRU}} = \frac{1}{B} \sum_{i=1}^{B} |\hat{y}_i - y_i| \tag{25}$$

where $y_i$ is the true accuracy of architecture $i$, expressed as a fraction in $[0, 1]$.

- **Alternative loss** (explored): Rank Consistency Loss (see §C.5)

- **Optimizer**: Adam with learning rate 0.001

- **Learning rate schedule**: OneCycleLR with max learning rate 0.003

- **Batch size**: 32

- **Epochs**: 100

- **Sequence augmentation**: Train on lengths $T' \in \{1, \ldots, 22\}$ with same target $y$

### C.3 Fusion Mechanism

At each time step $t$, embeddings from both CAEs are fused via concatenation:

$$\mathbf{z}_t = \begin{bmatrix} \mathbf{e}_t^W \\ \mathbf{e}_t^G \end{bmatrix} \in \mathbb{R}^{256} \tag{26}$$

where:

- $\mathbf{e}_t^W = \text{CAE}_W(\mathbf{X}_t^W) \in \mathbb{R}^{128}$: Weights embedding at time $t$

- $\mathbf{e}_t^G = \text{CAE}_G(\mathbf{X}_t^G) \in \mathbb{R}^{128}$: Gradients embedding at time $t$

- No additional learned projection is applied (simple concatenation)

The full sequence is then constructed:

$$\mathbf{Z} = [\mathbf{z}_1, \mathbf{z}_2, \ldots, \mathbf{z}_{T'}] \in \mathbb{R}^{T' \times 256} \tag{27}$$

## C.4   Design Choice Rationale and Alternatives Explored

This subsection states the ex-ante rationale for the three architectural choices that define the NAP2 pipeline – the feature-map encoder, the temporal predictor, and the predictor's hidden width and direction – and then documents one alternative architecture that was explored and discarded. Empirical validation of all three choices is provided by the ablations in App. H; an additional sensitivity check on the encoder family is reported in App. J.4.

### C.4.1   Design Choice Rationale

**Conv2D autoencoder.** The per-modality feature maps have shape $[L, T, F] = [65, 100, 12]$ – a natural $[H, W, C]$ image for 2D convolution: the layer ($L$) and training-step ($T$) axes carry local structure (adjacent layers share computational role; adjacent steps share training context), and $F$ is the per-position channel dimension. The encoder family is sensitivity-checked against a depth-blind DeepSets attention alternative on the size-controlled NB-101 sub-sample in App. J.4: Conv2D wins the standard regime (cross-space $\tau = 0.510$ vs 0.349 for DeepSets), confirming it as the right default for all main-paper experiments.

**BiGRU temporal predictor.** A single BiGRU operates across all query budgets snap1...snap23, whereas per-$k$ Static and Mean-Pool baselines require 23 separately trained models. On CIFAR-10 the ordered BiGRU beats Static-MLP at the shortest budget ($\tau_{\text{snap1}} = 0.747 \pm .018$ vs $0.738 \pm .007$) and stays within $\sim 0.01$ of MeanPool at saturation ($\tau_{\text{snap23}} = 0.821 \pm .016$ vs $0.831 \pm .003$); see App. H.2, Table 34. The BiGRU is therefore the right choice when one model must serve all query budgets. Bidirectional rather than unidirectional because backward context lets the predictor reinterpret early snapshots given final-snapshot convergence, an effect we observed consistently during model development.

**Hidden width $h$=128 and predictor hyperparameter selection.** The training pool is small (on the order of $10^3$ architectures $\times T$ snapshots), so over-parameterization is the dominant generalization risk. We selected the predictor configuration via a held-out Kendall $\tau$ search over hidden dimension $h \in \{64, 128, 256, 512, 1024, 2048\}$, loss function (MSE, MAE, soft-Spearman), learning-rate schedule (StepLR, OneCycleLR), augmentation range ($k = 1 \ldots 10$ vs $k = 1 \ldots 22$), and pooling strategy (last-hidden, mean, attention, dual-path). The selected configuration ($h$=128, MAE loss, OneCycleLR, $k = 1 \ldots 22$, dual-path attention pooling) achieved the best held-out KT with $\sim 29\times$ fewer parameters (659K) than the original LSTM $h$=2048 baseline. Wider hidden widths did not improve the metric and increased the train–test gap, consistent with the over-parameterization risk; full parameter counts in App. C.7.

### C.4.2   Combined LSTM (Separate Encoders)

An alternative architecture uses two separate LSTMs for weights and gradients, fusing only at the final hidden states:

$$\mathbf{h}_{T'}^W = \text{LSTM}_W(\mathbf{e}_1^W, \ldots, \mathbf{e}_{T'}^W) \tag{28}$$

$$\mathbf{h}_{T'}^G = \text{LSTM}_G(\mathbf{e}_1^G, \ldots, \mathbf{e}_{T'}^G) \tag{29}$$

$$\mathbf{h}_{\text{fused}} = [\mathbf{h}_{T'}^W; \mathbf{h}_{T'}^G] \in \mathbb{R}^{4096} \tag{30}$$

This variant was explored during predictor-architecture experimentation but did not improve over early fusion via concatenation, and was not retained.

### C.5 Loss Functions

### C.5.1 Primary Loss: Mean Absolute Error (MAE)

The main training objective for the BiGRU predictor:

$$\mathcal{L}_{\text{MAE}} = \frac{1}{B} \sum_{i=1}^{B} |\hat{y}_i - y_i| \tag{31}$$

**Rationale**: MAE is less sensitive to outliers than MSE and directly optimizes for ranking quality. Since the predictor is evaluated by rank correlation, only the relative ordering of predictions matters, not their absolute scale.

### C.5.2 Experimental Loss: Rank Consistency

A custom ranking loss based on soft Spearman correlation was explored:

$$\mathcal{L}_{\text{rank}} = 1 - \frac{6 \sum_{i=1}^{B} (\text{rank}(\hat{y}_i) - \text{rank}(y_i))^2}{B(B^2 - 1)} \tag{32}$$

where ranks are computed via soft-ranking approximation to maintain differentiability. This loss did not provide significant improvements over MAE in preliminary experiments.

### C.6 Target Normalization

Target accuracies are used in their raw form as fractions in $[0, 1]$ for all experiments (both in-domain and cross-dataset). No target normalization is applied. Since the predictor is evaluated by rank correlation (Kendall $\tau$), only the relative ordering of predictions matters. The log-normalization of feature maps (Section 3.3) handles scale differences across datasets on the input side, eliminating the need for target-side normalization.

### C.7 Model Size & Complexity

Table 14: Model Parameter Counts

| Component | Parameters | Type |
|---|---|---|
| CAE Encoder (W) | 9.17M | Trainable |
| CAE Decoder (W) | 9.17M | Trainable |
| CAE Encoder (G) | 9.17M | Trainable |
| CAE Decoder (G) | 9.17M | Trainable |
| BiGRU | 593K | Trainable |
| Attention + Dense | 66K | Trainable |
| **Total** | 37.99M | – |

**Parameter count details**:

- **CAE Encoder**: Three Conv2D layers (12→512, 512→256, 256→128 channels) with BatchNorm2d after each, totaling 9,169,536 parameters per encoder.

- **CAE Decoder**: Three ConvTranspose2D layers (128→256, 256→512, 512→12 channels) with BatchNorm2d after the first two, totaling 9,169,164 parameters per decoder.

- **BiGRU**: Two-layer bidirectional GRU with input size 256, hidden size 128, plus attention layer (256→1) and dense head (512→128→1), totaling approximately 659K parameters (593K for GRU core + 66K for

attention and dense layers). This represents a 29× reduction from the original LSTM predictor (19M parameters).

- CAEs are trained independently; only encoders are used during BiGRU training (decoders are discarded after CAE training).

## C.8 Visual Diagrams

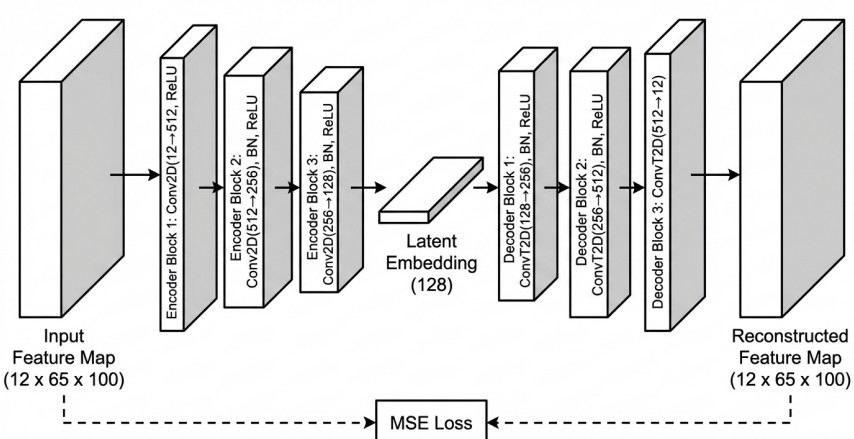

Figure 6: Convolutional auto-encoder (CAE) architecture with three-stage encoder (12→512→256→128) compressing feature maps to 128-dimensional embeddings, trained with MSE reconstruction loss.

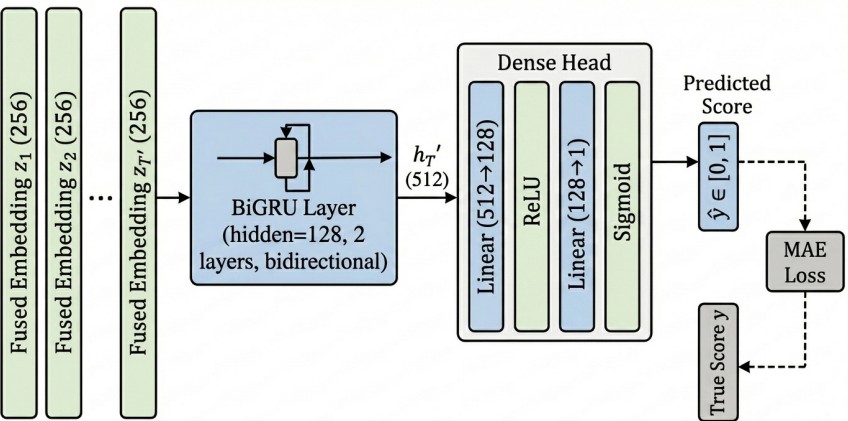

Fig. 1: The BiGRU predictor architecture described in the paper's Appendix C.2.

Figure 7: BiGRU predictor architecture processing fused embedding sequences through a 2-layer bidirectional GRU (hidden=128) with dual-path pooling (last hidden state + attention context), followed by a dense head to output performance scores $\hat{y} \in [0, 1]$, trained with MAE loss.

# D   Data & Preprocessing

This appendix specifies datasets, normalization parameters, and data augmentation used for training NAS-Bench-201 architectures to generate performance logs.

## D.1   Datasets

Table 15: Dataset Specifications

| Dataset | Size | Classes | Image Res. | Role |
|---------|------|---------|------------|------|
| CIFAR-10 | 50k / 10k | 10 | $32 \times 32$ | Primary |
| CIFAR-100 | 50k / 10k | 100 | $32 \times 32$ | Primary |
| ImageNet-16-120 | 151k / 6k | 120 | $16 \times 16$ | Primary |
| SVHN | 73k / 26k | 10 | $32 \times 32$ | Dist. shift |

## D.2   Normalization Parameters

Table 16: Per-Channel Normalization (applied as $(x/255 - \mu)/\sigma$)

| Dataset | Mean $\mu$ (R, G, B) | Std $\sigma$ (R, G, B) |
|---------|----------------------|------------------------|
| CIFAR-10 | $(0.4914, 0.4822, 0.4465)$ | $(0.2470, 0.2430, 0.2610)$ |
| CIFAR-100 | $(0.5071, 0.4867, 0.4408)$ | $(0.2675, 0.2565, 0.2761)$ |
| ImageNet-16-120 | Mean image subtraction[†] | |
| SVHN | $(0.5000, 0.5000, 0.5000)$ | $(0.5000, 0.5000, 0.5000)$ |

[†]ImageNet-16-120 uses mean image computed from training batches, subtracted before ToTensor conversion. Unlike other datasets, no per-channel normalization is applied after mean subtraction; the images are normalized to $[0, 1]$ via division by 255, then mean-subtracted, and converted to tensors directly.

## D.3   Data Augmentation

Table 17: Data Augmentation Applied During Architecture Training

| Operation | CIFAR-10/100 | ImageNet-16 | SVHN |
|-----------|--------------|-------------|------|
| Random Crop + Pad | $32 \times 32 + 4$ | $16 \times 16 + 2$ | None |
| Horizontal Flip | $p = 0.5$ | $p = 0.5$ | None |
| Normalize | Per-channel | Mean subtraction[*] | $[-1, 1]$ range |

**Pipeline**:

- **CIFAR-10/100**:   RandomHorizontalFlip($p$ = 0.5) → RandomCrop(padding) → ToTensor → Normalize($\mu, \sigma$)

- **ImageNet-16-120**: RandomHorizontalFlip($p = 0.5$) → RandomCrop(padding) → ToTensor (mean subtraction applied during data loading)

- **SVHN**: ToTensor → Normalize to $[-1, 1]$ range (no augmentation)

[*]ImageNet-16-120 applies mean image subtraction during data loading (before transforms), not per-channel normalization after ToTensor.

### D.4 Data Splits

#### D.4.1 Architecture Training

**CIFAR-10 and CIFAR-100**: Both datasets consist of 50,000 training samples and 10,000 test samples. Following the NAS-Bench-201 protocol, the full training set (50,000 samples) is used for training. The test set (10,000 samples) is split into a fixed validation subset and remaining test samples. Fixed validation indices are used to ensure reproducibility across all architecture training runs. This approach maintains consistency with the NAS-Bench-201 benchmark while providing a held-out validation set for architecture evaluation during training and a final test set for evaluation.

**ImageNet-16-120**: The dataset provides 151,700 training samples and approximately 6,000 validation samples as separate pre-processed sets. The validation set is used directly without further splitting.

**SVHN**: The dataset consists of 73,257 training samples and 26,032 test samples. The training set is split into 51,280 training samples and 21,977 validation samples using fixed indices to ensure reproducibility. The test set (26,032 samples) remains completely separate for final evaluation.

#### D.4.2 Predictor Training

Architectures (not data samples) are split for CAE/BiGRU training:

Table 18: Architecture Splits for Predictor Training

| Train | Validation | Test | Total |
|-------|------------|------|-------|
| 50–700 | 100 | 200 | 350–1000 |

Architectures sampled with stratified sampling across performance bins (50-55%, 55-60%, ..., 90-95% accuracy). Splits saved to `train_files.json`, `val_files.json`, `test_files.json`.

### D.5 Snapshot Protocol

Weights and gradients logged every $\Delta t = 100$ mini-batches during architecture training, yielding $T = 23$ snapshots over 16 epochs (CIFAR/ImageNet) or 40 epochs (SVHN). Snapshots compressed and saved as `weights_stats_<id>.bz2` and `gradients_stats_<id>.bz2`.

**Training Configuration**: SGD + Nesterov (momentum=0.9, weight decay=$5 \times 10^{-4}$), lr=0.1 with cosine decay, batch size=256. See Appendix C for full training protocol.

# E  Reproducibility & Computational Environment

This appendix documents the experimental design, randomness control, hardware specifications, and software environment to facilitate reproducibility.

## E.1  Experimental Design

We evaluate predictor performance across varying training set sizes to assess data efficiency. Each configuration is repeated over multiple independent trials to ensure statistical robustness.

Table 19: Experimental Design: Training Set Sizes and Trials

| Train Size | Trials | Total Runs |
|---|---|---|
| 50 | 10 | 10 |
| 100 | 10 | 10 |
| 200 | 10 | 10 |
| 300 | 10 | 10 |
| 400 | 10 | 10 |
| 500 | 10 | 10 |
| 600 | 10 | 10 |
| 700 | 10 | 10 |
| **Total Configurations** | **8** | **80 runs** |

**Split Configuration**: All experiments use fixed validation (100 architectures) and test (200 architectures) sets. Architectures are sampled with stratified sampling across performance bins to ensure balanced coverage of the performance distribution.

**Reporting**: All results presented as mean $\pm$ standard deviation over 10 independent trials.

## E.2  Randomness Control

We employ multi-level seeding to balance reproducibility with statistical robustness.

Table 20: Seed Configuration and Determinism Levels

| Component | Seed | Deterministic | Implementation |
|---|---|---|---|
| *Phase 1: Architecture Training (Log Generation)* | | | |
| Random/NumPy/PyTorch | 4 | Full | `set_seed()` with all flags |
| cuDNN operations | – | Full | `cudnn.deterministic=True` |
| cuDNN benchmark | – | Off | `cudnn.benchmark=False` |
| TensorFloat-32 | – | Off | `allow_tf32=False` |
| CUBLAS workspace | – | Set | `CUBLAS_WORKSPACE_CONFIG=':16:8'` |
| *Phase 2: Data Splitting* | | | |
| Train/Val/Test split | 42 | Full | `train_test_split(random_state=42)` |
| Split persistence | – | Yes | Saved to JSON, reused across trials |
| *Phase 3: Predictor Training (CAE + BiGRU)* | | | |
| Weight initialization | None | Stochastic | PyTorch default initialization |
| Optimizer state | None | Stochastic | Adam with random initial state |
| Data shuffling | None | Stochastic | DataLoader shuffling |
| *Statistical Robustness Strategy* | | | |
| Independent trials | – | – | 10 trials per configuration |
| Aggregation | – | – | Report mean $\pm$ SD |

### E.2.1 Determinism Flags (Architecture Training)

Full determinism is enforced during architecture training via `set_seed()` function:

```
random.seed(4)
np.random.seed(4)
torch.manual_seed(4)
torch.cuda.manual_seed_all(4)
os.environ['CUBLAS_WORKSPACE_CONFIG'] = ':16:8'
torch.backends.cudnn.deterministic = True
torch.backends.cudnn.benchmark = False
torch.backends.cudnn.allow_tf32 = False
```

**Rationale**: Architecture training must be deterministic to ensure identical weights/gradients logs across predictor training trials. Predictor training (CAE/BiGRU) uses stochastic initialization with multiple trials for statistical robustness.

### E.3 Hardware Specifications

Table 21: Computational Hardware

| Component | Specification | Details |
|---|---|---|
| **GPU** | NVIDIA GeForce GTX 1080 Ti | – |
| - Memory | 11 GB GDDR5X | – |
| - Architecture | Pascal (GP102) | Compute Cap. 6.1 |
| - TDP | 250W | – |
| **CPU** | Intel Xeon E5-1650 v4 | – |
| - Base Frequency | 3.60 GHz | – |
| - Cores / Threads | 6 / 12 | – |
| **Memory** | 125 GB DDR4 | – |
| **Cluster** | Slurm-managed HPC | University cluster |
| - Partitions | main, gtx1080, rtx2080, etc. | – |

**GPU Selection**: The GTX 1080 Ti was chosen to match prior NAS predictor work (White et al., 2021), enabling direct timing comparisons. Query times reported in the main paper are measured on this hardware.

### E.4 Software Environment

### E.5 Training Time & Computational Cost

*Note*: Times are approximate and vary with training set size. Reported query times in main paper (e.g., 54s, 108s, 207s) refer to architecture training only and are measured on GTX 1080 Ti.

### E.6 Code Availability

All code, trained models, and data splits are available at: [repository URL to be added]. The repository includes:

- Architecture training scripts with snapshot logging

- Statistics extraction and feature map creation

- CAE and BiGRU training code with all hyperparameters

- Pre-computed data splits (JSON files) for exact reproducibility

Table 22: Software Versions

| Software | Version |
|---|---|
| **Operating System** | Rocky Linux 9.5 (Blue Onyx) |
| Kernel | 5.14.0-503.40.1.el9_5.x86_64 |
| **Python** | 3.9.21 |
| PyTorch | $\geq$ 1.13 |
| CUDA (Driver) | 12.5 (Driver 555.42.06) |
| cuDNN | $\geq$ 8.0 |
| **Core Libraries** | |
| NumPy | 2.0.2 |
| SciPy | $\geq$ 1.10 |
| Pandas | $\geq$ 1.5 |
| Scikit-learn | $\geq$ 1.0 |
| **Infrastructure** | |
| Cluster Manager | Slurm |
| Conda Environment | Custom environment per component |

Table 23: Approximate Training Times (per trial, GTX 1080 Ti)

| Component | Time per Architecture | Time per Trial |
|---|---|---|
| Architecture Training (16 epochs) | 3-4 minutes | – |
| Snapshot T=23 | Included above | – |
| Statistics Extraction | 30 seconds | – |
| Feature Map Creation | 10 seconds | – |
| CAE Training (150 epochs) | – | 2-4 hours |
| BiGRU Training (100 epochs) | – | 20-40 min |
| **Total per Trial** | – | 4-7 hours (train_size=700) |
| **Total for 80 Experiments** | – | 320-560 GPU hours |

- Evaluation scripts and analysis notebooks

- Slurm job scripts for cluster deployment

## F  Top-K Identification Analysis

This appendix provides a comprehensive analysis of NAP2's ability to identify top-performing architectures through Top-K recall and Top-N Coverage metrics. While Kendall's Tau measures overall ranking quality, these metrics directly address practical NAS questions: *Can NAP2 reliably identify the best architectures early in training, and how many promising candidates can it identify simultaneously?*

### F.1  Metric Definitions

**Top-K Recall**: For each trial, we compute the probability that the true best architecture (the single architecture with highest final accuracy, i.e., true top-1) appears within the predicted top-$K$ architectures (ranked by predicted score). Formally:

$$\text{Top-}K \text{ Recall} = \frac{1}{T} \sum_{t=1}^{T} \mathbb{1}[\text{true top-1} \in \text{predicted top-}K] \tag{33}$$

where $T = 10$ is the number of trials (200 architectures per trial). Values range from 0 (never found) to 1 (always found).

**Top-N Coverage**: Measures the average proportion of true top-$N$ architectures that appear in predicted top-$K$ rankings. For each trial, we count how many of the true top-$N$ architectures appear in predicted top-$K$, then divide by $N$ to get the proportion. Values are averages of these proportions across trials, ranging from 0.0 (none found) to 1.0 (all found).

The distinction is crucial for NAS: Top-K Recall answers whether the single best architecture is found (binary), while Top-N Coverage reveals NAP2's ability to identify multiple promising candidates simultaneously (continuous)—enabling parallel exploration and robust ensemble evaluation in NAS workflows.

### F.2  In-Domain Results

Table 24 reports Top-K recall for in-domain setups. With the new pipeline, NAP2 attains very strong Top-K recall from the first mini-batch step on CIFAR-10 (Top-3≥0.7, Top-10=1.0) and ImageNet-16-120 (Top-20=1.0). On CIFAR-100, Top-20=1.0 from step 3 onward and Top-10 reaches the 0.9–1.0 range from step 5 onward. Performance is largely saturated by mini-batch step 5 (500 mini-batches), giving practitioners a usable early-stopping signal at all three target datasets.

Tables 25–27 report Top-N Coverage metrics, measuring the average proportion of true top-$N$ architectures found in predicted top-$K$ rankings. Column labels follow the pattern "Top-$N$ in Top-$K$": e.g., "Top-3 in Top-20" means the proportion of true top-3 architectures found in predicted top-20.

### F.3  Cross-Dataset Results

Tables 28–30 report Top-K recall for cross-dataset setups. NAP2 transfers strongly between CIFAR-10 and CIFAR-100 in both directions and from CIFAR-100 to ImageNet-16-120: CIFAR-10 → CIFAR-100 reaches Top-5=1.0 by step 2 and Top-10/Top-20=1.0 from the first step; CIFAR-100 → ImageNet-16-120 reaches Top-10=1.0 by step 3 and Top-20=1.0 by step 2; CIFAR-100 → CIFAR-10 reaches Top-20=1.0 from the first step and Top-10=1.0 by step 12.

### F.4  Key Insights & Practical Implications

**Main Findings**:

1. **Early Identification**: Top-20 recall reaches 1.0 from the first 100 mini-batches on CIFAR-10 and ImageNet-16-120, and reaches 1.0 by step 3 on CIFAR-100, enabling early search space narrowing in NAS frameworks

Table 24: Top-K recall for in-domain datasets (train_size=700). Values represent the probability that the true best architecture appears in NAP2's predicted top-$K$ rankings, averaged across 10 trials.

| mini-batch step ($\times 10^2$) | CIFAR-10 | | | | CIFAR-100 | | | | ImageNet-16-120 | | | |
|---|---|---|---|---|---|---|---|---|---|---|---|---|
| | Top-3 | Top-5 | Top-10 | Top-20 | Top-3 | Top-5 | Top-10 | Top-20 | Top-3 | Top-5 | Top-10 | Top-20 |
| 1 | 0.8 | 0.9 | 1.0 | 1.0 | 0.0 | 0.0 | 0.1 | 0.6 | 0.4 | 0.5 | 0.7 | 1.0 |
| 2 | 0.7 | 0.9 | 1.0 | 1.0 | 0.0 | 0.3 | 0.6 | 0.9 | 0.3 | 0.5 | 0.8 | 1.0 |
| 3 | 0.8 | 1.0 | 1.0 | 1.0 | 0.1 | 0.2 | 0.6 | 1.0 | 0.2 | 0.3 | 0.7 | 1.0 |
| 4 | 0.9 | 0.9 | 1.0 | 1.0 | 0.2 | 0.3 | 0.8 | 1.0 | 0.2 | 0.4 | 0.7 | 1.0 |
| 5 | 0.9 | 0.9 | 1.0 | 1.0 | 0.4 | 0.4 | 1.0 | 1.0 | 0.1 | 0.4 | 0.8 | 1.0 |
| 6 | 0.9 | 0.9 | 1.0 | 1.0 | 0.4 | 0.4 | 0.9 | 1.0 | 0.1 | 0.4 | 0.8 | 1.0 |
| 7 | 0.9 | 0.9 | 1.0 | 1.0 | 0.4 | 0.4 | 0.9 | 1.0 | 0.1 | 0.4 | 0.7 | 1.0 |
| 8 | 0.8 | 0.9 | 1.0 | 1.0 | 0.2 | 0.3 | 1.0 | 1.0 | 0.2 | 0.4 | 0.6 | 1.0 |
| 9 | 0.8 | 0.9 | 1.0 | 1.0 | 0.3 | 0.4 | 0.9 | 1.0 | 0.2 | 0.6 | 0.8 | 1.0 |
| 10 | 0.8 | 1.0 | 1.0 | 1.0 | 0.3 | 0.4 | 1.0 | 1.0 | 0.2 | 0.6 | 0.8 | 1.0 |
| 11 | 0.8 | 1.0 | 1.0 | 1.0 | 0.3 | 0.4 | 1.0 | 1.0 | 0.2 | 0.6 | 0.8 | 1.0 |
| 12 | 0.9 | 1.0 | 1.0 | 1.0 | 0.1 | 0.5 | 1.0 | 1.0 | 0.2 | 0.6 | 0.8 | 1.0 |
| 13 | 0.8 | 1.0 | 1.0 | 1.0 | 0.1 | 0.5 | 1.0 | 1.0 | 0.2 | 0.6 | 0.8 | 1.0 |
| 14 | 0.8 | 1.0 | 1.0 | 1.0 | 0.1 | 0.5 | 1.0 | 1.0 | 0.1 | 0.6 | 0.8 | 1.0 |
| 15 | 0.9 | 1.0 | 1.0 | 1.0 | 0.2 | 0.5 | 1.0 | 1.0 | 0.1 | 0.6 | 0.8 | 1.0 |
| 16 | 0.9 | 1.0 | 1.0 | 1.0 | 0.2 | 0.5 | 1.0 | 1.0 | 0.1 | 0.6 | 0.8 | 1.0 |
| 17 | 0.7 | 1.0 | 1.0 | 1.0 | 0.2 | 0.5 | 1.0 | 1.0 | 0.1 | 0.6 | 0.8 | 1.0 |
| 18 | 0.7 | 1.0 | 1.0 | 1.0 | 0.2 | 0.5 | 1.0 | 1.0 | 0.2 | 0.5 | 0.8 | 1.0 |
| 19 | 0.7 | 1.0 | 1.0 | 1.0 | 0.2 | 0.4 | 1.0 | 1.0 | 0.2 | 0.5 | 0.8 | 1.0 |
| 20 | 0.7 | 1.0 | 1.0 | 1.0 | 0.2 | 0.5 | 1.0 | 1.0 | 0.1 | 0.4 | 0.8 | 1.0 |
| 21 | 0.7 | 0.9 | 1.0 | 1.0 | 0.4 | 0.5 | 1.0 | 1.0 | 0.2 | 0.5 | 0.8 | 1.0 |
| 22 | 0.8 | 0.8 | 1.0 | 1.0 | 0.4 | 0.5 | 1.0 | 1.0 | 0.2 | 0.5 | 0.8 | 1.0 |
| 23 | 0.9 | 0.9 | 1.0 | 1.0 | 0.4 | 0.6 | 0.9 | 1.0 | 0.3 | 0.4 | 0.8 | 1.0 |

Table 25: Top-N Coverage in Top-$K$ for in-domain CIFAR-10 (train_size=700). Values are averaged across 10 trials.

| mini-batch step ($\times 10^2$) | Top-3 in Top-3 | Top-3 in Top-5 | Top-3 in Top-10 | Top-3 in Top-20 | Top-5 in Top-5 | Top-5 in Top-10 | Top-5 in Top-20 | Top-10 in Top-10 | Top-10 in Top-20 |
|---|---|---|---|---|---|---|---|---|---|
| 1 | 0.267 | 0.300 | 0.367 | 0.533 | 0.440 | 0.620 | 0.720 | 0.610 | 0.740 |
| 2 | 0.233 | 0.300 | 0.400 | 0.667 | 0.400 | 0.600 | 0.800 | 0.600 | 0.790 |
| 3 | 0.267 | 0.367 | 0.533 | 0.767 | 0.420 | 0.680 | 0.860 | 0.640 | 0.820 |
| 4 | 0.333 | 0.367 | 0.533 | 0.767 | 0.440 | 0.700 | 0.860 | 0.640 | 0.840 |
| 5 | 0.300 | 0.367 | 0.567 | 0.800 | 0.500 | 0.720 | 0.880 | 0.650 | 0.850 |
| 6 | 0.300 | 0.367 | 0.567 | 0.833 | 0.460 | 0.720 | 0.900 | 0.670 | 0.880 |
| 7 | 0.333 | 0.367 | 0.567 | 0.800 | 0.460 | 0.720 | 0.880 | 0.670 | 0.870 |
| 8 | 0.333 | 0.367 | 0.600 | 0.867 | 0.520 | 0.760 | 0.920 | 0.690 | 0.890 |
| 9 | 0.333 | 0.400 | 0.633 | 0.867 | 0.520 | 0.780 | 0.920 | 0.690 | 0.890 |
| 10 | 0.367 | 0.433 | 0.633 | 0.900 | 0.520 | 0.760 | 0.940 | 0.670 | 0.880 |
| 11 | 0.333 | 0.433 | 0.600 | 0.867 | 0.520 | 0.740 | 0.920 | 0.670 | 0.870 |
| 12 | 0.333 | 0.467 | 0.600 | 0.900 | 0.560 | 0.740 | 0.940 | 0.670 | 0.870 |
| 13 | 0.367 | 0.467 | 0.600 | 0.900 | 0.560 | 0.740 | 0.940 | 0.670 | 0.880 |
| 14 | 0.367 | 0.500 | 0.600 | 0.900 | 0.580 | 0.740 | 0.940 | 0.670 | 0.880 |
| 15 | 0.400 | 0.500 | 0.633 | 0.900 | 0.580 | 0.760 | 0.940 | 0.670 | 0.870 |
| 16 | 0.367 | 0.500 | 0.600 | 0.900 | 0.580 | 0.740 | 0.940 | 0.660 | 0.870 |
| 17 | 0.300 | 0.500 | 0.633 | 0.900 | 0.580 | 0.760 | 0.940 | 0.660 | 0.870 |
| 18 | 0.333 | 0.500 | 0.633 | 0.867 | 0.580 | 0.760 | 0.920 | 0.670 | 0.860 |
| 19 | 0.333 | 0.500 | 0.633 | 0.867 | 0.540 | 0.760 | 0.920 | 0.660 | 0.850 |
| 20 | 0.333 | 0.500 | 0.633 | 0.867 | 0.520 | 0.760 | 0.920 | 0.670 | 0.860 |
| 21 | 0.333 | 0.467 | 0.633 | 0.867 | 0.480 | 0.760 | 0.920 | 0.670 | 0.860 |
| 22 | 0.367 | 0.400 | 0.633 | 0.867 | 0.460 | 0.760 | 0.920 | 0.670 | 0.840 |
| 23 | 0.367 | 0.467 | 0.633 | 0.900 | 0.520 | 0.760 | 0.940 | 0.650 | 0.860 |

2. **Cross-Dataset Transfer**: Tables 28–30 show strong Top-K recall when trained on different datasets, demonstrating NAP2's transferability without retraining. Notably, CIFAR-10 → CIFAR-100 reaches Top-5=1.0 by step 2, CIFAR-100 → ImageNet-16-120 reaches Top-10=1.0 by step 3, and CIFAR-100 → CIFAR-10 reaches Top-10=1.0 by step 12

3. **Multi-Architecture Identification**: Top-N Coverage analysis reveals NAP2 successfully identifies multiple high-performing architectures simultaneously (e.g., Top-5 Coverage in Top-20 reaches 0.94 on CIFAR-10 by step 10 and 0.96 on ImageNet-16-120 by step 9), enabling parallel exploration of promising candidates

Table 26: Top-N Coverage in Top-$K$ for in-domain CIFAR-100 (train_size=700). Values are averaged across 10 trials.

| mini-batch step ($\times 10^2$) | Top-3 in Top-3 | Top-3 in Top-5 | Top-3 in Top-10 | Top-3 in Top-20 | Top-5 in Top-5 | Top-5 in Top-10 | Top-5 in Top-20 | Top-10 in Top-10 | Top-10 in Top-20 |
|---|---|---|---|---|---|---|---|---|---|
| 1 | 0.000 | 0.000 | 0.033 | 0.200 | 0.120 | 0.320 | 0.500 | 0.430 | 0.550 |
| 2 | 0.000 | 0.100 | 0.233 | 0.467 | 0.300 | 0.460 | 0.660 | 0.480 | 0.620 |
| 3 | 0.033 | 0.100 | 0.300 | 0.533 | 0.280 | 0.480 | 0.680 | 0.500 | 0.650 |
| 4 | 0.067 | 0.100 | 0.367 | 0.567 | 0.280 | 0.520 | 0.700 | 0.510 | 0.660 |
| 5 | 0.133 | 0.133 | 0.500 | 0.600 | 0.220 | 0.580 | 0.700 | 0.550 | 0.670 |
| 6 | 0.133 | 0.133 | 0.467 | 0.633 | 0.260 | 0.580 | 0.720 | 0.550 | 0.670 |
| 7 | 0.133 | 0.133 | 0.467 | 0.633 | 0.220 | 0.540 | 0.720 | 0.540 | 0.670 |
| 8 | 0.067 | 0.100 | 0.500 | 0.667 | 0.160 | 0.560 | 0.720 | 0.540 | 0.670 |
| 9 | 0.100 | 0.133 | 0.467 | 0.600 | 0.200 | 0.540 | 0.680 | 0.530 | 0.640 |
| 10 | 0.100 | 0.133 | 0.467 | 0.600 | 0.160 | 0.540 | 0.680 | 0.540 | 0.640 |
| 11 | 0.100 | 0.133 | 0.467 | 0.600 | 0.180 | 0.540 | 0.680 | 0.540 | 0.640 |
| 12 | 0.033 | 0.167 | 0.467 | 0.600 | 0.180 | 0.540 | 0.680 | 0.530 | 0.640 |
| 13 | 0.033 | 0.167 | 0.467 | 0.600 | 0.200 | 0.540 | 0.680 | 0.520 | 0.640 |
| 14 | 0.033 | 0.200 | 0.467 | 0.600 | 0.180 | 0.540 | 0.680 | 0.520 | 0.640 |
| 15 | 0.067 | 0.200 | 0.467 | 0.600 | 0.200 | 0.540 | 0.700 | 0.520 | 0.650 |
| 16 | 0.067 | 0.167 | 0.433 | 0.600 | 0.180 | 0.520 | 0.700 | 0.510 | 0.660 |
| 17 | 0.067 | 0.167 | 0.433 | 0.600 | 0.160 | 0.520 | 0.680 | 0.510 | 0.640 |
| 18 | 0.067 | 0.167 | 0.433 | 0.600 | 0.200 | 0.520 | 0.680 | 0.510 | 0.640 |
| 19 | 0.067 | 0.133 | 0.400 | 0.633 | 0.200 | 0.520 | 0.700 | 0.510 | 0.640 |
| 20 | 0.067 | 0.167 | 0.400 | 0.600 | 0.220 | 0.540 | 0.700 | 0.520 | 0.640 |
| 21 | 0.133 | 0.167 | 0.367 | 0.633 | 0.220 | 0.500 | 0.740 | 0.500 | 0.660 |
| 22 | 0.133 | 0.167 | 0.400 | 0.667 | 0.220 | 0.520 | 0.780 | 0.500 | 0.680 |
| 23 | 0.133 | 0.200 | 0.367 | 0.667 | 0.220 | 0.500 | 0.780 | 0.480 | 0.680 |

Table 27: Top-N Coverage in Top-$K$ for in-domain ImageNet-16-120 (train_size=700). Values are averaged across 10 trials.

| mini-batch step ($\times 10^2$) | Top-3 in Top-3 | Top-3 in Top-5 | Top-3 in Top-10 | Top-3 in Top-20 | Top-5 in Top-5 | Top-5 in Top-10 | Top-5 in Top-20 | Top-10 in Top-10 | Top-10 in Top-20 |
|---|---|---|---|---|---|---|---|---|---|
| 1 | 0.200 | 0.333 | 0.633 | 0.933 | 0.240 | 0.540 | 0.900 | 0.350 | 0.610 |
| 2 | 0.300 | 0.500 | 0.667 | 0.900 | 0.360 | 0.640 | 0.880 | 0.390 | 0.640 |
| 3 | 0.300 | 0.467 | 0.767 | 0.967 | 0.340 | 0.640 | 0.940 | 0.400 | 0.660 |
| 4 | 0.333 | 0.667 | 0.800 | 0.933 | 0.520 | 0.720 | 0.920 | 0.440 | 0.630 |
| 5 | 0.400 | 0.633 | 0.833 | 0.967 | 0.480 | 0.720 | 0.940 | 0.460 | 0.680 |
| 6 | 0.500 | 0.700 | 0.833 | 0.967 | 0.560 | 0.760 | 0.940 | 0.470 | 0.720 |
| 7 | 0.533 | 0.700 | 0.833 | 0.967 | 0.520 | 0.760 | 0.940 | 0.460 | 0.710 |
| 8 | 0.533 | 0.700 | 0.800 | 0.967 | 0.540 | 0.780 | 0.940 | 0.470 | 0.730 |
| 9 | 0.533 | 0.767 | 0.867 | 1.000 | 0.560 | 0.840 | 0.960 | 0.510 | 0.730 |
| 10 | 0.533 | 0.767 | 0.867 | 1.000 | 0.580 | 0.840 | 0.960 | 0.520 | 0.740 |
| 11 | 0.533 | 0.767 | 0.867 | 1.000 | 0.560 | 0.840 | 0.960 | 0.520 | 0.750 |
| 12 | 0.533 | 0.767 | 0.867 | 1.000 | 0.580 | 0.840 | 0.960 | 0.520 | 0.750 |
| 13 | 0.533 | 0.767 | 0.867 | 1.000 | 0.600 | 0.860 | 0.960 | 0.530 | 0.750 |
| 14 | 0.533 | 0.733 | 0.867 | 1.000 | 0.580 | 0.860 | 0.960 | 0.530 | 0.760 |
| 15 | 0.533 | 0.733 | 0.867 | 1.000 | 0.560 | 0.860 | 0.960 | 0.540 | 0.770 |
| 16 | 0.533 | 0.733 | 0.867 | 1.000 | 0.580 | 0.860 | 0.960 | 0.540 | 0.770 |
| 17 | 0.533 | 0.733 | 0.867 | 1.000 | 0.560 | 0.860 | 0.960 | 0.540 | 0.780 |
| 18 | 0.567 | 0.700 | 0.867 | 1.000 | 0.560 | 0.860 | 0.960 | 0.550 | 0.770 |
| 19 | 0.567 | 0.700 | 0.867 | 1.000 | 0.580 | 0.860 | 0.960 | 0.540 | 0.770 |
| 20 | 0.500 | 0.667 | 0.867 | 1.000 | 0.560 | 0.860 | 0.960 | 0.540 | 0.770 |
| 21 | 0.533 | 0.700 | 0.867 | 1.000 | 0.560 | 0.860 | 0.960 | 0.530 | 0.780 |
| 22 | 0.533 | 0.700 | 0.867 | 1.000 | 0.580 | 0.860 | 0.960 | 0.510 | 0.780 |
| 23 | 0.533 | 0.667 | 0.867 | 1.000 | 0.560 | 0.860 | 0.960 | 0.510 | 0.770 |

4. **Monotonic Improvement**: Both metrics improve with query time, with Top-K recall generally plateauing after a few hundred mini-batches

**Practical Benefits for NAS**:

- **Search Space Reduction**: Early identification of top-$K$ candidates allows NAS frameworks to reduce the search space from thousands to tens of architectures, dramatically reducing computational cost

- **Resource Allocation**: Focusing computational budget on promising architectures identified early (Top-20 recall $\geq 0.9$ from 100 mini-batches on CIFAR-10/ImageNet-16-120 in-domain, and from 200 mini-batches on CIFAR-100) maximizes efficiency

Table 28: Top-K recall for cross-dataset setups where models are trained on CIFAR-10 and tested on CIFAR-100 and ImageNet-16-120. Values represent the probability that the true best architecture appears in NAP2's predicted top-$K$ rankings, averaged across 10 trials.

| mini-batch step ($\times 10^2$) | CIFAR-100 | | | | ImageNet-16-120 | | | |
|---|---|---|---|---|---|---|---|---|
| | Top-3 | Top-5 | Top-10 | Top-20 | Top-3 | Top-5 | Top-10 | Top-20 |
| 1 | 0.5 | 0.9 | 1.0 | 1.0 | 0.0 | 0.0 | 0.0 | 0.0 |
| 2 | 0.6 | 1.0 | 1.0 | 1.0 | 0.0 | 0.0 | 0.0 | 0.1 |
| 3 | 0.4 | 1.0 | 1.0 | 1.0 | 0.0 | 0.0 | 0.0 | 0.1 |
| 4 | 0.6 | 1.0 | 1.0 | 1.0 | 0.0 | 0.0 | 0.0 | 0.1 |
| 5 | 0.6 | 1.0 | 1.0 | 1.0 | 0.0 | 0.0 | 0.0 | 0.2 |
| 6 | 0.7 | 1.0 | 1.0 | 1.0 | 0.0 | 0.0 | 0.0 | 0.2 |
| 7 | 0.6 | 1.0 | 1.0 | 1.0 | 0.0 | 0.0 | 0.0 | 0.3 |
| 8 | 0.6 | 1.0 | 1.0 | 1.0 | 0.0 | 0.0 | 0.0 | 0.3 |
| 9 | 0.7 | 1.0 | 1.0 | 1.0 | 0.0 | 0.0 | 0.0 | 0.4 |
| 10 | 0.7 | 1.0 | 1.0 | 1.0 | 0.0 | 0.0 | 0.0 | 0.4 |
| 11 | 0.7 | 1.0 | 1.0 | 1.0 | 0.0 | 0.0 | 0.1 | 0.4 |
| 12 | 0.7 | 1.0 | 1.0 | 1.0 | 0.0 | 0.0 | 0.1 | 0.4 |
| 13 | 0.6 | 1.0 | 1.0 | 1.0 | 0.0 | 0.0 | 0.1 | 0.4 |
| 14 | 0.7 | 1.0 | 1.0 | 1.0 | 0.0 | 0.0 | 0.1 | 0.4 |
| 15 | 0.7 | 1.0 | 1.0 | 1.0 | 0.0 | 0.0 | 0.1 | 0.4 |
| 16 | 0.7 | 0.9 | 1.0 | 1.0 | 0.0 | 0.0 | 0.1 | 0.5 |
| 17 | 0.7 | 0.9 | 1.0 | 1.0 | 0.0 | 0.0 | 0.1 | 0.6 |
| 18 | 0.8 | 0.9 | 1.0 | 1.0 | 0.0 | 0.0 | 0.1 | 0.5 |
| 19 | 0.8 | 0.9 | 1.0 | 1.0 | 0.0 | 0.0 | 0.1 | 0.5 |
| 20 | 0.8 | 0.9 | 1.0 | 1.0 | 0.0 | 0.0 | 0.1 | 0.5 |
| 21 | 0.7 | 1.0 | 1.0 | 1.0 | 0.0 | 0.0 | 0.1 | 0.5 |
| 22 | 0.8 | 0.9 | 1.0 | 1.0 | 0.0 | 0.0 | 0.1 | 0.5 |
| 23 | 0.8 | 0.9 | 1.0 | 1.0 | 0.0 | 0.0 | 0.1 | 0.6 |

Table 29: Top-K recall for cross-dataset setups where models are trained on CIFAR-100 and tested on CIFAR-10 and ImageNet-16-120. Values represent the probability that the true best architecture appears in NAP2's predicted top-$K$ rankings, averaged across 10 trials.

| mini-batch step ($\times 10^2$) | CIFAR-10 | | | | ImageNet-16-120 | | | |
|---|---|---|---|---|---|---|---|---|
| | Top-3 | Top-5 | Top-10 | Top-20 | Top-3 | Top-5 | Top-10 | Top-20 |
| 1 | 0.2 | 0.3 | 0.8 | 1.0 | 0.3 | 0.6 | 0.9 | 0.9 |
| 2 | 0.2 | 0.5 | 0.9 | 1.0 | 0.7 | 0.8 | 0.9 | 1.0 |
| 3 | 0.1 | 0.4 | 0.8 | 1.0 | 0.8 | 0.9 | 1.0 | 1.0 |
| 4 | 0.2 | 0.6 | 0.8 | 1.0 | 0.7 | 0.9 | 1.0 | 1.0 |
| 5 | 0.2 | 0.7 | 0.8 | 1.0 | 0.7 | 0.8 | 1.0 | 1.0 |
| 6 | 0.4 | 0.7 | 0.9 | 1.0 | 0.7 | 0.8 | 1.0 | 1.0 |
| 7 | 0.3 | 0.7 | 0.9 | 1.0 | 0.7 | 1.0 | 1.0 | 1.0 |
| 8 | 0.3 | 0.8 | 0.9 | 1.0 | 0.6 | 1.0 | 1.0 | 1.0 |
| 9 | 0.1 | 0.8 | 0.9 | 1.0 | 0.7 | 1.0 | 1.0 | 1.0 |
| 10 | 0.2 | 0.8 | 0.9 | 1.0 | 0.6 | 1.0 | 1.0 | 1.0 |
| 11 | 0.2 | 0.8 | 0.9 | 1.0 | 0.6 | 1.0 | 1.0 | 1.0 |
| 12 | 0.2 | 0.8 | 1.0 | 1.0 | 0.6 | 1.0 | 1.0 | 1.0 |
| 13 | 0.2 | 0.8 | 1.0 | 1.0 | 0.7 | 1.0 | 1.0 | 1.0 |
| 14 | 0.1 | 0.8 | 1.0 | 1.0 | 0.7 | 1.0 | 1.0 | 1.0 |
| 15 | 0.2 | 0.8 | 1.0 | 1.0 | 0.7 | 1.0 | 1.0 | 1.0 |
| 16 | 0.2 | 0.8 | 1.0 | 1.0 | 0.7 | 1.0 | 1.0 | 1.0 |
| 17 | 0.2 | 0.7 | 1.0 | 1.0 | 0.7 | 1.0 | 1.0 | 1.0 |
| 18 | 0.2 | 0.7 | 1.0 | 1.0 | 0.7 | 1.0 | 1.0 | 1.0 |
| 19 | 0.2 | 0.7 | 1.0 | 1.0 | 0.7 | 1.0 | 1.0 | 1.0 |
| 20 | 0.2 | 0.7 | 1.0 | 1.0 | 0.7 | 1.0 | 1.0 | 1.0 |
| 21 | 0.2 | 0.4 | 1.0 | 1.0 | 0.7 | 1.0 | 1.0 | 1.0 |
| 22 | 0.3 | 0.6 | 1.0 | 1.0 | 0.7 | 1.0 | 1.0 | 1.0 |
| 23 | 0.2 | 0.5 | 1.0 | 1.0 | 0.8 | 1.0 | 1.0 | 1.0 |

- **Early Stopping**: Top-K metrics provide concrete signals for early stopping—once Top-20 recall plateaus, additional training provides limited value

- **Parallel Exploration**: High Top-N Coverage (e.g., Top-5 Coverage in Top-20 $\geq 0.85$ on CIFAR-10 and ImageNet-16-120, and reaching 0.78 on CIFAR-100) enables simultaneous evaluation of multiple promising candidates, accelerating the search process

Table 30: Top-K recall for cross-dataset setups where models are trained on ImageNet-16-120 and tested on CIFAR-10 and CIFAR-100. Values represent the probability that the true best architecture appears in NAP2's predicted top-$K$ rankings, averaged across 10 trials.

| mini-batch step ($\times 10^2$) | CIFAR-10 | | | | CIFAR-100 | | | |
|---|---|---|---|---|---|---|---|---|
| | Top-3 | Top-5 | Top-10 | Top-20 | Top-3 | Top-5 | Top-10 | Top-20 |
| 1 | 0.1 | 0.4 | 0.7 | 0.9 | 0.0 | 0.0 | 0.4 | 0.7 |
| 2 | 0.1 | 0.5 | 0.7 | 0.9 | 0.0 | 0.0 | 0.2 | 0.5 |
| 3 | 0.2 | 0.3 | 0.7 | 1.0 | 0.0 | 0.0 | 0.3 | 0.7 |
| 4 | 0.3 | 0.3 | 0.6 | 1.0 | 0.0 | 0.1 | 0.3 | 0.7 |
| 5 | 0.3 | 0.3 | 0.7 | 1.0 | 0.0 | 0.2 | 0.4 | 0.7 |
| 6 | 0.4 | 0.4 | 0.5 | 1.0 | 0.0 | 0.2 | 0.4 | 0.6 |
| 7 | 0.3 | 0.3 | 0.6 | 0.9 | 0.0 | 0.2 | 0.4 | 0.6 |
| 8 | 0.3 | 0.4 | 0.5 | 0.9 | 0.1 | 0.2 | 0.4 | 0.5 |
| 9 | 0.3 | 0.3 | 0.5 | 0.9 | 0.1 | 0.2 | 0.4 | 0.5 |
| 10 | 0.3 | 0.3 | 0.5 | 0.9 | 0.1 | 0.2 | 0.3 | 0.5 |
| 11 | 0.3 | 0.3 | 0.5 | 0.9 | 0.1 | 0.2 | 0.4 | 0.5 |
| 12 | 0.3 | 0.3 | 0.5 | 0.9 | 0.1 | 0.3 | 0.3 | 0.5 |
| 13 | 0.3 | 0.3 | 0.5 | 0.9 | 0.1 | 0.3 | 0.3 | 0.5 |
| 14 | 0.3 | 0.3 | 0.5 | 0.9 | 0.1 | 0.3 | 0.3 | 0.5 |
| 15 | 0.3 | 0.3 | 0.5 | 0.9 | 0.1 | 0.2 | 0.3 | 0.5 |
| 16 | 0.3 | 0.3 | 0.5 | 0.9 | 0.1 | 0.2 | 0.3 | 0.5 |
| 17 | 0.3 | 0.3 | 0.5 | 0.8 | 0.1 | 0.2 | 0.3 | 0.5 |
| 18 | 0.3 | 0.3 | 0.5 | 0.8 | 0.1 | 0.2 | 0.3 | 0.6 |
| 19 | 0.3 | 0.4 | 0.6 | 0.8 | 0.1 | 0.2 | 0.3 | 0.6 |
| 20 | 0.3 | 0.4 | 0.6 | 0.8 | 0.1 | 0.3 | 0.3 | 0.6 |
| 21 | 0.3 | 0.3 | 0.5 | 0.8 | 0.1 | 0.3 | 0.3 | 0.6 |
| 22 | 0.3 | 0.3 | 0.5 | 0.8 | 0.2 | 0.3 | 0.4 | 0.6 |
| 23 | 0.2 | 0.3 | 0.5 | 0.8 | 0.2 | 0.4 | 0.4 | 0.6 |

- **Transfer Learning**: Strong cross-dataset Top-K recall enables applying models trained on fast datasets (CIFAR-10) to larger tasks (CIFAR-100), and from CIFAR-100 to ImageNet-16-120, without retraining

# G   Theoretical Foundations

This appendix provides a detailed theoretical treatment of the principles underlying NAP2's design. We formalize the connections between our meta-features and established generalization theory, justify the temporal modeling approach, and analyze the conditions under which cross-dataset transfer is expected to succeed.

## G.1   Norm-Based Features and PAC-Bayes Generalization Bounds

We begin by formalizing the relationship between NAP2's norm-based features and known generalization bounds. Consider a feedforward network $f_w$ with $L$ layers, where $W_l$ denotes the weight matrix of layer $l$. The PAC-Bayes framework McAllester (1999) provides generalization bounds of the form:

$$L_{\text{test}}(w) \leq L_{\text{train}}(w) + \mathcal{O}\left( \sqrt{\frac{\text{KL}(q \,\|\, p)}{n}} \right) \tag{34}$$

where $q$ is a posterior distribution centered on the learned weights $w$, $p$ is a data-independent prior, $n$ is the number of training examples, and KL denotes the Kullback–Leibler divergence. When both $q$ and $p$ are Gaussian, the KL term reduces to a function of $\|w\|^2$ (plus terms depending on the variances), establishing a direct link between weight norms and generalization.

More specifically, Neyshabur et al. (2017) showed that for a network with ReLU activations, the generalization bound scales as:

$$\mathcal{O}\left( \frac{B^2}{\gamma^2} \cdot \prod_{l=1}^{L} \|W_l\|_\sigma^2 \cdot \sum_{l=1}^{L} \frac{\|W_l\|_F^2}{\|W_l\|_\sigma^2} \cdot \frac{\ln L}{n} \right) \tag{35}$$

where $\|W_l\|_\sigma$ is the spectral norm of layer $l$, $\|W_l\|_F$ is the Frobenius norm, $B$ is the input norm bound, and $\gamma$ is the classification margin. The terms $\|W_l\|_F$ and $\|W_l\|_\sigma$ are both captured (or closely approximated) by NAP2's per-layer L2 norm features: the L2 norm computed over all weights in a layer corresponds to the Frobenius norm, while the maximum singular value is bounded by the Frobenius norm and reflected in the max and variance statistics across units.

Similarly, Bartlett et al. (2017) derived margin-based bounds that depend on the product of spectral norms $\prod_l \|W_l\|_\sigma$ weighted by per-layer $(2,1)$-matrix norms. Our L1 norm features (the sum of absolute values per layer) relate to these $(2,1)$-norms, providing the predictor with a signal that tracks this specific complexity measure.

The key insight is that NAP2 does not need to compute these bounds explicitly—it only needs features that correlate with the quantities appearing in them. By providing per-layer L1 and L2 norms, along with distributional statistics (variance, standard deviation) that characterize the spread of unit-level norms within each layer, NAP2 gives the temporal predictor sufficient information to implicitly learn a function that approximates these complexity–generalization relationships.

## G.2   Distributional Shape Features and Spectral Properties

The kurtosis and skewness features in NAP2 are motivated by connections between weight matrix distributional properties and generalization quality. Martin & Mahoney (2021) applied random matrix theory to study the empirical spectral density (ESD) of weight matrices in trained neural networks. Their central finding is that well-generalizing networks tend to develop weight matrices whose ESDs exhibit heavy-tailed behavior, well-approximated by a power-law distribution. The fitted power-law exponent $\hat{\alpha}$ was shown to be strongly predictive of test accuracy across a range of architectures and datasets, without access to test data.

Computing full ESDs requires singular value decomposition at each layer, which is expensive. However, the tail behavior of a distribution is intimately connected to its higher-order moments:

- **Kurtosis** measures the heaviness of the tails relative to a Gaussian distribution. A distribution with power-law tails (the hallmark of well-generalizing networks per Martin & Mahoney) exhibits high

kurtosis. Intuitively, per-layer kurtosis of the weight distribution can thus serve as a computationally cheap proxy for the ESD tail exponent.

- **Skewness** captures asymmetry in the weight distribution. During training, the development of asymmetric weight distributions can reflect the emergence of specialized features. Symmetric distributions (zero skewness) are characteristic of initialization, while increasing skewness during training reflects the breaking of this symmetry as the network learns.

Together, kurtosis and skewness provide a compact characterization of the distributional shape that captures generalization-relevant spectral properties at a fraction of the computational cost of full spectral analysis. The covariance feature further captures inter-unit relationships within a layer, reflecting the degree of functional specialization: low covariance indicates redundant units, while structured covariance suggests efficient use of the layer's representational capacity.

### G.3 Gradient Features and Loss Landscape Curvature

Gradient-based features provide complementary information about the local geometry of the loss surface. The *empirical* Fisher information matrix at parameters $w$ is defined as:

$$\hat{F}(w) = \frac{1}{n} \sum_{i=1}^{n} \nabla_w \ell(w, x_i) \cdot \nabla_w \ell(w, x_i)^\top \tag{36}$$

where $\ell(w, x_i)$ is the per-sample loss. The diagonal entries of $\hat{F}$ correspond to the squared gradient magnitudes per parameter, and its eigenspectrum characterizes the curvature of the loss surface in different directions. While computing the full Fisher is prohibitively expensive, our per-layer gradient statistics capture summary information about these quantities:

- **Gradient L2 norms** approximate the trace of the per-layer Fisher block, which equals the sum of its eigenvalues. This provides a scalar summary of the total curvature magnitude at each layer.

- **Gradient variance and standard deviation** across units within a layer reflect the anisotropy of the local curvature. High variance indicates that some directions in parameter space have much steeper loss curvature than others—a condition associated with sharp minima and potentially poor generalization Keskar et al. (2016).

- **Gradient kurtosis** captures whether the curvature is concentrated in a few extreme directions (high kurtosis) or spread more uniformly (low kurtosis), reflecting the effective dimensionality of the loss landscape curvature.

The combination of weight and gradient features at each snapshot provides a joint characterization of the network's position in parameter space and the local geometry of the loss surface at that position. This is analogous to characterizing a physical system by both its state (position) and its dynamics (forces/accelerations), which is strictly more informative than either alone.

### G.4 Formal Justification for Temporal Modeling

Let $\phi(w_t) \in \mathbb{R}^d$ denote the feature vector extracted from the network weights at training step $t$ (encompassing both weight and gradient statistics). A static predictor observes a single $\phi(w_{t_0})$ and must predict final performance $y$ from this point estimate alone. NAP2, by contrast, observes a sequence $\{\phi(w_{t_1}), \phi(w_{t_2}), \ldots, \phi(w_{t_k})\}$ and can therefore access temporal information including:

1. **First-order dynamics (velocity):** $\Delta\phi_t = \phi(w_{t+1}) - \phi(w_t)$, which captures the rate of change of each feature. Rapidly decreasing gradient norms indicate fast convergence, while oscillating norms suggest instability.

2. **Second-order dynamics (acceleration):** $\Delta^2\phi_t = \Delta\phi_{t+1} - \Delta\phi_t$, which captures whether the optimization is accelerating, decelerating, or oscillating. This is relevant to phenomena such as the edge of stability Cohen et al. (2021), where the sharpness of the loss surface is constrained by the step size.

3. **Trajectory shape:** The overall pattern of the sequence—monotonic convergence, oscillation with decreasing amplitude, sudden transitions—provides information about the loss landscape topology that cannot be inferred from any finite number of derivatives at a single point.

Formally, consider two architectures $A_1$ and $A_2$ such that $\phi(w_{t_0}^{A_1}) = \phi(w_{t_0}^{A_2})$ at some snapshot $t_0$—that is, they are indistinguishable from a single observation. If their trajectories diverge (e.g., $A_1$'s gradient norms decrease monotonically while $A_2$'s oscillate), the sequence-based predictor can distinguish them while the static predictor cannot. The bidirectional GRU architecture is a natural choice for modeling such sequential dependencies, as its gating mechanism captures both short-term fluctuations and long-term trends, while bidirectionality allows the model to condition on the full observed trajectory when making predictions.

The empirical observation that NAP2's performance improves rapidly over the first approximately 10 snapshots and then plateaus (Section 6.1) is consistent with this framework: the early snapshots provide the most informative trajectory features (capturing the initial transient dynamics), while later snapshots contain diminishing marginal information as the optimization settles into a more predictable regime.

### G.5 Conditions for Cross-Dataset Transfer

We provide a more formal analysis of why NAP2's features should support cross-dataset transfer. Let $\mathcal{D}_s$ and $\mathcal{D}_t$ denote the source and target datasets, and let $f_A$ denote a fixed architecture $A$. We argue that NAP2's transfer relies on the following conditions, each with theoretical and empirical support:

**Condition 1: Architecture-dependent loss landscape structure.** For a fixed architecture $A$, the topology of the loss landscape $\mathcal{L}_{A,\mathcal{D}}(w) = \mathbb{E}_{(x,y)\sim\mathcal{D}}[\ell(f_A(x;w), y)]$—including the distribution and connectivity of local minima—is primarily determined by $A$. Choromanska et al. (2015) showed that under simplifying assumptions, the loss function of a deep network with random i.i.d. inputs has a structure where local minima concentrate near a global band, with the number and depth of these minima determined by the network size (a property of $A$, not $\mathcal{D}$). Li et al. (2018) provided complementary empirical evidence that network architecture—particularly depth, width, and the presence of skip connections—has a dramatic effect on loss landscape geometry, with these structural properties remaining qualitatively consistent across training runs.

**Condition 2: Qualitative preservation of optimization dynamics.** The early training dynamics—the specific sequence of weight updates, gradient magnitudes, and their evolution—are governed by the interaction between the architecture's parameterization and the optimizer. While the specific numerical values change with the dataset, the qualitative patterns (which layers converge fastest, how gradient magnitudes relate across layers, whether optimization is stable or oscillatory) are largely determined by architectural properties such as depth, skip connections, and activation functions. The implicit bias literature Gunasekar et al. (2018) provides theoretical support: gradient descent converges to solutions with specific structural properties determined by the architecture's parameterization.

**Condition 3: Rank-based evaluation and input normalization remove dataset-specific calibration.** Since NAP2 is evaluated by rank correlation (Kendall $\tau$), only the relative ordering of predictions matters, not their absolute scale. Target accuracies are used in their raw form as fractions in $[0, 1]$. On the input side, log-normalization of feature maps ($x \mapsto \text{sign}(x) \cdot \log(1 + |x|)$) compresses the dynamic range of per-layer statistics, placing feature maps from different datasets into a comparable range without requiring per-dataset calibration. If architectures $A_1$ and $A_2$ satisfy $\text{acc}(A_1, \mathcal{D}_s) > \text{acc}(A_2, \mathcal{D}_s)$ because $A_1$ has better optimization dynamics on $\mathcal{D}_s$, and if Conditions 1 and 2 hold, then the same relative ordering is likely preserved on $\mathcal{D}_t$.

**When transfer may fail.** These conditions are not absolute. If the target dataset has qualitatively different properties—for example, extremely different input dimensionality, label structure, or noise characteristics—the optimization dynamics may diverge sufficiently to break the architecture-dependent patterns. The SVHN experiment (Section 5.3) provides an encouraging result: even under significant distribution shift (natural images vs. digit crops with very different accuracy distributions), transfer remains effective, suggesting the architecture-dependent signal in early training dynamics is robust. However, we expect that transfer to fundamentally different tasks (e.g., from image classification to sequence modeling) would require at minimum a retraining of the predictor, as the feature distributions would shift substantially.

### G.6 Summary of Design Choice Justifications

We summarize the theoretical justification for each major design choice in NAP2 in Table 31.

Table 31: Summary of theoretical justifications for NAP2's design choices.

| Design Choice | Theoretical Justification |
|---|---|
| L1 and L2 norm features | Track quantities appearing in PAC-Bayes and spectral-norm generalization bounds Neyshabur et al. (2017); Bartlett et al. (2017) |
| Kurtosis and skewness | Computationally efficient proxies for heavy-tail spectral properties linked to generalization Martin & Mahoney (2021) |
| Gradient statistics | Capture local loss landscape curvature via relationship to the Fisher information matrix eigenspectrum |
| Combined weight + gradient features | Joint state-and-dynamics characterization; strictly more informative than either alone |
| Temporal sequence of snapshots | Captures trajectory velocity and acceleration, enabling discrimination between architectures indistinguishable from single-point observations |
| Bidirectional GRU predictor | Gating captures short- and long-term dynamics; bidirectionality conditions on full observed trajectory; dual-path attention pooling provides interpretable snapshot importance |
| 100 mini-batch snapshot interval | Balances temporal resolution with storage overhead; sufficient to capture early transient dynamics where most predictive signal concentrates |
| Log-normalization of feature maps | Compresses heavy-tailed feature distributions across datasets, enabling shared encoder representations without target-side rescaling |
| Cross-dataset transfer | Supported by architecture-dependent loss landscape structure Choromanska et al. (2015); Li et al. (2018) and qualitative preservation of optimization dynamics Gunasekar et al. (2018) |

## H  Ablation Studies

This appendix presents three ablation studies on the design choices of NAP2: (i) modality (weights, gradients, both), (ii) temporal modeling (ordered sequence vs. static / mean-pool / shuffled baselines), and (iii) feature families (general, distribution, norm-based statistics from Section 3.2). All ablations use the pipeline described in the main paper (BiGRU dual-path with $h = 128$, log-normalized feature maps, augmentation $k = 1 \ldots 22$, L1 loss, OneCycleLR), 10 random seeds per condition, and the same train/val/test splits as the main paper experiments. Numeric values report mean $\pm$ standard deviation across the 10 seeds.

### H.1  Modality: Weights vs. Gradients vs. Both

**Protocol.** Each architecture's $[T' \times 256]$ embedding sequence (where $T'$ is the number of snapshots used at inference, up to 23) is sliced into the gradient half (first 128 dimensions, produced by the gradient CAE

encoder) or the weight half (last 128 dimensions, produced by the weight CAE encoder), or kept as the full 256-dimensional concatenation. The same BiGRU dual-path predictor is trained from scratch on each modality slice.

### In-domain results

Table 32: Modality ablation, in-domain (train_size=700). Mean Kendall Tau $\pm$ std over 10 seeds.

| Dataset | Condition | snap1 | snap3 | snap5 | snap10 | snap15 | snap23 |
|---|---|---|---|---|---|---|---|
| CIFAR-10 | Both (default) | 0.747±.018 | 0.790±.012 | 0.804±.016 | 0.814±.017 | 0.817±.017 | 0.821±.016 |
| | Weight-only | 0.726±.006 | 0.801±.008 | 0.819±.005 | 0.831±.005 | 0.838±.005 | 0.839±.004 |
| | Gradient-only | 0.735±.019 | 0.773±.004 | 0.785±.010 | 0.790±.011 | 0.793±.012 | 0.795±.012 |
| CIFAR-100 | Both (default) | 0.678±.007 | 0.720±.006 | 0.739±.006 | 0.754±.010 | 0.757±.010 | 0.760±.012 |
| | Weight-only | 0.634±.012 | 0.703±.009 | 0.739±.008 | 0.755±.007 | 0.754±.007 | 0.756±.007 |
| | Gradient-only | 0.654±.006 | 0.713±.012 | 0.734±.014 | 0.758±.016 | 0.763±.016 | 0.767±.019 |
| ImageNet-16-120 | Both (default) | 0.701±.011 | 0.734±.008 | 0.746±.013 | 0.761±.017 | 0.767±.018 | 0.767±.019 |
| | Weight-only | 0.631±.011 | 0.697±.012 | 0.730±.014 | 0.760±.010 | 0.774±.010 | 0.780±.009 |
| | Gradient-only | 0.680±.014 | 0.726±.009 | 0.742±.008 | 0.756±.008 | 0.763±.007 | 0.763±.009 |

### Cross-dataset results

Table 33: Modality ablation, cross-dataset (all 6 source→target pairs). Mean Kendall Tau $\pm$ std over 10 seeds.

| Pair | Condition | snap1 | snap3 | snap5 | snap10 | snap15 | snap23 |
|---|---|---|---|---|---|---|---|
| C10→C100 | Both (default) | 0.639±.021 | 0.672±.022 | 0.683±.020 | 0.702±.011 | 0.709±.010 | 0.717±.009 |
| | Weight-only | 0.608±.018 | 0.674±.007 | 0.686±.010 | 0.686±.011 | 0.688±.011 | 0.693±.009 |
| | Gradient-only | 0.630±.008 | 0.655±.016 | 0.669±.020 | 0.677±.021 | 0.680±.023 | 0.685±.024 |
| C10→IMG | Both (default) | 0.619±.011 | 0.637±.018 | 0.634±.023 | 0.646±.024 | 0.664±.021 | 0.685±.021 |
| | Weight-only | 0.576±.011 | 0.608±.010 | 0.623±.015 | 0.656±.009 | 0.671±.008 | 0.678±.008 |
| | Gradient-only | 0.608±.017 | 0.640±.009 | 0.651±.008 | 0.659±.011 | 0.663±.012 | 0.669±.014 |
| C100→C10 | Both (default) | 0.713±.018 | 0.753±.019 | 0.769±.017 | 0.777±.017 | 0.782±.017 | 0.777±.015 |
| | Weight-only | 0.685±.021 | 0.755±.011 | 0.747±.006 | 0.722±.007 | 0.726±.007 | 0.720±.009 |
| | Gradient-only | 0.703±.009 | 0.740±.011 | 0.741±.009 | 0.735±.018 | 0.736±.024 | 0.725±.034 |
| C100→IMG | Both (default) | 0.643±.009 | 0.670±.011 | 0.679±.010 | 0.687±.011 | 0.694±.013 | 0.698±.013 |
| | Weight-only | 0.608±.012 | 0.648±.011 | 0.644±.014 | 0.644±.008 | 0.650±.009 | 0.656±.008 |
| | Gradient-only | 0.616±.008 | 0.662±.007 | 0.680±.009 | 0.694±.012 | 0.702±.013 | 0.708±.015 |
| IMG→C10 | Both (default) | 0.739±.016 | 0.777±.018 | 0.784±.020 | 0.789±.018 | 0.787±.017 | 0.786±.016 |
| | Weight-only | 0.655±.027 | 0.723±.020 | 0.739±.013 | 0.753±.006 | 0.759±.004 | 0.754±.004 |
| | Gradient-only | 0.739±.008 | 0.782±.006 | 0.793±.007 | 0.799±.013 | 0.796±.012 | 0.794±.009 |
| IMG→C100 | Both (default) | 0.657±.012 | 0.708±.016 | 0.724±.022 | 0.734±.025 | 0.733±.026 | 0.731±.027 |
| | Weight-only | 0.602±.009 | 0.640±.012 | 0.656±.009 | 0.674±.005 | 0.678±.005 | 0.684±.006 |
| | Gradient-only | 0.659±.008 | 0.711±.009 | 0.730±.011 | 0.739±.014 | 0.739±.015 | 0.736±.016 |

### Discussiony

Combining weights and gradients gives the strongest default at short query times. At snap1, the joint representation wins or ties in 8 of 9 in-domain and cross-dataset settings; the only exception is IMG→C100, where gradient-only is higher by 0.002. It is also never the worst across those short-budget settings. At longer budgets, single modalities can outperform the joint representation in dataset-specific ways: weight-only on in-domain CIFAR-10 and ImageNet-16-120, gradient-only on three of four cross-dataset directions involving ImageNet. Weight features mostly carry architectural priors; gradient features mostly carry optimization dynamics. When the deployment regime is unknown, the joint representation is the safer choice.

## H.2 Temporal Modeling: Ordered vs. Static vs. Mean-Pool vs. Shuffled

**Protocol.** Four conditions are compared at each query budget $k \in \{1, \ldots, 23\}$:

- **Ordered** (default): one BiGRU dual-path predictor on the ordered snapshot sequence, queried at any $k$. A single deployable model serves all query budgets via the augmentation strategy described in the main paper.

- **Static (per-$k$ MLP)**: a dedicated MLP on snapshot $k$ alone, trained and evaluated independently for each $k$. This produces 23 separate models, each specialized to one query budget.

- **Mean-Pool (per-$k$ MLP)**: a dedicated MLP on the mean of snapshots $1 \ldots k$. Also 23 separate models.

- **Shuffled**: the BiGRU on a randomly permuted snapshot order. Same architecture as Ordered, with sequence ordering destroyed.

All conditions use the both-modality 256-dimensional embedding, 10 seeds per cell.

**In-domain results**

Table 34: Temporal ablation, in-domain (train_size=700). Mean Kendall Tau $\pm$ std over 10 seeds.

| Dataset | Condition | snap1 | snap3 | snap5 | snap10 | snap15 | snap23 |
|---------|-----------|-------|-------|-------|--------|--------|--------|
| CIFAR-10 | Ordered (default) | 0.747±.018 | 0.790±.012 | 0.804±.016 | 0.814±.017 | 0.817±.017 | 0.821±.016 |
| | Static (per-$k$ MLP) | 0.738±.007 | 0.777±.012 | 0.790±.013 | 0.798±.013 | 0.806±.003 | 0.799±.004 |
| | MeanPool (per-$k$ MLP) | 0.738±.007 | 0.790±.005 | 0.802±.006 | 0.816±.004 | 0.824±.003 | 0.831±.003 |
| | Shuffled (BiGRU) | 0.792±.005 | 0.809±.006 | 0.813±.007 | 0.813±.007 | 0.813±.007 | 0.813±.008 |
| CIFAR-100 | Ordered (default) | 0.678±.007 | 0.720±.006 | 0.739±.006 | 0.754±.010 | 0.757±.010 | 0.760±.012 |
| | Static (per-$k$ MLP) | 0.672±.008 | 0.706±.011 | 0.729±.010 | 0.763±.007 | 0.739±.005 | 0.757±.008 |
| | MeanPool (per-$k$ MLP) | 0.672±.008 | 0.706±.021 | 0.732±.015 | 0.772±.010 | 0.786±.003 | 0.789±.003 |
| | Shuffled (BiGRU) | 0.730±.013 | 0.754±.012 | 0.753±.008 | 0.755±.007 | 0.756±.008 | 0.756±.007 |
| ImageNet-16-120 | Ordered (default) | 0.701±.011 | 0.734±.008 | 0.746±.013 | 0.761±.017 | 0.767±.018 | 0.767±.019 |
| | Static (per-$k$ MLP) | 0.707±.007 | 0.717±.009 | 0.733±.006 | 0.779±.006 | 0.783±.004 | 0.773±.007 |
| | MeanPool (per-$k$ MLP) | 0.707±.007 | 0.730±.007 | 0.749±.003 | 0.768±.005 | 0.781±.004 | 0.792±.004 |
| | Shuffled (BiGRU) | 0.739±.006 | 0.759±.006 | 0.766±.005 | 0.768±.006 | 0.769±.006 | 0.769±.006 |

**Cross-dataset results**

Table 35: Temporal ablation, cross-dataset. Mean Kendall Tau $\pm$ std over 10 seeds. Note: Static and Mean-Pool conditions show anomalous collapse on CIFAR-10$\rightarrow$\{CIFAR-100, ImageNet\} directions (highlighted), under investigation; the Ordered BiGRU and Shuffled BiGRU conditions are not affected.

| Pair | Condition | snap1 | snap3 | snap5 | snap10 | snap15 | snap23 |
|---|---|---|---|---|---|---|---|
| C10$\rightarrow$C100 | Ordered (default) | 0.639$\pm$.021 | 0.672$\pm$.022 | 0.683$\pm$.020 | 0.702$\pm$.011 | 0.709$\pm$.010 | 0.717$\pm$.009 |
| | Static (per-$k$ MLP)$^\dagger$ | 0.588$\pm$.015 | 0.451$\pm$.075 | 0.405$\pm$.020 | 0.432$\pm$.049 | 0.417$\pm$.039 | 0.397$\pm$.051 |
| | MeanPool (per-$k$ MLP)$^\dagger$ | 0.588$\pm$.015 | 0.580$\pm$.023 | 0.574$\pm$.018 | 0.475$\pm$.038 | 0.415$\pm$.024 | 0.374$\pm$.043 |
| | Shuffled (BiGRU) | 0.707$\pm$.027 | 0.716$\pm$.030 | 0.718$\pm$.027 | 0.722$\pm$.025 | 0.723$\pm$.026 | 0.728$\pm$.016 |
| C10$\rightarrow$IMG | Ordered (default) | 0.619$\pm$.011 | 0.637$\pm$.018 | 0.634$\pm$.023 | 0.646$\pm$.024 | 0.664$\pm$.021 | 0.685$\pm$.021 |
| | Static (per-$k$ MLP)$^\dagger$ | 0.562$\pm$.013 | 0.552$\pm$.046 | 0.534$\pm$.016 | 0.584$\pm$.033 | 0.504$\pm$.042 | 0.507$\pm$.048 |
| | MeanPool (per-$k$ MLP)$^\dagger$ | 0.562$\pm$.013 | 0.610$\pm$.014 | 0.604$\pm$.006 | 0.566$\pm$.019 | 0.543$\pm$.021 | 0.520$\pm$.060 |
| | Shuffled (BiGRU) | 0.671$\pm$.011 | 0.685$\pm$.009 | 0.688$\pm$.008 | 0.690$\pm$.008 | 0.691$\pm$.008 | 0.693$\pm$.010 |
| C100$\rightarrow$C10 | Ordered (default) | 0.713$\pm$.018 | 0.753$\pm$.019 | 0.769$\pm$.017 | 0.777$\pm$.017 | 0.782$\pm$.017 | 0.777$\pm$.015 |
| | Static (per-$k$ MLP) | 0.703$\pm$.011 | 0.737$\pm$.005 | 0.734$\pm$.006 | 0.769$\pm$.005 | 0.769$\pm$.004 | 0.769$\pm$.007 |
| | MeanPool (per-$k$ MLP) | 0.703$\pm$.011 | 0.750$\pm$.008 | 0.754$\pm$.010 | 0.753$\pm$.015 | 0.763$\pm$.008 | 0.786$\pm$.006 |
| | Shuffled (BiGRU) | 0.731$\pm$.008 | 0.757$\pm$.009 | 0.758$\pm$.007 | 0.760$\pm$.006 | 0.762$\pm$.006 | 0.762$\pm$.008 |
| C100$\rightarrow$IMG | Ordered (default) | 0.643$\pm$.009 | 0.670$\pm$.011 | 0.679$\pm$.010 | 0.687$\pm$.011 | 0.694$\pm$.013 | 0.698$\pm$.013 |
| | Static (per-$k$ MLP) | 0.663$\pm$.005 | 0.667$\pm$.007 | 0.687$\pm$.006 | 0.697$\pm$.006 | 0.688$\pm$.008 | 0.691$\pm$.005 |
| | MeanPool (per-$k$ MLP) | 0.663$\pm$.005 | 0.680$\pm$.006 | 0.693$\pm$.004 | 0.691$\pm$.007 | 0.690$\pm$.003 | 0.704$\pm$.005 |
| | Shuffled (BiGRU) | 0.652$\pm$.014 | 0.671$\pm$.014 | 0.673$\pm$.010 | 0.676$\pm$.011 | 0.677$\pm$.011 | 0.677$\pm$.013 |
| IMG$\rightarrow$C10 | Ordered (default) | 0.739$\pm$.016 | 0.777$\pm$.018 | 0.784$\pm$.020 | 0.789$\pm$.018 | 0.787$\pm$.017 | 0.786$\pm$.016 |
| | Static (per-$k$ MLP) | 0.741$\pm$.005 | 0.785$\pm$.007 | 0.786$\pm$.003 | 0.787$\pm$.006 | 0.777$\pm$.006 | 0.767$\pm$.002 |
| | MeanPool (per-$k$ MLP) | 0.741$\pm$.005 | 0.792$\pm$.004 | 0.789$\pm$.005 | 0.793$\pm$.003 | 0.794$\pm$.004 | 0.801$\pm$.003 |
| | Shuffled (BiGRU) | 0.758$\pm$.009 | 0.780$\pm$.009 | 0.780$\pm$.010 | 0.782$\pm$.010 | 0.784$\pm$.010 | 0.784$\pm$.008 |
| IMG$\rightarrow$C100 | Ordered (default) | 0.657$\pm$.012 | 0.708$\pm$.016 | 0.724$\pm$.022 | 0.734$\pm$.025 | 0.733$\pm$.026 | 0.731$\pm$.027 |
| | Static (per-$k$ MLP) | 0.658$\pm$.012 | 0.722$\pm$.012 | 0.744$\pm$.006 | 0.762$\pm$.008 | 0.736$\pm$.011 | 0.735$\pm$.007 |
| | MeanPool (per-$k$ MLP) | 0.658$\pm$.012 | 0.721$\pm$.007 | 0.747$\pm$.004 | 0.758$\pm$.006 | 0.760$\pm$.007 | 0.766$\pm$.003 |
| | Shuffled (BiGRU) | 0.719$\pm$.010 | 0.726$\pm$.007 | 0.728$\pm$.007 | 0.727$\pm$.007 | 0.727$\pm$.007 | 0.727$\pm$.007 |

$^\dagger$ Anomalous collapse on C10-source cross-dataset directions; likely embedding-space mismatch between per-$k$ MLP training and CAE pretraini

**Discussion**

The Ordered BiGRU is competitive from snap1 through snap23 with a single deployable model serving all query budgets, while per-$k$ MeanPool requires training and selecting 23 separate models per dataset. With that 23$\times$ model-count cost, MeanPool is modestly higher at snap23 in all 7 non-anomalous settings (CIFAR-10 in-domain 0.831 vs. Ordered 0.821, ImageNet in-domain 0.792 vs. 0.767, with the other 5 deltas in between). MeanPool also beats per-$k$ Static at snap23 in the same 7 settings, so multi-snapshot aggregation helps even without explicit sequence order. The Shuffled BiGRU is higher than Ordered at very short query times because our augmentation exposes it to later-snapshot embeddings under early-truncation queries; Ordered then overtakes it at long query times in the four non-anomalous cross-dataset settings. The CIFAR-10$\rightarrow$\{CIFAR-100, ImageNet\} collapse is confined to the per-$k$ MLP conditions (likely an embedding-space mismatch between per-$k$ MLP training and CAE pretraining); both BiGRU variants are unaffected, and we leave the resolution to future work.

### H.3 Feature Families: General vs. Distribution vs. Norm-Based

**Protocol.** The 12 statistical features defined in Section 3.2 are partitioned into three families:

- **General** (7 channels): mean, variance, median, std, max, min, percentiles

- **Distribution** (3 channels): covariance, skewness, kurtosis

- **Norm** (2 channels): $L_1$-norm, $L_2$-norm

For each subset, the full pipeline is rerun: feature maps are extracted using only the channels in the subset, the convolutional autoencoders are retrained on these channel-sliced maps, architectures are re-encoded, and a fresh BiGRU predictor is trained. Both modalities (gradient + weight) are used throughout. Scope:

CIFAR-100 vertical slice (in-domain CIFAR-100, plus CIFAR-100 $\rightarrow$ {CIFAR-10, ImageNet-16-120}), 10 seeds per condition.

**In-domain CIFAR-100**

Table 36: Feature-family ablation, in-domain CIFAR-100. Mean Kendall Tau $\pm$ std over 10 seeds.

| Condition | snap1 | snap3 | snap5 | snap10 | snap15 | snap23 |
|---|---|---|---|---|---|---|
| All 12 features (default) | 0.678±.007 | 0.720±.006 | 0.739±.006 | 0.754±.010 | 0.757±.010 | 0.760±.012 |
| General only (7ch) | 0.609±.009 | 0.656±.010 | 0.668±.011 | 0.690±.014 | 0.698±.016 | 0.704±.014 |
| Distribution only (3ch) | 0.614±.022 | 0.684±.018 | 0.712±.013 | 0.740±.010 | 0.745±.009 | 0.743±.006 |
| Norm only (2ch) | 0.724±.011 | 0.772±.007 | 0.795±.008 | 0.809±.008 | 0.815±.008 | 0.816±.008 |
| General + Distribution (10ch) | 0.621±.007 | 0.686±.006 | 0.701±.007 | 0.731±.007 | 0.740±.008 | 0.742±.010 |
| General + Norm (9ch) | 0.737±.009 | 0.773±.008 | 0.786±.009 | 0.800±.010 | 0.803±.009 | 0.803±.009 |
| Distribution + Norm (5ch) | 0.676±.004 | 0.714±.005 | 0.725±.005 | 0.735±.004 | 0.738±.004 | 0.745±.002 |

**Cross-dataset CIFAR-100 $\rightarrow$ CIFAR-10**

Table 37: Feature-family ablation, cross-dataset CIFAR-100 $\rightarrow$ CIFAR-10. Mean Kendall Tau $\pm$ std over 10 seeds.

| Condition | snap1 | snap3 | snap5 | snap10 | snap15 | snap23 |
|---|---|---|---|---|---|---|
| All 12 features (default) | 0.713±.018 | 0.753±.019 | 0.769±.017 | 0.777±.017 | 0.782±.017 | 0.777±.015 |
| General only (7ch) | 0.664±.009 | 0.701±.007 | 0.713±.007 | 0.727±.007 | 0.730±.007 | 0.728±.010 |
| Distribution only (3ch) | 0.666±.020 | 0.690±.028 | 0.691±.037 | 0.712±.039 | 0.721±.036 | 0.727±.034 |
| Norm only (2ch) | 0.701±.011 | 0.729±.014 | 0.739±.013 | 0.735±.014 | 0.727±.016 | 0.705±.018 |
| General + Distribution (10ch) | 0.656±.017 | 0.690±.015 | 0.687±.017 | 0.695±.016 | 0.709±.017 | 0.718±.013 |
| General + Norm (9ch) | 0.665±.013 | 0.679±.008 | 0.681±.008 | 0.674±.012 | 0.675±.016 | 0.667±.016 |
| Distribution + Norm (5ch) | 0.728±.012 | 0.771±.011 | 0.782±.008 | 0.799±.007 | 0.802±.007 | 0.805±.006 |

**Cross-dataset CIFAR-100 $\rightarrow$ ImageNet-16-120**

Table 38: Feature-family ablation, cross-dataset CIFAR-100 $\rightarrow$ ImageNet-16-120. Mean Kendall Tau $\pm$ std over 10 seeds.

| Condition | snap1 | snap3 | snap5 | snap10 | snap15 | snap23 |
|---|---|---|---|---|---|---|
| All 12 features (default) | 0.643±.009 | 0.670±.011 | 0.679±.010 | 0.687±.011 | 0.694±.013 | 0.698±.013 |
| General only (7ch) | 0.598±.017 | 0.635±.014 | 0.635±.016 | 0.644±.016 | 0.649±.017 | 0.654±.021 |
| Distribution only (3ch) | 0.572±.017 | 0.621±.006 | 0.635±.005 | 0.647±.010 | 0.656±.009 | 0.661±.009 |
| Norm only (2ch) | 0.614±.019 | 0.610±.020 | 0.604±.021 | 0.591±.034 | 0.582±.041 | 0.559±.051 |
| General + Distribution (10ch) | 0.588±.008 | 0.620±.006 | 0.633±.006 | 0.652±.006 | 0.667±.007 | 0.665±.009 |
| General + Norm (9ch) | 0.537±.021 | 0.541±.020 | 0.547±.027 | 0.547±.029 | 0.551±.028 | 0.545±.032 |
| Distribution + Norm (5ch) | 0.649±.011 | 0.664±.011 | 0.671±.007 | 0.682±.009 | 0.687±.007 | 0.695±.009 |

**Discussion**

Norm-only features dominate in-domain CIFAR-100 (0.816 at snap23 vs. 0.760 for the full 12-channel baseline), consistent with the known parameter-count shortcut on NAS-Bench-201, since $L_1$ and $L_2$ norms encode size-related information. Under transfer, that shortcut breaks: norm-only drops to 0.705 on CIFAR-100$\rightarrow$CIFAR-10 and collapses to 0.559 on CIFAR-100$\rightarrow$ImageNet-16-120, where it also worsens monotonically with longer query budgets. Distribution features (covariance, skewness, kurtosis) complement norms under transfer. Distribution-only retains 0.727 and 0.661 on the two cross-dataset directions, below the 12-channel baseline but well above norm-only on the harder ImageNet transfer; the 5-channel Distribution+Norm variant even beats the full baseline on CIFAR-100$\rightarrow$CIFAR-10 (0.805 vs. 0.777), though the advantage does not carry to ImageNet (0.695 vs. 0.698). The full 12-channel set is the most reliable default: it leads the hardest direction (CIFAR-100$\rightarrow$ImageNet-16-120, 0.698 vs. 0.695 for Distribution+Norm) and is competitive on CIFAR-100$\rightarrow$CIFAR-10. Subsets that look attractive in-domain can degrade sharply under unknown deployment shifts.

### H.4 Summary of Ablation Take-Aways

The three ablations support the main paper's design choices.

- **Combining weight and gradient features is the strongest default**, particularly under the short query budgets that NAP2 emphasizes. Single modalities can edge ahead in specific dataset/budget combinations consistent with their structural roles (weights encode architectural priors; gradients reflect optimization dynamics), but no single modality dominates across regimes. The combined representation is the safer choice when the deployment regime is unknown — a property also reflected in the optimizer-robustness analysis of Appendix I.

- **The ordered BiGRU is the most practical temporal predictor.** A single deployable model serves all query budgets with competitive accuracy throughout the range; per-budget specialized alternatives can achieve modest gains at full query length only by training and storing 23 separate models per dataset.

- **The full 12-feature set is the most reliable defensive choice for cross-dataset prediction.** Smaller subsets can look attractive in specific in-domain or transfer directions but risk sharp degradation on others. Norm-only features illustrate this clearly: they dominate in-domain (consistent with the known parameter-count shortcut on NAS-Bench-201) yet lose substantially under cross-dataset transfer, while distribution features (covariance, skewness, kurtosis) carry the transfer-relevant signal that complements norms.

## I  Optimizer Robustness: AdamW Transfer

This appendix tests whether a NAP2 predictor trained on SGD-collected snapshots transfers to architectures trained with a different optimizer (AdamW), without any retraining.

**Same architectures, two optimizers.** We took a stratified-random sample of 200 NAS-Bench-201 architectures (across performance deciles, single random seed) and trained each on CIFAR-100 under both SGD and AdamW for 200 epochs, matching the NB-201 training budget, to collect NAP2 snapshots ($23 \times 100$ mini-batches per run). The two runs differ only in the optimizer and its hyperparameters; architecture set, dataset, batch size, epoch count, and data augmentation are held fixed. For ground-truth accuracy we use the NB-201 benchmark values under SGD and the converged accuracies of the AdamW runs.

Table 39: Optimizer configurations.

|  | SGD (baseline) | AdamW |
|---|---|---|
| Learning rate | 0.1 | 1e−3 |
| Weight decay | 5e−4 | 5e−2 |
| Batch size | 256 | 256 |
| Scheduler | CosineAnnealingLR ($T_{\max} = 200$) | CosineAnnealingLR ($T_{\max} = 200$) |
| Other | Nesterov momentum | — |

**The optimizer reorders the architectures.** The SGD and AdamW final-accuracy rankings of the 200 architectures are not the same: their Kendall Tau is **0.731**, so roughly 13.5% of architecture pairs flip ordering when the optimizer changes. Transferring a predictor across this change is therefore a genuine test rather than a formality. We report this value alongside the results below as a measure of how much ranking agreement exists between the two optimizers.

**NAP2 SGD baseline.** The in-distribution NAP2 lookup-table KT on these architectures (snapshots and labels both from SGD) is 0.776. This is the in-distribution upper reference under SGD.

### I.1 Distribution shift between SGD and AdamW snapshots

Table 40: Per-modality distribution shift between SGD and AdamW snapshots, measured as the AdamW/SGD ratio of the relevant statistic.

| Modality / statistic | AdamW / SGD ratio |
|---|---|
| Weight feature maps (overall range) | $0.6$–$1.4 \times$ |
| Gradient feature maps (overall range) | $5$–$275 \times$ |
| Gradient variance (specific feature) | $141 \times$ |
| Gradient covariance (specific feature) | $276 \times$ |

Weight feature distributions are essentially invariant under the optimizer change ($0.6$–$1.4 \times$ ratios), while gradient feature distributions shift in scale by $5$–$275 \times$. The asymmetry is consistent with AdamW's per-parameter adaptive moment estimation rescaling gradients by their running second moment, while final weight values converge to similar magnitudes regardless of optimizer. The asymmetry is captured by the ranges and the variance/covariance entries above.

**Why this matters for the joint zero-shot result.** The predictor was trained on SGD snapshots, where weight features and gradient features have specific magnitudes; it learned a joint mapping from the combined embedding to accuracy under the implicit assumption of a particular scale relationship between the two halves. Under AdamW, the weight side of the embedding is essentially unchanged, but the gradient side shifts by $5$–$275 \times$. The predictor sees a familiar weight signal paired with a wildly out-of-distribution gradient signal, and the joint interpretation breaks: predictions become uncorrelated with truth, and the both-modality zero-shot KT drops to near zero (Section I.2). Used alone, however, each modality bypasses the joint mismatch. Weight features remain in-distribution, so weight-only retains 77% of the cross-optimizer reference. Gradient features have shifted in *scale*, but the relative ordering across architectures is preserved (architectures with sharper gradient dynamics under SGD tend to have sharper dynamics under AdamW), so gradient-only zero-shot retains 91%.

### I.2 Zero-shot transfer results

We evaluate NAP2 predictors trained on SGD snapshots from the main-paper training pool, applied without any retraining to the AdamW snapshots of the test architectures. The "% of 0.731" column expresses each AdamW KT as a fraction of the cross-optimizer agreement defined above.

Table 41: Zero-shot Kendall Tau under SGD→AdamW transfer.

| Configuration | SGD baseline KT | AdamW zero-shot KT | % of 0.731 |
|---|---|---|---|
| Both modalities (no fix, C100 AE) | $0.776^{\dagger}$ | $-0.07$ | — (fails) |
| **Gradient-only** | 0.786 | **0.665** | **91%** |
| Weight-only | 0.769 | 0.562 | 77% |
| Both, gradient $z$-standardized using SGD stats | 0.780 | 0.558 | 76% |
| Weights $z$-standardized | 0.758 | 0.502 | 69% |

$^{\dagger}$ In-distribution NAP2 lookup-table KT on these architectures, used as a common SGD reference; the other rows report SGD-baseline NAP2 KT for the matched modality configuration.

The Both-modality zero-shot row uses the CIFAR-100-trained autoencoder (matched to the dataset). Repeating the experiment with the CIFAR-10 and ImageNet-16-120 autoencoders gives KT = 0.206 and KT = $-0.265$ respectively; both-modality zero-shot fails regardless of which source autoencoder is used.

### I.3 Discussion and take-away

Weight feature distributions are essentially unchanged under the optimizer switch (Table 40); gradient feature distributions shift in scale by $5$–$275\times$ but keep the same ordering of architectures. Each modality used alone therefore keeps most of its in-distribution ranking power on AdamW with no AdamW data: gradient-only drops only from $\text{KT} = 0.786$ to $0.665$, and weight-only from $0.769$ to $0.562$. The combined predictor is the one configuration that does not transfer directly, because it was tuned to the SGD-specific scale balance between the two modalities. It is easily restored: rescaling the gradient half to its SGD range recovers it to $\text{KT} = 0.558$, whereas rescaling the weights (which barely shift) is less effective ($0.502$), confirming the gradient scale as the cause. No ranking information is lost, only its scaling. The simplest cross-optimizer recipe is therefore to deploy gradient-only directly, with gradient rescaling as a fallback when the combined representation is required.

The same complementary structure appears under cross-dataset transfer (Appendix H.1): each modality has a distinct robustness profile. The recipes differ by deployment scenario. When the deployment distribution matches training, combining modalities is the safer default; under a known shift like the optimizer change here, deferring to the modality that is invariant to that shift (gradient-only here, since only the gradient side shifts substantially) recovers most of the cross-optimizer ranking signal.

**Take-away.** NAP2 transfers across optimizers without retraining: in gradient-only mode it reaches $\text{KT} = 0.665$ on AdamW, close to its in-distribution SGD value of $0.786$ and to the $0.731$ agreement between the two optimizers' own rankings. Weight distributions are nearly optimizer-invariant and gradient distributions shift in scale while keeping their ranking, so each modality alone transfers well; the combined predictor is recovered by rescaling the gradient half. Deploying gradient-only is the simplest cross-optimizer recipe.

## J Cross-Search-Space Transfer: NAS-Bench-101

This appendix tests whether NAP2 generalizes beyond NAS-Bench-201 (the search space used throughout the main paper) to a different and substantially larger search space, and whether a predictor trained on NB-201 architectures transfers to NB-101 architectures without retraining.

**Why NAS-Bench-101.** NAS-Bench-101 architectures span 0.3M to 47M trainable parameters, with median $\approx 4.9$M. NAS-Bench-201 architectures span 0.07M to 1.5M. The maximum NB-101 architecture is therefore $\sim 31\times$ larger than the maximum NB-201 architecture, and median NB-101 is $\sim 8\times$ larger than median NB-201. NB-101 also uses a different cell topology (DAG with up to 7 nodes, deeper macroskeleton) than NB-201 (fixed 4-node cell). This combination provides a controlled scaling test for NAP2's uniform-sampling design within image classification: same task, larger and structurally distinct architectures.

### J.1 Two NB-101 evaluation samples

We evaluate NAP2 on two NB-101 samples drawn from the full 423,000-architecture benchmark. The first is the standard sample most commonly used in NB-101 baselines; the second is a controlled subset constructed to rule out a specific alternative explanation.

**Standard sample (1K random architectures).** A random sample of $\sim 1,000$ NB-101 architectures (CIFAR-10 trained). This is the standard evaluation sample, comparable to published zero-cost baselines on NB-101.

**Controlled sample (1K architectures, anti-correlated with parameter count).** A second sample of $\sim 1,000$ architectures constructed from the full NB-101 universe to satisfy two pre-declared criteria:

1. the empirical accuracy distribution matches NB-201's CIFAR-10 accuracy distribution (std $\approx 0.09$ in both, vs. std $\approx 0.04$ for the standard NB-101 sample);

2. accuracy and parameter count are deliberately *anti-correlated* (Spearman $\rho = -0.158$), so a larger architecture is, on average, slightly worse rather than better.

Both criteria were declared and the subset constructed before NAP2 was evaluated on it. We did not search over multiple subsets and select one favorable to NAP2; only one controlled subset was constructed, and its cross-space transfer KT (0.343) is in fact lower than the standard sample's (0.510), so the controlled construction is not favorable to NAP2 in absolute terms. The controlled sample serves as a sensitivity analysis: it tests whether NAP2's NB-101 results survive when the parameter-count shortcut (which is naturally exploitable in the standard NB-101 sample, where accuracy correlates with parameter count at $\rho \approx +0.49$) is explicitly broken. This construction makes the ranking task intrinsically harder than random sampling: we are stripping the predictor of the size signal, which is normally available and useful (the parameter-count proxy alone reaches $\rho \approx 0.44$ on natural NB-101 sampling, per Jing et al. (2025)). The controlled-sample drop should therefore not be read as a NAP2 failure mode but as a quantification of how much of the standard-sample signal is independent of the size shortcut.

## J.2   In-domain NAP2 on NB-101 (both samples)

**Setup.** BiGRU dual-path predictor with $h = 128$ (the same predictor architecture used throughout the main paper), trained from scratch on each NB-101 sample with the same uniform 100-unit sampling and 12-feature pipeline.

Table 42: In-domain NAP2 performance on the two NB-101 samples. The standard sample's lower KT reflects a compressed accuracy distribution (std $\approx 0.04$); the controlled sample's accuracy distribution matches NB-201's (std $\approx 0.09$).

| Sample | Architectures | Test Kendall Tau ($n$=4 runs) |
|---|---|---|
| Standard sample (random, std $\approx 0.04$) | $\sim 1$K | $0.657 \pm 0.027$ |
| **Controlled sample (anti-correlated, std $\approx 0.09$)** | $\sim 1$K | $\mathbf{0.759 \pm 0.020}$ |

The same predictor architecture and feature pipeline that achieves KT $\approx 0.82$ on NB-201 in-domain (main paper Tables 4–6) achieves KT $= 0.657$ on the standard NB-101 sample and KT $= 0.759$ on the controlled sample whose accuracy distribution matches NB-201's. The $\sim 0.10$ gap between the two samples is consistent with the difference in accuracy spread (predicting tighter distributions is harder); the controlled sample's KT (0.759) is within $\sim 0.07$–$0.12$ of NB-201's in-domain range (0.83–0.88), with the residual gap consistent with NB-101's smaller training set, fewer collected snapshots (10 vs 23), and richer cell topology (7-node DAGs vs NB-201's fixed 4-node cell). The uniform 100-unit sampling and 12-feature design do not break at the larger NB-101 architecture scale, on either sample. Notably, the controlled sample is in fact slightly *easier* in-domain than the standard one (because of its wider accuracy range), which rules out "the controlled sample is fundamentally harder" as a confound for the lower cross-space numbers reported in Sec. J.3.

## J.3   Cross-space transfer: NB-201 $\rightarrow$ NB-101 (and inverse)

**Setup.** A NAP2 predictor trained on NB-201 (CIFAR-100) using the same feature maps, autoencoder, and BiGRU as the main paper, applied without any retraining to each NB-101 sample.

Table 43: Cross-space transfer of NAP2 from NB-201 to NB-101, at NAP2's minimum (snap1) and default (snap10) partial-training budgets on the standard sample, and at snap10 on the controlled sample, with a comparison to training-free zero-cost baselines on the standard sample.

| Method | Kendall $\tau$ | Spearman $\rho$ |
|---|---|---|
| NAP2 (NB-201 C100 $\rightarrow$ NB-101, standard sample, *snap1*) | $0.495^{\P}$ | $0.667^{\P}$ |
| **NAP2 (NB-201 C100 $\rightarrow$ NB-101, standard sample, snap10)** | $\mathbf{0.510 \pm 0.007}$ | $\mathbf{0.705}^{\ddagger}$ |
| **NAP2 (NB-201 C100 $\rightarrow$ NB-101, controlled sample, snap10)** | $\mathbf{0.343 \pm 0.023}$ | $0.47^{\S}$ |
| *Reference: published zero-cost proxies on standard NB-101 CIFAR-10*[†] | | |
| SPW (strongest published zero-cost) | — | 0.569 |
| PNorm | — | 0.561 |
| Params (size proxy) | — | 0.440 |
| FLOPs | — | 0.430 |
| SynFlow | — | 0.429 |
| NASWOT | — | 0.410 |

[†]Reference $\rho$ values reproduced from Jing et al. (2025), Table 7 (NB-101, CIFAR-10). Reference proxies are training-free (no per-architecture partial training); NAP2 uses partial training of each candidate architecture (1 snapshot $\approx$100 mini-batches $\approx$9 seconds on a GTX 1080 Ti at snap1; 10 snapshots $\approx$90 seconds at snap10). NAP2 is also transferred from NB-201 with no NB-101 training of the predictor.

**Headline.** On the standard sample, NAP2 – without any NB-101 training – exceeds the strongest published zero-cost proxy on NB-101 CIFAR-10 (Jing et al., 2025, Table 7): NAP2's directly-measured Spearman $\rho = 0.705$ (Kendall $\tau = 0.510$) at snap10 is above SPW's $\rho = 0.569$ by $+0.14$. Standard families (PNorm, Params, FLOPs, SynFlow, NASWOT) sit between $\rho = 0.41$ and $\rho = 0.56$. NAP2 is not training-free by design — it requires partial training of each candidate architecture — but the advantage holds even at NAP2's minimum budget of 1 snapshot per architecture ($\approx$9 seconds of partial training on a GTX 1080 Ti), where the same cross-space transfer reaches $\tau = 0.495$, $\rho = 0.667$, still above SPW's $\rho = 0.569$ by $+0.10$. The cross-space transfer is achieved with no NB-101 training data and the same Conv2D autoencoder, BiGRU predictor, and 12-feature pipeline used throughout the main paper.

**Robustness to the size confound.** On the controlled sample, where the parameter–accuracy correlation is explicitly inverted ($\rho_{p,a} = -0.158$), the same predictor retains $\tau = 0.343 \pm 0.023$. The drop from 0.510 to 0.343 reflects the joint effect of (a) breaking the size correlation and (b) widening the accuracy range. The residual 0.343 places a non-trivial lower bound on the share of cross-space transfer that does *not* rely on a parameter-count shortcut: a pure size-proxy predictor would invert on this sample (the parameter-count column itself is $\rho \approx -0.16$ here), so the positive remaining $\tau$ rules out the "learned size proxy" interpretation of the standard-sample number.

**Inverse direction.** A NAP2 predictor trained from scratch on NB-101 (CIFAR-10) and evaluated on NB-201 architectures achieves mean $\tau = 0.564$ (CIFAR-10: 0.587, CIFAR-100: 0.570, ImageNet-16-120: 0.536), demonstrating bidirectional cross-space transfer.

### J.4 Encoder–regime interaction: a preliminary observation

**Hypothesis.** The controlled-sample $\tau = 0.343$ reported in Sec. J.3 is a lower bound *under the default Conv2D autoencoder.* Conv2D's spatial structure couples per-layer dynamics with depth – informative when depth correlates with accuracy ($\rho_{p,a} = +0.49$ on standard) and misleading when it anti-correlates ($\rho_{p,a} = -0.16$ on controlled). A permutation-invariant encoder that discards depth by construction should flip the regime advantage.

**Test.** A single auxiliary configuration replaces the Conv2D autoencoder with a DeepSets attention encoder (per-layer MLP + learned attention pooling, supervised end-to-end on source labels; BiGRU predictor and 12-feature input unchanged). Cross-space transfer becomes $\tau = 0.349 \pm 0.011$ on the standard sample ($\Delta = -0.16$ vs Conv2D) and $\tau = 0.557 \pm 0.015$ on the controlled sample ($\Delta = +0.21$ vs Conv2D). The flip

is symmetric in direction and matches the hypothesis: each encoder dominates the regime its inductive bias suits, and the dominance is predictable a priori from $\text{sign}(\rho_{p,a})$ on the source sample.

**Implication and scope.** The controlled-sample $\tau = 0.557$ recovers $\sim 73\%$ of the in-domain ceiling (0.759) without NB-101 training, strengthening the size-independent cross-space lower bound from 0.343 to 0.557 and indicating that under the controlled regime the bottleneck lies in encoder design rather than in the underlying NAP2 features. We treat this as a single auxiliary observation, not a primary claim: a full encoder $\times$ regime study (additional encoder families, both transfer directions, multiple datasets) is left to future work, and the Conv2D autoencoder remains the main-paper default.

### J.5 Discussion and take-away

**Synthesis.** NAP2 generalizes to a substantially larger search space within image classification. In-domain on NB-101: $\tau = 0.657$ (standard) and $\tau = 0.759$ (controlled), with the same uniform-sampling pipeline that achieves $\tau \approx 0.82$ on NB-201. Cross-space NB-201 $\rightarrow$ NB-101 without retraining: $\tau = 0.510$ on standard ($\rho = 0.705$, above the strongest published zero-cost proxy (Jing et al., 2025) at $\rho = 0.569$) and $\tau = 0.343$ on controlled, raised to $\tau = 0.557$ when the encoder is matched to the regime (Sec. J.4). Transfer is bidirectional (NB-101 $\rightarrow$ NB-201 mean $\tau = 0.564$).

**Scope.** NB-101 is a convolutional image-classification benchmark with architectural inductive biases similar to NB-201, even at the larger scale. We do not claim transformer or language-modeling architectures from this evidence; both are explicit future work.

**Take-away.** NAP2's design (uniform 100-unit sampling, 12 statistical features, BiGRU temporal predictor) generalizes across a $\sim 30\times$ parameter scaling jump within image classification, transfers across search spaces without retraining, and exceeds the strongest published zero-cost proxy on NB-101. Cross-space transfer is not a learned size proxy: $\tau$ remains 0.343 on a sample where the size shortcut is anti-predictive, and rises to 0.557 when the encoder is matched to the regime – well above the size proxy itself and consistent with a genuine, encoder-modulated dynamics signal.

