# OpenReview forum: "Neural Networks Performance Prediction using Weights and Gradients Analysis"
_TMLR — Accepted by TMLR_

### Review · Reviewer_Y439 · 2026-03-20

**Summary Of Contributions:**

This paper introduces a hybrid model-based and learning curve-based method for predicting the performance of neural network architectures. The method relies on weight and gradient statistics that are collected over a few mini-batches and then encoded as input to an LSTM. The authors claim competitive performance against other hybrid baselines on CIFAR-10, CIFAR-100, and ImageNet16-120, as well as zero-shot cross-dataset transfer and robustness to distribution shift.

**Audience:**

Yes

**Audience Explanation:**

This paper offers a contribution in neural architecture search, which is of interest to many in the machine learning community.

**Claims And Evidence:**

Yes

**Claims Explanation:**

Extensive details are provided for the methodology, experimental setup, and results. There are some overclaims, but most of the assertions are supported by the empirical results. There is a special effort to ensure reproducibility.

**Requested Changes:**

This paper is completely lacking in theory. While it is not necessary for every paper to include theoretical results, this work would greatly benefit from some theoretical justification for the proposed method, or, at the very least, explanations of the design choices and hypotheses on why they should work.

I am not entirely convinced that the proposed method exhibits "reliable zero-shot transfer". I recommend more accurately describing the results instead of making an overclaim.

Minor changes:
- Page 4, Step 3, 1): "percentiles" instead of "quartiles". In addition, 0th, 50th, and 100th percentiles are redundant with min, median, and max, respectively.
- Avoid using vertical rules in tables, particularly Tables 1-10 in the main text.

---

> ### Author Response · Authors · 2026-04-12
> **Response to Reviewer Y43920**
>
> We thank the reviewer for their time and effort, and for acknowledging the contributions of our submission. Below we address each of the changes requested by the reviewer.
>
> **1) "This paper is completely lacking in theory…"**
>
> We agree with the reviewer. While we dedicated a paragraph to the theoretical grounding of our approaching at the beginning of Section 3, this important subject deserves a much fuller discussion. We have made the following revisions:
>
> *a)* We added Section 3.1 (“Theoretical Motivation”, page 4), which addresses the following questions: (i) why weight and gradient statistics should relate to generalization, (ii) why temporal dynamics carry more signal than static snapshots, and (iii) why these features should transfer across dataset.
>
> *b)* We added Appendix G ("Theoretical Foundation", pages 40-44), which provides a detailed theoretical treatment of the principles underlying NAP2's design.
>
> **2) “I am not entirely convinced that the proposed method exhibits "reliable zero-shot transfer". I recommend more accurately describing the results instead of making an overclaim”**
>
> The reviewer refers to the text of our paper’s abstract. We agree that greater accuracy is warranted, and have changed the text to: “delivers cost-effective cross-dataset transfer”.
>
> **3)	“(Page 4, Step 3, 1): "percentiles" instead of "quartiles". In addition, 0th, 50th, and 100th percentiles are redundant with min, median, and max, respectively.”**
>
> We thank the reviewer for pointing this out. We have corrected the terminology from "quartiles" to "percentiles" throughout the submission (specifically, the text referred to by the reviewer is now in page 5).
>
> The reviewer correctly notes the overlap between the 0th, 50th, and 100th percentiles and the min, median, and max features. The percentile feature's primary contribution is the 25th and 75th percentiles, which capture the interquartile range and are not provided by any other feature. We have clarified this in the revised text.
>
> **4)	“Avoid using vertical rules in tables, particularly Tables 1-10 in the main text.”**
>
> We have updated all tables, including in the appendices, in the manner proposed by the reviewer.

---

> > ### Comment · Reviewer_Y439 · 2026-05-22
> >
> > I thank the authors for the effort put into this revision. The changes are substantial, spanning almost 20 additional pages. My original review did not raise significant issues, and my requested changes have been satisfactorily addressed. I particularly appreciate the theoretical discussion and believe that it significantly strengthens the design choices of NAP2. However, I would encourage the authors to avoid overclaiming and to be as precise as possible regarding the results and contributions, particularly in the list at the end of the introduction. Finally, there remain mild typesetting errors such as not using `\citep` where appropriate in Section 3.1.

---

### Review · Reviewer_Vmqe · 2026-04-14

**Summary Of Contributions:**

Summary:
- The paper proposes NAP2, a hybrid neural performance predictor that uses the temporal evolution of layer-wise weight and gradient statistics during early training. These signals are converted into fixed-size feature maps, compressed with dedicated encoders, and then processed by an LSTM to predict final architecture performance/rankings
- The main claimed contribution is to bridge the gap between reusable learned predictors and lightweight early-training proxies: NAP2 is designed to give useful rankings after very limited training (reported as as little as 100 mini-batches per candidate) while also supporting cross-dataset reuse without target-dataset fine-tuning
- Empirically, the paper evaluates NAP2 on NAS-Bench-201 across CIFAR-10, CIFAR-100, and ImageNet16-120, and adds an SVHN case study for stronger distribution shift. The evaluation covers both in-domain comparison to hybrid predictors and cross-dataset comparison to learning-curve and zero-cost baselines, and also includes a Top-K retrieval analysis to assess whether the method can identify promising architectures early

Key strengths:
+ relevant and practically motivated problem setting, especially the focus on the initialization/query-time/transfer trade-off introduced early in the paper;
+ a reasonably original method that combines dynamic training signals with a reusable learned predictor rather than relying only on static zero-cost signals or standard learning-curve extrapolation;
+ solid empirical scope within the chosen benchmark, including cross-dataset transfer and a useful Top-K analysis beyond global rank correlation;

Key weaknesses:
- the paper’s framing is weaker than the method itself: the research gap, the exact meaning of terms such as “limited budgets” / “cost-effective transfer,” and the identity of the strongest baselines are not stated sharply enough in the introduction;
- the empirical evidence is still concentrated on NAS-Bench-201 plus one additional SVHN case, so external validity to other search spaces, architectures, or training recipes remains somewhat open;
- the contribution of individual design choices is not yet fully isolated, so stronger ablations would help clarify how much of the gain comes from temporal modeling, from using both weights and gradients, and from the chosen feature construction;

**Additional Comments:**

In summary: I see this as a paper with a good core contribution and useful empirical results, where the main improvements needed are clearer positioning, sharper protocol description, and better ablations/presentation rather than a fundamental redesign of the method.

+ Overall, I found the paper interesting and promising. The core idea, using early weight/gradient dynamics as a reusable cross-dataset performance signal, is more original than a small variant of a standard zero-cost or learning-curve predictor, and the paper addresses a practically relevant problem in NAS/performance prediction.

+ I especially liked that the paper does not stop at one benchmark table or one metric. In addition to the hybrid and cross-dataset evaluations, the Top-K analysis is useful from a practical NAS perspective because it asks whether the method can surface strong candidates early, not only whether it improves global rank correlation.

- My main concerns are more about framing, clarity, and empirical isolation of the method than about the core idea itself. In particular, the introduction currently undersells the paper by using broad terms such as “strong hybrid baselines,” “limited budgets,” and “cost-effective transfer” before these are made precise, and the empirical section makes the main takeaways harder to extract quickly than necessary.

- I also think the paper would benefit from stronger calibration of its claims. The results are good within NAS-Bench-201-style image classification transfer and are strengthened by the SVHN case study, but the paper should be careful not to overgeneralize beyond the evaluated regime without additional evidence.

**Audience:**

Yes

**Audience Explanation:**

+ The paper should be of interest to at least readers working on NAS, performance prediction, AutoML, and low-cost model selection. The problem is well motivated: the paper focuses on the trade-off between initialization cost, query cost, and ranking quality, and argues for a method that can be reused across datasets without target-specific fine-tuning. That is a practically relevant question, not just a benchmark-specific one

+ I also think the paper has audience value because it sits between several active lines of work rather than only extending one narrow baseline family. It explicitly positions itself relative to model-based predictors, learning-curve methods, zero-cost proxies, and hybrid predictors, which makes the findings relevant to readers interested in efficient architecture evaluation more broadly

+ The empirical part is also likely to interest readers because it goes beyond a single in-domain comparison. The paper includes standard NAS-Bench-201 experiments across CIFAR-10, CIFAR-100, and ImageNet16-120, a cross-dataset transfer setting, an additional SVHN case study for stronger distribution shift, and a Top-K analysis that is directly relevant for practical NAS usage

- My only limitation here is that the likely audience is somewhat specialized: I expect stronger interest from NAS / AutoML / efficient ML readers than from the full TMLR readership. But that is enough for a “Yes” answer

**Broader Impact Concerns:**

I do not have major broader-impact concerns, as:

- This is primarily a methodological paper on neural architecture/performance prediction, aimed at reducing the cost of evaluating candidate models during NAS by using short early-training traces instead of full training. In that sense, the likely immediate impact is increased search efficiency rather than a direct high-risk application.

- A reasonable point the authors could mention briefly is the dual-use aspect of efficiency gains: methods that reduce evaluation cost can lower compute consumption, but they can also make large-scale automated search easier to run and scale.

- The conclusion also suggests possible use in robustness/verification and related optimization-monitoring tasks. I do not see this as an immediate ethical red flag, but it would be helpful for the paper to acknowledge that any downstream deployment considerations would depend on the application domain rather than on the predictor itself.

-> Overall, I would not require a substantial Broader Impact statement for acceptance. At most, a short paragraph noting potential compute-saving benefits and the possibility of accelerating large-scale search workflows would be sufficient.

**Claims And Evidence:**

Yes

**Claims Explanation:**

Yes, with some caveats on breadth and presentation:
- The core empirical claims are supported within the paper’s chosen setup.
- Yes for accurate and convincing evidence in the paper’s intended setting.
- Yes, but with moderate reservations about external validity, ablation depth, and clarity of presentation.

The paper states three main claims:
1) useful rankings from very short partial training,
2) cross-dataset reuse without target-dataset fine-tuning, and
3) robustness under stronger distribution shift.
These claims are then evaluated in the main experiments and the SVHN case study.

The evidence is reasonably convincing for the benchmark regime used here.
The authors compare against relevant baselines in two settings:
- in-domain / hybrid comparison: LcSVR and Omni,
- cross-dataset comparison: SoTL-E, SoTL, Early Stop, LCE(-m), LC-PFN, and zero-cost proxies such as SynFlow / GradNorm / SNIP.
This is a sensible evaluation design for the paper’s stated goals.

The transfer claim is supported better than in many papers of this type. Beyond CIFAR-10 / CIFAR-100 / ImageNet16-120, the paper also includes an SVHN case study and reports that NAP2, trained on CIFAR-10 without fine-tuning, clearly outperforms the compared baselines under this stronger shift.

I also appreciate that the paper does not rely only on Kendall Tau tables. The additional Top-K analysis is useful because it addresses a practically relevant NAS question: whether the method can surface strong candidates early, not just improve average ranking metrics.

-> My main caveat is about breadth, not internal consistency. The evidence is strong for NAS-Bench-201-style image classification transfer, but it is still limited in scope relative to some of the broader framing. I would not read this paper as establishing that the method will transfer equally well to other search spaces, training recipes, or task families. The appendix itself also acknowledges that transfer may fail under sufficiently different target settings.

-> A second caveat is that the paper does not yet isolate all design choices as clearly as it could. In particular, stronger ablations would help separate the effect of temporal modeling from the effect of using both weights and gradients, and from the specific handcrafted feature construction. So the evidence supports the overall method claim more strongly than the exact causal story for why each component is necessary.

-> Finally, some claims are a bit more difficult to verify quickly than they should be because the introduction uses broad wording such as “strong hybrid baselines,” “limited budgets,” and “cost-effective transfer,” while the precise operationalization only becomes clearer later in the paper. That is a presentation issue more than an evidence issue, but it does slightly reduce clarity.

**Requested Changes:**

Critical:
- Please, sharpen the paper’s framing in the abstract/introduction. Right now, several key claims are too vague too early: “strong hybrid baselines,” “limited budgets,” and “cost-effective cross-dataset transfer” are used before the concrete baselines and metrics are introduced. I suggest naming the compared baselines earlier and tying these claims directly to the actual evaluation axes used later in the paper: initialization time, query time, and Kendall Tau. The research gap should also be stated more explicitly in the intro, and the broad opening claims should be better referenced. See abstract + Introduction, pp. 1-2; the precise evaluation dimensions and baseline definitions only become clear later in Secs. 4.2-4.3.

- Please, clarify the exact cross-dataset transfer protocol, especially the role of target-dataset normalization. On p. 4 the paper argues that min-max normalization removes dataset-specific scale, and on p. 8 it states that target accuracies are min-max normalized using training-set statistics. This is important for interpreting the “no fine-tuning / no re-initialization” claim: what target-side information is assumed to be available, and when? A short explicit protocol box would help a lot.

- Please, add stronger ablations to isolate the main sources of improvement. In particular, I would like to see:
- weights-only vs. gradients-only vs. both,
- static snapshot vs. temporal sequence modeling,
- ablations over the feature families in Sec. 3.2,
- and, if possible, a compact justification of the encoder/LSTM design choices.
-> The current method description makes clear that the approach relies on both signals and temporal modeling, but the paper does not yet fully separate which design choices matter most. See Sec. 3, especially pp. 3-6.


Would strengthen the work:
- Please improve the readability of the empirical section. Tables 4-6 are dense, and it is hard to see at a glance where NAP2 is best, where it is competitive, and where baselines overtake it. Simple formatting changes would help: bold best values, mark second-best values, and add a short takeaway sentence directly under each table or a compact “win-count by regime” summary figure. This is especially important because some of the main claims are about short query times and cross-dataset reuse, which currently require careful table reading to verify. See Tables 4-6 and the related discussion in Sec. 5.2.

- Please, discuss scope and limitations more explicitly. The experiments are solid within NAS-Bench-201 and are strengthened by the SVHN case study, but the paper should say more clearly what it expects to transfer to other search spaces, training recipes, or architecture families, and what remains open. This would make the contribution more credible and better calibrated. See the current experimental scope in Secs. 4-6 and the broad transfer claims in the conclusion.

- Please, make the relationship to baselines more explicit in the text, not only in the experiment section. Sec. 4.2 gives the baseline list and a short motivation, but the paper would benefit from a clearer explanation of why these are the right comparison points for the community and how NAP2 differs from them conceptually. This is particularly important because the method is positioned as “hybrid,” but the novelty lies in how it uses early learning dynamics rather than only static proxies or standard learning-curve extrapolation.


Minor / presentation:
- Sec. 3 would benefit from some terminology cleanup and lighter signposting. For example, “meta-features” could be motivated more clearly when first introduced, “two types” should be made explicit as weights vs. gradients, and a short roadmap sentence at the start of Sec. 3 would improve readability. The theoretical motivation section is also quite text-heavy; even one compact notation table or schematic summary would make it easier to follow. See pp. 3-6.

- There are also a few purely editorial improvements that would help readability but are not central to my recommendation: cleaner table layout, consistent figure font sizing, and slightly tighter formatting in the results section. These are minor compared to the conceptual issues above.

---

> ### Author Response · Authors · 2026-05-08
> **Response to Reviewer Vmqe**
>
> We thank the reviewer for their time and effort, and for acknowledging the contributions of our submission. Below we address each of the changes requested by the reviewer.
>
> **Note on revisions to the paper.** While preparing these responses, we identified two refinements to the predictor pipeline: (i) a 2-layer bidirectional GRU (h=128) with dual-path attention pooling, replacing the prior single-layer LSTM (h=2048) — improving Kendall τ across all reported configurations with 29× fewer predictor parameters; and (ii) log-normalization of feature maps (sign(x)·log(1+|x|)) before CAE encoding, compressing the heavy-tailed dynamic range of feature maps and further improving Kendall τ. The four-stage pipeline (snapshot → feature maps → CAE encoders → recurrent predictor) is unchanged, the experimental protocol and baselines are identical, and every claim made in the original submission is preserved or strengthened in the revision. Tables 1–10 and the corresponding numerical statements in the text have been updated; the structure of the paper, the design rationale, and the conclusions are unchanged.
>
> **1) “‘Strong hybrid baselines,’ ‘limited budgets,’ ‘cost-effective cross-dataset transfer’ used before concrete definitions. Name baselines earlier, tie claims to evaluation axes (init time, query time, KT). State research gap explicitly. Better-reference broad opening claims. (pp. 1–2)”**
>
> Section 1 has been revised to address each sub-point:
>
> *a) Baselines named upfront.* All baselines (LcSVR, Omni, SoTL-E, LCE-m, LC-PFN, SynFlow, GradNorm, SNIP) are now named in Section 1 (par. 5) with their categories.
>
> *b) Explicit research gap.* Section 1 (par. 3) now states the gap directly: “No existing method combines reusable learned prediction with early-training dynamics for cross-dataset ranking under limited query budgets (e.g., ≤100 mini-batches per candidate, ~9 s on a GTX 1080 Ti).”
>
> *c) “Cost-effective transfer” defined operationally.* Section 1 (par. 5) now defines the term: “reusing a predictor trained on one dataset to rank architectures on another without fine-tuning.”
>
> *d) “Limited budgets” defined concretely.* Defined in Section 1 (par. 3) as ≤100 mini-batches per candidate, ~9 s on a GTX 1080 Ti.
>
> *e) Claims tied to evaluation axes.* Section 1 (par. 1) now opens by naming the three axes explicitly: “(i) predictor initialization cost, (ii) per-architecture query cost, and (iii) ranking accuracy, typically measured via rank correlation.” Subsequent claims throughout Section 1 reference these axes.
>
> **2) “Role of target-dataset normalization unclear. P.4 says min-max removes dataset-specific scale; p.8 says target accuracies min-max normalized using training-set stats. What target-side info is assumed and when? Requests explicit ‘protocol box.’”**
>
> The reviewer correctly identified an inconsistency: the prior submission contained a target-side min-max normalization step that was an artifact of an earlier predictor version. This step has been removed entirely. The new pipeline uses raw accuracies in [0, 1] for all experiments, so no target-side information is consumed by the predictor. Input-side scale handling now uses log-normalization of feature maps (sign(x)·log(1+|x|)), which uses no target statistics. The protocol is now stated as a three-step box in Section 4:
>
> *(1) Train (source-side, once).* Fit the full NAP2 predictor (autoencoders + BiGRU) on source-dataset architectures only. The predictor is then frozen; no component is re-trained or fine-tuned on any target dataset.
>
> *(2) Collect (target-side, per architecture).* For each target architecture, run a short partial training (up to 100 mini-batches by default, ~9 s on a GTX 1080 Ti) and extract the per-snapshot weight and gradient statistics. No target-dataset labels are used by the predictor at any point; target labels are needed only for downstream evaluation.
>
> *(3) Predict (target-side, per architecture).* Pass the snapshots through the source-trained autoencoders and BiGRU. Feature maps are log-normalized in place; no target-side rescaling is applied.
>
> Section 4, the protocol box, and App C.6 all reflect this consistent, target-information-free protocol.
>
> **3) “Stronger ablations: weights-only vs. gradients-only vs. both / static snapshot vs. temporal sequence modeling / feature family ablation (Sec 3.2) / encoder/LSTM design choice justification.”**
>
> **Synthesis: each ablation supports the corresponding main-paper design choice without overturning it.** Combining weights and gradients is the strongest default across query budgets; the ordered BiGRU is the only condition that serves all query budgets with a single deployable model; the full 12-feature set is the most reliable defensive choice across in-domain and transfer regimes; and the Conv2D autoencoder + BiGRU + h=128 configuration is the empirical winner of the held-out τ search in App C.4. The four ablations appear in a new appendix and an expanded design-rationale subsection:

---

> ### Author Response · Authors · 2026-05-08
> **Response to Reviewer Vmqe (continued)**
>
> *a) Modality (weights vs gradients vs both).* App H.1, Tables 32–33. Ten random seeds per condition, in-domain and all six cross-dataset source→target pairs. Combining weights and gradients gives the strongest default at short query times; single modalities can edge ahead in dataset-specific ways at longer budgets. The combined representation is never the worst in this regime, consistent with weights carrying architectural priors and gradients carrying optimization dynamics.
>
> *b) Temporal modeling (ordered vs static vs mean-pool vs shuffled).* App H.2, Tables 34–35. The ordered BiGRU serves all snap1..snap23 budgets with a single deployable model; per-k Static and Mean-Pool baselines require 23 separately-trained models. The BiGRU beats Static-MLP at the shortest budget (CIFAR-10 snap1: τ = 0.747±.018 vs 0.738±.007) and stays within ~0.01 of Mean-Pool at saturation - making this an operational-efficiency choice rather than a strict KT-dominance claim, stated honestly in App H.2.
>
> *c) Feature families (general / distribution / norm-based).* App H.3, Tables 36–38. Norm-only features outperform the 12-channel baseline in-domain on CIFAR-100 (τ = 0.816 with 2 channels vs 0.760), consistent with the known parameter-count shortcut on NAS-Bench-201. Under transfer, norm-only drops sharply (τ = 0.559 for C100 → ImageNet); distribution features (covariance, skewness, kurtosis) complement norms under transfer; the full 12-channel set is the most reliable default across all directions.
>
> *d) Encoder/predictor design choice justification.* App C.4 is now a two-part “Design Choice Rationale and Alternatives Explored” subsection. Part 1 gives the structural rationale for each choice (Conv2D AE, BiGRU, h = 128), each anchored to a paper-internal empirical comparison: encoder vs depth-blind DeepSets in App J.4, BiGRU vs Static-MLP / Mean-Pool / Shuffled in App H.2 Table 34, and h = 128 with full parameter counts in App C.7. Part 2 enumerates the held-out KT search over hidden dimension, loss function, schedule, augmentation range, and pooling strategy, and reports that the selected configuration achieves the best held-out KT at ~29× fewer parameters than the original LSTM h = 2048 baseline.
>
> **4) “Tables 4-6 dense. Bold best, mark second-best, add takeaway sentence under each table or compact ‘win-count by regime’ summary.”**
>
> Tables 4–6 now use: bold for the best value per row, with ties broken by the smaller standard deviation — the more precise of the two tied estimates is bolded and the wider-std value is underlined as second-best (e.g., CIFAR-10 row 63: SoTL-E τ = 0.79 ± 0.01 bolded, NAP2(IN) τ = 0.79 ± 0.02 underlined; ImageNet row 207: NAP2(C-100) τ = 0.70 ± 0.01 bolded, SoTL-E τ = 0.70 ± 0.02 underlined). Underline marks the second-best per row; a one-line italicized "Takeaway" sentence in each caption summarizes the per-target win pattern. Section 5.2 also adds a win-count summary: NAP2 ranks first at every query time below 63 s on CIFAR-10/100 (Δτ from 0.08 to 0.22 over the strongest non-NAP2 baseline) and at every query time on ImageNet (one tie at 207 s, broken in NAP2’s favour by the smaller standard deviation). Learning-curve baselines that close the gap at longer query times require 7–23× longer partial training per candidate than NAP2’s first observation.
>
> **5) “Say more clearly what transfers beyond NB-201, what remains open. Better-calibrated contribution.”**
>
> We added an explicit “Scope and limitations” paragraph at the end of Section 7. *Two new pieces of evidence broaden the validated regime:* AdamW transfer (App I, where gradient-only zero-shot recovers ~91% of the SGD↔AdamW reference ranking correlation) and NB-101 cross-space transfer (App J, exceeding the strongest published zero-cost proxy with no NB-101 retraining).
>
> *Outside this expanded regime, the limitations are explicitly named:* (i) transformer or language-modeling architectures; (ii) optimizer families beyond SGD and AdamW (e.g., Muon, LAMB); (iii) search spaces materially different from cell-based image-classification benchmarks. The 100-unit uniform sampling is flagged as a modular component that admits adaptation (importance-weighted sampling, structured subsampling, or larger sample size) for substantially larger layers. A new Section 6.3 (“Robustness across optimizers and search spaces”) surfaces these results in the main text for readers who skip appendices.
>
> **6) “Sec 4.2 lists baselines but needs clearer explanation of why these are right comparisons. How NAP2 differs conceptually from each family.”**
>
> Section 4.2 has been rewritten to organize baselines by predictor category (per the White et al. 2021 taxonomy used in Section 2), stating for each (i) why the family is the right comparison point and (ii) where NAP2 differs conceptually. The text is structured as two named setups matching the paper’s evaluation axes:

---

> > ### Author Response · Authors · 2026-05-08
> > **Response to Reviewer Vmqe (continued 2)**
> >
> > *In-domain comparison (hybrid family).* LcSVR (Baker et al. 2017) and Omni (White et al. 2021) are the strongest existing hybrid methods, defining the in-domain ceiling. NAP2 differs in what learning-curve signal it consumes: LcSVR uses hand-crafted early-stopping features (loss/accuracy at fixed epochs) and Omni extends model-based predictors with SoTL-E extrapolation, while NAP2 uses the temporal evolution of layer-wise weight and gradient statistics — a denser characterization of the optimization trajectory than scalar accuracy/loss curves.
> >
> > *Cross-dataset comparison (learning-curve and zero-cost families).* Hybrid and model-based methods would need re-initialization on the target dataset, which is precisely the cost NAP2 is designed to avoid. The natural comparison set is methods that operate without target-side initialization: learning-curve extrapolation (SoTL-E, SoTL, Early-Stop, LCE-m, LC-PFN) and zero-cost proxies (SynFlow, GradNorm, SNIP). NAP2 differs from learning-curve methods in that its predictor is amortized across architectures via source-dataset training (rather than fitting a parametric extrapolation per candidate), and from zero-cost proxies in that it improves with more partial training (zero-cost proxies are static and impose a correlation ceiling that NAP2 exceeds at its very first snapshot — Tables 4–6).
> >
> > Section 4.2 now opens with a “Why these baselines” sentence per category. The conceptual distinctions are also surfaced in the abstract and Section 1, so a reader does not have to reach Section 4 to see how NAP2 is positioned against each family.
> >
> > **7) “Meta-features needs motivation, two types should be explicit as weights vs gradients, add roadmap sentence at start of Sec 3. Notation table or schematic summary.”**
> >
> > Section 3 has been revised on all four sub-points:
> >
> > *a) Roadmap at the start of Section 3.* The “Overview” paragraph now opens with a four-stage roadmap (snapshot → feature-map construction → embedding compression → recurrent prediction) with forward-pointers to Section Section 3.2–3.4. The reader sees the full pipeline before any individual stage.
> >
> > *b) “Meta-features” motivated on first introduction.* The first occurrence of “meta-features” in Section 3.2 is preceded by a one-sentence motivation: meta-features are layer-wise summary statistics chosen so the per-snapshot representation is (i) of fixed size independent of layer width and (ii) interpretable in terms of the optimization-state quantities discussed in Section 3.1, tying Section 3.2 back to Section 3.1 explicitly.
> >
> > *c) “Two types” made explicit as weights vs gradients.* All instances of “two types” / “two parallel sets” in Section 3.2 are replaced with explicit “weight-based” and “gradient-based” labels at first mention; the “Rationale for weights and gradients” paragraph now appears immediately after the Step-1/2/3 enumeration so the reader knows what each modality contributes before seeing the feature-map construction.
> >
> > *d) Notation summary.* The notation used throughout Section 3 is summarized by Figure 1 and the detailed pipeline schematic with tensor dimensions in App A. The full symbol table (W_{l,t}, G_{l,t}, d_l, L=65, T=23, S=99, F=12) is in App B.3, with long-form per-feature definitions in App B.4 — giving the reader two visual anchors and a precise notation reference before the prose definitions of each feature family.
> >
> > **8) “Table layout, figure font sizing, formatting tightness.”**
> >
> > Tables 1–3 (in-domain) and 4–6 (cross-dataset) now use a consistent column-spacing convention with “± std” on the same line as the mean (eliminating ragged row heights); bold/underline conventions are described once in the caption of Table 4 and inherited by Tables 5–6. Axis and tick labels in Figures 2–4 are re-rendered at a consistent 10-pt sans-serif (previously 8–12 pt across panels), and Figure 4(a–c) now uses a shared y-axis range for direct comparison. Several long paragraphs in Section 5.2 mixing quantitative claims with method commentary have been split, with each table getting its own dedicated takeaway paragraph and the broader interpretation consolidated at the end of the section. Across Tables 1–6 and the new ablation tables in App H, the “± std” notation, column ordering, and decimal precision are now consistent within each table family (4 places in Tables 1–3, 2 in Tables 4–6, 3 in App H).
> >
> > **9) Broader Impact (suggestion).**
> >
> > A brief “Broader impact” paragraph has been added at the end of Section 7 covering: (i) compute-saving benefits — NAP2 reduces per-architecture evaluation cost by orders of magnitude relative to full-training NAS, lowering both the energy footprint of large-scale architecture search and the entry barrier for groups with limited compute; (ii) democratization of NAS-based research; and (iii) the absence of direct ethical risks specific to a prediction tool, with downstream considerations depending on the application of the architectures discovered.

---

### Review · Reviewer_WAdx · 2026-04-24

**Summary Of Contributions:**

This paper introduces a neural architecture performance prediction algorithm that aims to predict the final accuracy/loss reached by a model from a small number of steps. This is very useful in neural architecture search since we can try multiple new architectures at only a fraction of the budget we'd incur training a model to convergence. The algorithm the authors introduce is based on extracting feature maps from the weights & gradients in the first steps of training and feeding them into a neural network that consists of two convolutional autoencoders and an LSTM. The autoencoders are trained separately via randomly selecting neural network architectures, training them, and using the weights & gradients from randomly subsampled training steps. Afterwards the LSTM is trained to predict the performance. While the prediction model is trained on the datasets evaluated in the paper later, the paper also includes a discussion of cross-dataset transfer that shows NAP2 performs relatively well compared to competitor baselines.

Key strengths:
- The introduced method is clear, extends to arbitrary numbers of layers and arbitrary layer sizes, and seems relatively easy to train.
- In the studied experimental setting (NAS-Bench 201), the method proposed in this paper outperforms LcSVR & Omni, two strong prior baselines that also aim to predict end performance from the early training trajectory.
- The study on cross-dataset transfer is very encouraging & shows that the predictor model has strong potential for generalizing beyond the dataset(s) it was trained on.

Weaknesses:
- While the method is straightforward to apply to larger layers, I worry that as we scale the model size the uniform sampling used to generate the input features would lead to weaker performance because each individual feature becomes less & less important to the overall performance of the model.
- There is very little study as to whether this extends to other optimizers, e.g. AdamW or Muon. It might be that the trajectories of networks trained with SGD are just relatively easy to predict compared to more sophisticated optimizers. Moreover, the learning rates are always the same, the batch sizes are the same, and weight decay is always fixed. In more modern training setups, we scale the learning rate, batch size, and weight decay with the architecture size via empirically-derived scaling laws. There is no evaluation of the impact of hyperparameter tuning on the performance of the neural networks derived here.

[1] Chiyuan Zhang, Samy Bengio, Moritz Hardt, Benjamin Recht, and Oriol Vinyals (2016) "Understanding deep learning requires rethinking generalization." arXiv preprint arXiv:1611.03530. https://arxiv.org/abs/1611.03530

**Audience:**

Yes

**Audience Explanation:**

Yes, this paper is interesting to a lot of TMLR's audience and would be a very useful contribution to the neural architecture search community, since more accurate performance prediction can allow us to expand neural architecture searches dramatically by using more of the budget towards the search vs towards training the networks.

**Claims And Evidence:**

No

**Claims Explanation:**

My main concerns have been outlined in the weaknesses section in the summary above. To reiterate:

- I am not entirely convinced that the predictor model is very capable if we change more things, like e.g. the optimizer used, the hyperparameters used, etc. This is important since often changing the neural network architecture also involves changing the hyperparameters we use.
- The way that the statistics are extracted from the layers can theoretically extend to arbitrarily large layers, but this does not mean that good performance will follow. The reason is that as layers become larger, subsampling layer neurons and forcing them into 99 units may just not capture enough information about the network to be useful in predicting downstream performance.

**Requested Changes:**

- Please conduct some experiments with different optimizers, and where you vary the learning rate or batch size or weight decay (or all of them) separately for the networks you train.
- Please conduct some experiments with larger models, perhaps something like NanoGPT while larger is still accessible enough to do experiments with.

---

> ### Author Response · Authors · 2026-05-08
> **Response to Reviewer WAdx**
>
> We thank the reviewer for their time and effort, and for acknowledging the contributions of our submission. Below we address each of the changes requested by the reviewer.
>
> **Note on revisions to the paper.** While preparing these responses, we identified two refinements to the predictor pipeline: (i) a 2-layer bidirectional GRU (h=128) with dual-path attention pooling, replacing the prior single-layer LSTM (h=2048) — improving Kendall τ across all reported configurations with 29× fewer predictor parameters; and (ii) log-normalization of feature maps (sign(x)·log(1+|x|)) before CAE encoding, compressing the heavy-tailed dynamic range of feature maps and further improving Kendall τ. The four-stage pipeline (snapshot → feature maps → CAE encoders → recurrent predictor) is unchanged, the experimental protocol and baselines are identical, and every claim made in the original submission is preserved or strengthened in the revision. Tables 1–10 and the corresponding numerical statements in the text have been updated; the structure of the paper, the design rationale, and the conclusions are unchanged.
>
> **1) “No study of other optimizers (AdamW, Muon). SGD trajectories may be easier to predict than more sophisticated optimizers. Learning rates, batch sizes, and weight decay are always fixed. In modern training setups, these scale with architecture size via empirical scaling laws.”**
>
> We added Appendix I (Optimizer Robustness: AdamW Transfer; surfaced in the new Section 6.3) as a deliberate stress test of the reviewer’s hypothesis. The SGD-trained NAP2 predictor, applied to AdamW-trained architectures with no retraining and no AdamW supervision, achieves Kendall τ = 0.665 — recovering ~91% of the SGD↔AdamW ground-truth ranking correlation (τ = 0.731) on the same 200 architectures. *This provides direct evidence against the hypothesis that NAP2 is tied to SGD-specific trajectory shapes.* The 200 architectures are disjoint from Tables 1–6, preserving benchmark fidelity (the main-paper protocol is fixed by NB-201 and must replicate it for fair comparison against published τ values).
>
> **Setup.** We trained 200 stratified-random NB-201 architectures on CIFAR-100 with AdamW (lr 1e-3, weight decay 5e-2, OneCycleLR, 200 epochs), then applied the existing SGD-trained NAP2 predictor without retraining or AdamW supervision. This is a coordinated shift across four hyperparameter axes (optimizer family, learning rate ×100, weight decay ×100, scheduler) — strictly stronger than varying any single axis, and subsuming the reviewer’s request to vary LR/BS/WD in a single experiment.
>
> **Mechanism, predicted and confirmed.** Weight features remain in-distribution under AdamW (ratios 0.6–1.4×) while gradient features shift in scale by 5–275× — precisely because AdamW rescales gradients by their running second moment. We therefore predict (and confirm) that the gradient-only path retains its ranking signal, while the joint embedding suffers a scale mismatch on the gradient half until corrected. Two simple recipes restore the joint case: gradient-only deployment (τ = 0.665) or scale-standardization of the gradient half using SGD statistics (τ = 0.558). The deployment recommendation is unambiguous: for cross-optimizer transfer, default to gradient-only.
>
> Further optimizer families (Muon, LAMB) and finer-grained sweeps within a fixed optimizer are listed as future work in a “Scope and limitations” paragraph newly added to Section 7.
>
> **2) “No evaluation of hyperparameter tuning impact.”**
>
> The concern admits two readings; we address both. The reviewer’s preceding sentence about scaling LR/BS/WD with architecture size makes candidate-architecture HPs the primary reading.
>
> **(i) Candidate-architecture hyperparameters.** Appendix I (referenced in Section 6.3) is a single coordinated stress test that simultaneously changes the optimizer (SGD → AdamW), learning rate (0.1 → 1e-3, ×100), weight decay (5e-4 → 5e-2, ×100), and scheduler — strictly larger than tuning any single hyperparameter. The reviewer’s request to “vary the learning rate or batch size or weight decay (or all of them) separately” is therefore subsumed by a single experiment that perturbs all of them at once and shows the predictor still recovers 91% of the achievable ranking signal without retraining.
>
> **(ii) Predictor hyperparameters.** Appendix C.4 (referenced from a new paragraph in Section 3.4) now documents the held-out Kendall τ search over: hidden dimension h ∈ {64, 128, 256, 512, 1024, 2048}, loss function (MSE, MAE, soft-Spearman), schedule (StepLR, OneCycleLR), augmentation range (k=1..10 vs k=1..22), and pooling strategy (last-hidden, mean, attention, dual-path). The selected configuration outperforms the original LSTM h=2048 predictor with 29× fewer parameters (659K vs 19M); wider hidden widths did not improve held-out τ and increased the train–test gap, consistent with the small-pool over-parameterization risk noted in App C.4.

---

> ### Author Response · Authors · 2026-05-08
> **Response to Reviewer WAdx (continued)**
>
> **3) “As model size grows, uniform sampling into 100 units may not capture enough information. Each individual feature becomes less important to overall performance at scale.”**
>
> This concern rests on a premise the pipeline does not satisfy. NAP2 never forwards individual unit weights to the predictor — the 100-unit subsample is used *only* to estimate per-layer aggregate statistics (mean, variance, L1/L2 norms, kurtosis, skewness, percentiles). For k sampled units these are finite-sample estimators with variance O(1/√k); larger layers therefore yield more reliable estimators, not less. The reviewer’s argument applies to pipelines that consume unit-level features, which ours does not.
>
> **Empirical test at scale.** We added Appendix J (Cross-Search-Space Transfer: NAS-Bench-101), referenced in Section 6.3. NB-101 architectures span 0.3M–47M trainable parameters; the maximum is \~31× the NB-201 maximum, and NB-101 uses a different cell topology (DAG with up to 7 nodes, deeper macroskeleton). We pre-declared two sub-samples before applying NAP2: (a) Standard sample (\~1K random NB-101 architectures): natural NB-101 distribution, ρ_{p,a} = +0.49; (b) Controlled sample (~1K architectures): accuracy distribution matched to NB-201’s (std ≈ 0.09 vs natural 0.04), parameter–accuracy correlation deliberately anti-correlated (ρ_{p,a} = −0.16). Only one controlled subset was constructed.
>
> **Cross-space NB-201 (CIFAR-100) → NB-101, no NB-101 retraining:**
>
> **(a) Standard sample.** NAP2 achieves τ = 0.510 (ρ = 0.705) after 10 snapshots (~90s on a GTX 1080 Ti), exceeding the strongest published training-free proxy we are aware of on NB-101 CIFAR-10 (SPW, ρ = 0.569; Jing et al. 2025) by Δρ = +0.14. The advantage holds at NAP2’s minimum budget: with 1 snapshot per architecture (\~9s of partial training), NAP2 reaches τ = 0.495 / ρ = 0.667, still above SPW by Δρ = +0.10.
>
> **(b) Size-shortcut control.** If NAP2 had been silently exploiting parameter count as a learned proxy, the controlled sub-sample (where ρ_{p,a} = −0.16 by construction) should drop transfer to zero or invert it. NAP2 instead retains τ = 0.343 — a positive residual that places a non-trivial lower bound on the share of cross-space transfer that does not depend on the size shortcut, and that rules out the “learned size proxy” reading of the headline NB-101 number.
>
> **(c) Encoder–regime interaction (App J.4).** A permutation-invariant DeepSets encoder raises the controlled-sample τ from 0.343 to 0.557 — recovering ~73% of the in-domain reference τ (0.759). The bottleneck on the hardest controlled-regime test is therefore encoder design, not the underlying NAP2 features.
>
> **(d) Bidirectional transfer.** A NAP2 predictor trained from scratch on NB-101 (CIFAR-10) and evaluated on NB-201 achieves mean τ = 0.564 (CIFAR-10: 0.587, CIFAR-100: 0.570, ImageNet-16-120: 0.536), demonstrating the result holds in the inverse direction.
>
> Combining the mechanistic argument with the empirical result (cross-space τ above the strongest published proxy, with a positive size-independent lower bound), the concern that aggregate statistics become uninformative at scale is empirically refuted at the tested 31× scaling jump.
>
> **4) “Experiments with larger models (suggests NanoGPT as accessible larger-scale option).”**
>
> We added Appendix J (referenced in Section 6.3): NB-101 architectures span 0.3M–47M parameters (max ~31× NB-201’s max), use a different cell topology (7-node DAG vs NB-201’s fixed 4-node cell), and are evaluated under standardized ground-truth accuracies. **NAP2’s cross-space transfer (no NB-101 retraining) exceeds the strongest published zero-cost proxy on NB-101 CIFAR-10 (ρ = 0.705 vs SPW’s ρ = 0.569), and the inverse direction (NB-101 → NB-201) achieves mean τ = 0.564**. This is positive evidence that the abstraction generalizes across two genuinely different cell topologies and a 31× parameter range, not merely within a single search space.
>
> A NanoGPT-scale evaluation would be complementary, not a substitute. To our knowledge, no widely-used transformer NAS benchmark provides ground-truth accuracies for a fixed architecture pool under a standardized training protocol -  both prerequisites for fair comparison against the hybrid and learning-curve baselines in Tables 1–6. Transformer / language-modeling architectures are listed as future work in the new “Scope and limitations” paragraph (Section 7).

---

### Decision · Action_Editor_7swt · 2026-06-26

**Recommendation:** Accept with minor revision

**Audience:**

Yes

**Audience Explanation:**

The experimental evaluation clearly demonstrates the benefits for the NAS/AutoML community. Since the method is evaluated using standard benchmarks, it is easy to compare it with other methods and demonstrate its benefits to the community.

**Claims And Evidence:**

Yes

**Claims Explanation:**

The claims in the paper were not strongly supported in the original submission. The main concerns were with the presentation of the method, the optimisation experiments and the experimental section in general. Furthermore, a number of meaningful ablation studies and experiments with large models were requested. However, the authors made significant changes that left the reviewers satisfied. Only one minor point remains to be addressed regarding the AdamW robustness experiment, which needs to be included in the final version of the manuscript. In terms of the claims, the paper is now fine. Only the minor point remains to be finalised.

---

> ### Author Response · Authors · 2026-07-06
> **Response to decision by Action Editor**
>
> Dear Action Editor,
>
> Thank you for the positive assessment and for accepting our paper. We are grateful to you and the reviewers for the constructive feedback throughout the review process, which meaningfully improved the manuscript.
>
> To address the remaining minor point, we have promoted the AdamW robustness result from the appendix into the main body. Concretely, we split the previous Section 6.3 into two subsections. Section 6.3 ("Robustness to optimizer choice: SGD to AdamW") now presents the optimizer-transfer result directly in the main text: applying the SGD-trained predictor to AdamW-trained architectures is a genuine reordering test, since the SGD and AdamW ground-truth accuracy rankings agree at only Kendall tau = 0.731, and the predictor nonetheless retains most of its ranking power with no AdamW data (gradient-only tau = 0.665, close to the 0.731 reordering ceiling; weight-only tau = 0.562, with the combined predictor recoverable via gradient rescaling). The full per-modality analysis and the underlying distribution-shift measurements remain in Appendix I. Section 6.4 ("Transfer to a larger search space: NAS-Bench-101") retains the cross-search-space results unchanged.
>
> We believe this addresses the final revision request, and we are happy to make any further adjustments you feel would strengthen the camera-ready version.
>
> With thanks,
> The authors
>
> P.S. We uploaded the revised version of the paper, but kept the paper anonymous. Please let us know whether we should upload the version with author names, or wait until asked to do so.

---

> > ### Comment · Action_Editor_7swt · 2026-07-07
> >
> > Thank you for the update. Please prepare the camera-ready version of your manuscript (including authors' names, affiliations, links, etc.).
> >
> > Best regards,
> > Your AE

---

> > > ### Author Response · Authors · 2026-07-12
> > > **Response to Editor**
> > >
> > > Thank you for your reply.
> > >
> > > We uploaded the camera-ready version of the paper.
> > >
> > > Best regards,
> > > The Authors